# Global climate change and the Baltic Sea ecosystem: direct and indirect effects on species, communities and ecosystem functioning

**Markku Viitasalo[1*] and Erik Bonsdorff[2]**

[1] Finnish Environment Institute, Marine Research Centre, Latokartanonkaari 11, FI-00790 Helsinki, Finland

[2] Environmental and marine biology, Faculty of Science and Engineering, Åbo Akademi University, FI-20500 Turku, Finland

*Correspondence to: Markku Viitasalo (markku.viitasalo@syke.fi)

**Abstract**

Climate change has multiple effects on Baltic Sea species, communities and ecosystem functioning, through changes in physical and biogeochemical environmental characteristics of the sea. Associated indirect and secondary effects on species interactions, trophic dynamics and ecosystem function are expected to be significant. We review studies investigating species-, population- and ecosystem-level effects of abiotic factors that may change due to global climate change, such as temperature, salinity, oxygen, pH, nutrient levels, and the more indirect biogeochemical and food web processes, primarily based on peer-reviewed literature published since 2010.

For phytoplankton, clear symptoms of climate change, such as prolongation of the growing season are evident, and can be explained by the warming, but otherwise climate effects vary from species to species and area to area. Several modelling studies project a decrease of phytoplankton bloom in spring and an increase in cyanobacteria blooms in summer. The associated increase in N:P ratio may contribute to maintaining the 'vicious circle of eutrophication'. However, uncertainties remain because some field studies claim that cyanobacteria have not increased and some experimental studies show that responses of cyanobacteria to temperature, salinity and pH vary from species to species. An increase of riverine DOM may also decrease primary production, but the relative importance of this process in different sea areas is not well known. Bacteria growth is favoured by increasing temperature and DOM, but complex effects in the microbial food web are probable. Warming

of seawater in spring also speeds up zooplankton growth and shortens the time lag between
phytoplankton and zooplankton peaks, which may lead to decreasing of phytoplankton in
spring. In summer, a shift towards smaller size zooplankton and a decline of marine copepod
species has been projected.
In deep benthic communities, continued eutrophication promotes high sedimentation
and keeps food conditions for zoobenthos good. If nutrient abatement proceeds, improving
oxygen conditions will first increase zoobenthos biomass but the subsequent decrease of
sedimenting matter will disrupt the pelagic-benthic coupling and lead to a decreased
zoobenthos biomass. In the shallower photic systems, heatwaves may produce
eutrophication-like effects, e.g., overgrowth of bladderwrack by epiphytes, due to a trophic
cascade. If salinity also declines, marine species such as bladderwrack, eelgrass and blue
mussel may decline. Freshwater vascular plants will be favoured but they cannot replace
macroalgae on rocky substrates. Consequently also invertebrates and fish benefiting from
macroalgal belts may suffer. Climate induced changes in the environment also favour
establishment of non-indigenous species, potentially affecting food web dynamics in the
Baltic Sea.
As for fish, salinity decline and continuing of hypoxia is projected to keep cod stocks
low, whereas the increasing temperature has been projected to favour sprat and certain coastal
fish. Regime shifts and cascading effects have been observed in both pelagic and benthic
systems, as a result of several climatic and environmental effects acting synergistically.
Knowledge gaps include uncertainties in projecting the future salinity level as well as
stratification, and potential rate of internal loading, under different climate forcings. This
weakens our ability to project how pelagic productivity, fish populations and macroalgal
communities may change in the future. 3D ecosystem models, food web models and 2D
species distribution models would benefit from integration, but progress is slowed down by
scale problems and inability of models to consider the complex interactions between species.
Experimental work should be better integrated into empirical and modelling studies of food
web dynamics, to get a more comprehensive view of the responses of the pelagic and benthic
systems to climate change, from bacteria to fish. Also, to better understand the effects of
climate change on biodiversity of the Baltic Sea, more emphasis should be placed on studies
of shallow photic environments.
The fate of the Baltic Sea ecosystem will depend on various intertwined environmental
factors, and on development of the society. Climate change will probably delay the effects of
nutrient abatement and tend to keep the ecosystem in its 'novel' state. Several modelling
studies however conclude that nutrient reductions will be a stronger driver for ecosystem
functioning of the Baltic Sea than climate change. Such studies highlight the importance of
studying the Baltic Sea as an interlinked socio-ecological system.
**Keywords**: Climate change, biodiversity, species, communities, food webs, ecosystem
functioning, Baltic Sea
**1.  Introduction**
Global climate change affects the marine ecosystem through ocean warming, acidification,
deoxygenation and through changes in nutrient loading and water circulation, which may all
impact marine biological processes from genes to populations, communities, and ecosystems
(Brierley and Kingsford, 2009; Henson et al., 2017). The biological consequences range from
shifts in species abundance and distributions, changes in dispersal patterns and modification
of species interactions to altered food webs and decreasing ocean productivity (Hoegh-
Guldberg and Bruno, 2010; Philippart et al., 2011; Doney et al., 2012; Burrows et al., 2019).
The changes in biological processes also affect marine ecosystem services and threaten
human food security, especially in the most vulnerable areas (Barange et al., 2014).

Climate change has multiple effects also on the Baltic Sea, impacting species,

communities, and ecosystem functioning. As in the ocean, the effects are usually mediated
via climate affected oceanographic or biogeochemical processes and via associated indirect
effects on species interactions, trophic dynamics, and ecosystem function mechanisms. These
potentially affect the biota inhabiting the Baltic Sea, as well as the human society (Paasche et
al., 2015; Hyytiäinen et al., 2019; Pihlainen et al., 2020; Stenseth et al., 2020).

The effects of climate change on the Baltic Sea ecosystem may differ from those

projected for the oceanic areas as the Baltic Sea differs in many respects from the oceans and
even from the coastal ecosystems surrounding the other regional seas and oceans. The
communities of the Baltic Sea are formed of a peculiar combination of marine, limnetic and
brackishwater taxa. The long winter and the strong seasonal cycle give the area sub-arctic
properties, especially in the northern areas. The Baltic Sea has also been shown to warm up
faster than most other sea areas of the world (Belkin, 2009; Sherman et al., 2009), albeit with
large differences between sub-basins (Kniebusch et al., 2019; Dutheil et al., 2021).The Baltic
Sea is also strongly affected by its watershed, which is more than four times larger than its
surface area and is inhabited by ca. 85 million people (Omran and Negm, 2020). The marine
ecosystem therefore receives excess nutrients and other elements and contaminants from the
land via rivers, through the air, and by leaking from the sediments of the Baltic Sea.
Furthermore, the irregular inflows of more saline and oxic North Sea water, which at specific
basin-wide weather conditions enter the Baltic sea through the Danish Straits (Matthäus and
Schinke, 1994; Lehmann et al., 2022) and influence the state and functioning of the Baltic
Sea.

All these pathways of chemical elements and oceanographic and biogeochemical

processes may be affected by global climate change and the quasi-cyclic climate phenomena
such as the North Atlantic Oscillation (NAO). It has also been suggested that impacts and
symptoms of global climate change are accumulating faster in the Baltic Sea than in other
coastal areas of the oceans, and that Baltic Sea thus can be considered as "a time machine for
the future coastal ocean" (Reusch et al., 2018).

However, attribution of the observed ecosystem changes to global (anthropogenic)

climate change is challenging because of the multiple synergistic effects between climate and
other environmental drivers, such as eutrophication, harmful substances, habitat modification,
fishing and introduction of non-indigenous species, which all may have strong impacts on
ecosystems and their functioning in time and space (Reusch et al., 2018; Stenseth et al., 2020;
Bonsdorff, 2021). Therefore, profound knowledge of the mechanisms and processes
governing Baltic Sea ecosystem under climate change are vital for the understanding and
management of the Baltic Sea (Reusch et al., 2018; Bonsdorff, 2021; Blenckner et al., 2021).

The overall effects of climate change on the Baltic Sea have been reviewed in earlier

synthesis-studies (The BACC Author Team, 2008; The BACC II Author Team, 2015), in
which also climate impacts on the marine ecosystem were assessed (Dippner et al., 2008;
Viitasalo et al., 2015). Since then, a wealth of field, experimental and modelling studies have
shed more light onto the complex interactions between the climate change and the Baltic Sea
system (Meier et al., 2022b).

In this paper, we review research on climate change effects on the Baltic Sea species,

habitats, and ecosystem functioning, primarily based on research published in 2010—2021.
We include both studies investigating direct effects of climate related parameters on
organisms, as well as studies that investigate the more indirect processes affecting the
structure and functioning of the Baltic Sea ecosystem through biogeochemistry and food web
interactions. Evidence is compiled from empirical field studies that show past changes and
responses of species, populations, and communities to climate-affected parameters such as
temperature, salinity, oxygen and pH. A large number of experimental studies, investigating
species responses to the same parameters in micro- or mesocosms, are reviewed. Studies
investigating the complex effects of climate change on the interactions between species and
trophic groups, i.e., phytoplankton, bacteria, cyanobacteria, zooplankton, and fish, as well as
algae or vascular plants and invertebrates grazing on them, are also analysed. Modelling
studies, based on coupled oceanographic-biogeochemical models or other types of species-
level or food web models, are reviewed. Based on the published research we draw
conclusions about the role of climate driven environmental variables on shaping the structure
and functioning of the Baltic Sea ecosystem and identify knowledge gaps and current issues
of dissensus. Areas in need of more research are recommended.


**2. Definitions**

We review studies that shed light to the possible climate effects on the Baltic Sea ecosystem,
by studying oceanographic and biogeochemical parameters which have been projected to
change due to climate change. As such changes may be affected by both anthropogenic
global climate change and by natural climate variations, it is first necessary to define certain
key terms used in this review.

By *global climate change* we refer to the past and contemporary increase in global

temperature, caused by anthropogenic emissions of $CO_2$ and other greenhouse gases, and its
effects on various climatic as well as oceanographic and biogeochemical parameters. By
*climate change*, in turn, we refer to a large-scale shift in climatic parameters affecting the
Baltic Sea region, that may be caused either by global climate change, by cyclic climate
fluctuations (such as North Atlantic Oscillation, NAO) or by irregular or stochastic variation
in climate parameters. We are not considering short-term (between-year or seasonal) weather
variations, but mainly include studies that attempt to reveal organisms' responses to longer
term (several years – decades) variability in climate.

For *ecosystem functioning* we use Tilman's (2001) definition, "the rate, level, or

temporal dynamics of one or more ecosystem processes such as primary production, total
plant biomass, or nutrient gain, loss, or concentration". By *functional diversity* we mean "the
range and value of those species and organismal traits that influence ecosystem functioning"
(Tilman, 2001). A *functional group*, is "a set of species that have similar traits and that thus
are likely to be similar in their effects on ecosystem functioning" (Tilman, 2001).

With *biogeochemical processes*, we refer to various biogeochemical cycles and processes, which often involve cycling and transfer of allochtonous or autochtonous essential nutrients and/or minerals and organic carbon, and which are either driven or influence biological activity in species. With *trophic dynamics* we refer to interactions between trophic levels or functional groups, such as phytoplankton, bacteria, cyanobacteria, nano- and microflagellates, micro-, meso- and microzooplankton, zoobenthos and fish, as well as algae and vascular plants and invertebrates living amongst them.

*Trophic efficiency* is defined as "the efficiency of energy flow between trophic levels, and is the percentage of energy from a trophic level that is used by the organisms of the next trophic level for growth and reproduction" (Hine, 2019).

## 3. Review methods

The search for relevant papers was implemented mainly using Web of Science (WoS) website tool (https://apps.webofknowledge.com/), maintained by Clarivate. The search was focused on years 2010-2021 and was performed using several search terms in various combinations. These included (always) "Baltic Sea" and (in various combinations) "climate", "climate change", "global climate change", "marine ecosystem", "temperature", "salinity", "acidification" and "pH", as well as taxonomic groups such as "phytoplankton", "cyanobacteria", "bacteria", "zooplankton", "microzooplankton", "mesozooplankton", "flagellates", "macroalgae", "zoobenthos", "benthic animals" and "fish" as well as "microbial loop". Marine birds and mammals were not included. The words were used as both search terms Title and Topic, and several reference lists were derived and merged.

Some papers from 2021 and 2022 were found and downloaded with an unstructured search performed with Google Scholar, as this website tool includes more recent publications than WoS. In some cases, references before 2010 were also included, if it was necessary to back up the statements with older studies.

The search resulted in 500+ papers, of which many were not relevant with the current review, i.e., were not concerning effects of climate change on species, habitats or ecosystem functioning in the Baltic Sea. The most relevant studies were saved into library groups of EndNote X9.2 reference management software (Clarivate Analytics), and the contents were scrutinized in more detail.

Because of the focus period, 2010-2021, the review is not a full systematic review of all
research done on climate change effects on the Baltic Sea ecosystem this far. Also, certain
taxonomic groups and study types were less thoroughly reviewed than others. Fish studies in
particular were not comprehensively scrutinized, because the complex responses of fish
populations to climate, eutrophication and fisheries have recently been addressed by a large
number of studies and would merit their own review. Also, we have not reviewed all
experimental studies that have dealt with environmental variables that may change with
climate change. Our goal is to highlight the variety of field, experimental and modelling
studies and to summarise what can be concluded from the recent evidence on the possible
effects of climate change on the Baltic Sea.

**4.  Effects on species and communities**
**4.1. Phytoplankton**
Climate change may have direct effects on the physiology and phenology on phytoplankton,
through physical and chemical parameters, and indirectly through hydrodynamics, e.g.,
stratification, and availability of light and nutrients. Top-down forces, i.e., grazing on
phytoplankton, may also be modified in various ways if grazer populations change.
The growing season of phytoplankton has been significantly prolonged with warming
temperatures during the recent past decades. A satellite-based study suggested that the length
of the period with chlorophyll concentration of at least 3 mg m$^{-3}$ has in the Baltic Sea
doubled, from 110 days in 1998 to 220 days in 2013 (Kahru et al., 2016).  Another study
using phytoplankton sampling data from the Bay of Mecklenburg, western Baltic Sea,
confirmed that the phytoplankton growing season, which in 1988—1992 on average lasted
from March to August, now (2014-2017), now  extends from February to December
(Wasmund et al., 2019), with a longer gap between the spring and late summer peaks. This
prolongation was tentatively explained by increased sunshine in spring and higher
temperature in the autumn, inducing changes in species composition and settling rates of
phytoplankton, remineralization of organic matter by bacteria, and  grazing rates by
zooplankton (Wasmund et al., 2019).
The spring species communities have also shifted from dominance of early blooming
diatoms to later blooming dinoflagellates and the mixotrophic ciliate *Mesodinium rubrum*
(Klais et al., 2011; Hällfors et al., 2013; Kuosa et al., 2017; Hjerne et al., 2019), probably due
to changes in climate and weather patterns, including ice cover, solar irradiation and wind
conditions (Klais et al., 2013; Hjerne et al., 2019). In the long-term data, variable results can
be seen, according to area and dominating species group. For instance, spring phytoplankton
biomass increased in the Baltic Proper but decreased in the Belt Sea area (1979-2005); both
areas showed antagonism between communities dominated by diatoms or dinoflagellates, and
the trends were therefore oscillating rather than linear (Wasmund et al., 2011). Symptoms of
a regime shift were identified, and changes were attributed to approximately 10-year
fluctuations in temperature, salinity, and nutrients. A linkage to global climate change was
not detected (Wasmund et al., 2011).
Some studies have attributed the springtime shifts in phytoplankton phenology and
community structure to changes in environmental conditions driven by global climate change.
A fifteen-year study (2000-2014) using FerryBox observations, covering the area between
Helsinki (Gulf of Finland) and Travemünde (Mecklenburg Bight), confirmed that spring
bloom intensity was mainly determined by winter nutrient concentration, while bloom timing
and duration co-varied with meteorological conditions. The authors conclude that the bloom
magnitude has been affected by the reduction of nutrient loading from land, while bloom
phenology can also be modified by global climate change affecting seasonal oceanographic
and biogeochemical processes (Groetsch et al., 2016).
It has also been suggested that, in the future climate, higher temperatures and less ice
will cause an earlier bloom of both diatoms and dinoflagellates, with increased dinoflagellate
dominance (Hjerne et al., 2019). Experimental (mesocosm) evidence supports findings that
warming up of water and changes in light conditions will accelerate the spring bloom, induce
a decline in peak biomass and favour small size cells, either directly or via increased grazing
by copepods  (Sommer et al., 2012). On the other hand, this development may be
counteracted by increases of windiness and cloudiness, which have also been projected by
certain modelling studies (Hjerne et al., 2019). Recent studies have however indicated that
the projections for spring and summer wind and radiation are uncertain (Christensen et al.,
2022), and future weather changes and associated spring bloom dynamics therefore remain
obscure.
Climate change effects, i.e., temperature increase, salinity decline and acidification
have been shown to have variable results on the toxic dinoflagellate *Alexandrium ostenfeldii*.
Generally, the growth rates, as well as saxitoxin production, increased with higher
temperature and elevated $pCO_2$, but the responses were variable between strains (Kremp et
al., 2012; Kremp et al., 2016). In contrast, temperature or salinity changes did not have a
significant effect on cyst germination of this species (Jerney et al., 2019).

Climate change also increases concentration of water carbon dioxide, a compound

necessary for primary production, and ocean acidification (OA) could therefore enhance
productivity of phytoplankton. However, the results of experimental studies investigating
effects of $pCO_2$ on phytoplankton are variable. The biomass of southern Baltic autumn
phytoplankton (kept in 1400-L indoor mesocosms for 21 days) increased when $pCO_2$ was
increased from 439 ppm to 1040 ppm, also under warm conditions (Sommer et al., 2015). In
other experiments, OA had little effects on community composition, fatty acid composition or
biovolumes of phytoplankton in spring or autumn (Paul et al., 2015; Bermudez et al., 2016;
Olofsson et al., 2019). Even when (positive) effects were detected, they were mainly caused
by an associated decrease of grazing by copepod nauplii in low temperature treatments (Paul
et al., 2016).

There are also studies that have indicated a connection between phytoplankton and the

North Atlantic Oscillation (NAO). A decline in the intensity of NAO in the 1990s was
suggested to have caused by less cloudy conditions, giving more irradiance, and less windy
conditions, inducing stronger stratification of surface water (Hjerne et al., 2019). If the shifts
are driven by variations in NAO or the Baltic Sea Index (BSI, a regional index similar to the
NAO), they may be temporary and reversible, whereas shifts caused by global climate
change may be more enduring.

In the northern Baltic Proper, Åland Sea and the Gulf of Finland, the biomasses of

Chrysophyceae, Prymnesiophyceae and Cyanophyceae have increased and the phytoplankton
biomass maximum, which in the 1980's was in spring and mainly consisted of diatoms, is
now in July-August and is dominated by filamentous cyanobacteria (Suikkanen et al., 2013).
This shift was explained by a complex interaction between eutrophication, climate induced
warming, and increased top-down pressure, as well as changes in DIN:DIP ratio in summer
(Suikkanen et al., 2013). In the Gulf of Bothnia, a gradual decline in salinity was also an
important factor for phytoplankton community change in 1979 to 2012 (Kuosa et al., 2017).

It is obvious that climatic influences are intertwined with other processes and

parameters affecting phytoplankton, especially anthropogenic nutrient loading from land and
internal loading of nutrients from the sediments. There is however a discrepancy on the
relative effects of eutrophication, climate change and other environmental and anthropogenic
factors in explaining past variations in phytoplankton communities and biomass. Also,
several studies have identified complex variations in phytoplankton communities that cannot
be easily explained by any of the studied factors or environmental parameters.
A study comparing historic phytoplankton communities from 1903-1911 with the
present ones (1993-2005) in the northern Baltic Proper and the Gulf of Finland observed an
undefined "period effect", characterized by a decline of diatoms and increase of
dinoflagellates, that was not well explained by the available environmental variables
(temperature, salinity, and general regional climatological data). Although data on
biogeochemical parameters was not available for the period 1903-1911, the authors
interpreted the observed community change as evidence of the direct and/or indirect influence
of eutrophication (Hällfors et al., 2013).
A study investigating summer phytoplankton time series (HELCOM monitoring 1979-
2012) across the Baltic Sea found that there were no common interannual patterns. Instead,
the class trends, e.g. that of cryptophytes, may be affected by anomalies in the BSI, although
a mechanistic explanation for the relationship could not be found (Griffiths et al., 2020).
Other studies did not find any explanation for the observed changes in the biovolumes of
different taxa, e.g. decrease of diatoms and increase of certain dinoflagellate taxa, and
concluded that phytoplankton community in the Baltic Sea is not in a steady state (Olli et al.,
2011), or noted that stochastic dynamics at local scales confound any commonalities between
phytoplankton groups (Griffiths et al., 2020).
To sum up, the past changes in phytoplankton community composition have been very
variable, and usually cannot be explained by a single factor. Some clear signs of climate
change, such as prolongation of the growing season are evident, and can be explained by the
warming and associated biogeochemical processes, but the changes in species and
communities vary from area to area and have multiple reasons, including climate change,
changes in nutrient dynamics, as well as changes in trophic interactions.

**4.2. Cyanobacteria**

Filamentous diazotrophic cyanobacteria benefit from warm temperatures and stratified water,
and they tend to bloom during the hottest and calmest periods of summer (Munkes et al.,
2021). Several modelling studies suggest that the climate induced increase in stratification
(Liblik and Lips, 2019), together with potentially increasing hypoxia, and consequent release
of phosphorus from the anoxic sediments, will increase cyanobacteria blooms in the Baltic
Sea (Meier et al., 2011a; Neumann et al., 2012; Chust et al., 2014; Lessin et al., 2014;
Andersson et al., 2015; Ryabchenko et al., 2016).
Many field studies have also stated that cyanobacteria have already increased along
with the warming of the Baltic Sea. In the northern Baltic Proper, Åland Sea and the Gulf of
Finland, the biomasses of Cyanophyceae have increased, which has been explained by an
interaction between warming, eutrophication and increased top-down pressure on species of
the spring bloom, as well as changes in DIN:DIP ratio in summer (Suikkanen et al., 2013).
Also, in the Gulf of Bothnia, eutrophication and cyanobacteria have increased in
summer (Fleming-Lehtinen et al., 2015; Kuosa et al., 2017), and extensive cyanobacteria
blooms have in the past few years been detected with satellite methods in the Bothnian Sea,
an area usually devoid of such phenomena (unpublished monitoring and satellite records
collected by the Finnish Environment Institute). The increase of cyanobacteria in the
Bothnian Sea has been attributed to an increased freshwater flow and, since 2000, to an
increased intrusion of more saline and phosphorus rich Baltic Proper water into the Bothnian
Sea. These changes have increased stratification, lowered oxygen conditions, and led to a
decline in N:P ratio of the Bothnian Sea, which has favoured the development of
cyanobacteria blooms in the area (Rolff and Elfwing, 2015; Ahlgren et al., 2017; Kuosa et al.,
2017)

It has also been suggested that the various drivers of climate change may contribute to
increase blooms and toxicity of cyanobacteria in the Baltic Sea. For instance, the intracellular
toxin concentration of the cyanobacterium *Dolichospermum* sp. may increase with elevated
temperature (+4°C) (Brutemark et al., 2015; Wulff et al., 2018) and with decreased salinity
(from 6 to 3) (Wulff et al., 2018). As toxins of both dinoflagellates (Sopanen et al., 2011) and
cyanobacteria (Karjalainen et al., 2006; Karjalainen et al., 2007; Engström-Öst et al., 2017)
can accumulate in Baltic Sea zooplankton and induce lower grazing rates and higher
mortality, these studies suggest that toxic dinoflagellates and filamentous cyanobacteria may
get, due to their toxic effects and unpalatability, a competitive advantage over diatoms and
other phytoplankton in a future Baltic Sea.
A few long-term studies have not found an increase in cyanobacteria during the past.
Two recent studies compiling monitoring data from the Baltic Sea for 1979-2012 (Griffiths et
al., 2020) and 1979-2017 (Olofsson et al., 2020) did not find any evidence for an overall
increase of diatzotrophic filamentous cyanobacteria during this period. Biovolume of the
hepatotoxic *Nodularia spumigena* did not change, and that of the non-toxic *Aphanizomenon*
sp. increased in the north and declined in the south (Olofsson et al., 2020). Also, a study that
compared years 1903-1911 and 1993-2005 concluded that cyanophyte biomass has not
increased in summer and have decreased in spring and autumn (Hällfors et al., 2013). It has
been suggested that, although cyanobacteria do prefer warmer temperatures, the effect of
ongoing warming can better be seen in changes in phenology of cyanobacteria rather than as
an increase of biomass (Griffiths et al., 2020). Also, a connection between the amount of
cyanobacteria and the Baltic Sea Index has been identified (Griffiths et al., 2020).

Hypothetically, ocean acidification could benefit cyanobacteria through increased

availability of carbon dioxide in water. The available studies do not give a definitive answer,
however. When $pCO_2$ was experimentally increased, the production of single-celled
cyanobacterium *Cyanothece* increased, while that of *Nodularia* sp. decreased (Eichner et al.,
2014). Also, increase of temperature from 16 to 18—20 °C, led to an earlier peak of
cyanobacteria, while the biomass of cyanobacteria, especially that of nitrogen-fixer
*Dolichospermum* sp. declined (Berner et al., 2018). Further, in mesocosm studies an increase
of $pCO_2$ (from 360 to 2030 µatm) coupled with an increase in water temperature (from 16.6
to 22.4 °C) had a *negative* impact on the biomass of the diatzotrophic cyanobacteria
*Nodularia spumigena* (in 1400-L mesocosms, 28 days) (Paul et al., 2018). Another
experimental study (using 75 ml cell culture flasks), investigating the effects of increased
temperature (from 12 to 16 °C), decreased salinity (from 7 to 4), and elevated $pCO_2$ (from
380 to 960 ppm), found that only temperature had an effect on biovolume and photosynthetic
activity of *Nodularia spumigena* and *Aphanizomenon* sp. (Karlberg and Wulff, 2013). The
two species however had antagonistic effects on each other: biovolumes were lower when
grown together than when grown separately, indicating species interactions.

If the biomasses of *Nodularia* sp. and *Dolichospermum* decrease due to increased

acidification, nitrogen input into the Baltic Sea as well as carbon export to heterotrophic
bacteria via cyanobacteria might decline (Eichner et al., 2014; Berner et al., 2018). This could
however be balanced by the potential increase of *Cyanothece*, which is also a nitrogen-fixer
(Eichner et al., 2014).

To sum up, there are species specific responses to climate change and associated

oceanographic parameters within cyanobacteria. Several field and modelling studies suggest
that the climate induced increase in temperature and stratification, together with increasing
hypoxia and release of phosphorus from the sediments, has increased cyanobacteria biomass
and will continue to favour cyanobacteria blooms also in the future . However, the results of
certain empirical and experimental studies give a more multifaceted picture of cyanobacteria
response to climate change. The past increase of cyanobacteria is not as obvious as might be
expected, responses vary from species to species, and processes affecting amount of
cyanobacteria in the Baltic Sea can be modified, counteracted, or amplified by various
environmental processes and food web interactions.
**4.3. Mesozooplankton**
The Baltic Sea mesozooplankton species originate either from marine or freshwater
environments, and some are typically brackishwater. It is therefore plausible that they
respond to long-term variations in oceanographic parameters. Several field studies have
confirmed that marine copepod species (e.g., *Pseudocalanus* spp. and *Temora longicornis)*
declined during the 1980s and 1990s, while euryhaline and limnetic, smaller-sized copepod
species (*Acartia* spp. and *Eurytemora* spp.) increased in abundance (Suikkanen et al., 2013;
Hänninen et al., 2015), and the decline of marine taxa has usually been proposed to be linked
to a decrease of salinity (Suikkanen et al., 2013; Hänninen et al., 2015). It has also been
experimentally shown that close to the physiological tolerance limit for salinity (below 7
psu), respiration of copepods (*Acartia longiremis*) increases and feeding rate decreases (in
610 ml bottles, 24 h experiments), indicating a disruption of the energetic balance under low
salinity (Dutz and Christensen, 2018).
Environmental impacts on the physiology of the more sensitive species may also affect
the reproductive success of zooplankton (Möller et al., 2015). The increase of euryhaline taxa
has been, directly or indirectly, attributed to the temperature increase (Mäkinen et al., 2017).
It has also been suggested that species that reside in the upper water layers, such as the
copepod *Acartia* sp., are mostly affected by temperature driven increase in food availability,
whereas species inhabiting the deep layers, such as older stages of *Pseudocalanus acuspes*,
are more dependent on salinity and predation pressure (Otto et al., 2014a; Otto et al., 2014b;
Mäkinen et al., 2017).
The effects of climate-driven variations in temperature and ocean acidification (OA) on
zooplankton have been studied experimentally. In *Acartia* sp., warming decreased egg
viability, nauplii development and adult survival (in 1.2-L bottles, during 60-hours) (Vehmaa
et al., 2013). In other experiments, both warming (Garzke et al., 2015) and OA (Vehmaa et
al., 2016) had negative effects on adult female size. This suggests that the projected warming
combined with ocean acidification may have negative effects on the populations of these
copepods in the future Baltic Sea.
Changes in zooplankton functional groups, such as a shift from raptorially and
suspension-feeding copepods and cladocerans to a dominance by small filter-feeding rotifers
and cladocerans, have also been shown as results of warming (Suikkanen et al., 2013;
Jansson et al., 2020). OA also promoted the growth of suspension-feeding cladocerans,
because of a $CO_2$ driven increase of cyanobacteria (Lischka et al., 2017).
Furthermore, a switch from predominantly herbivorous feeding by copepods to
predation on ciliates has been observed in a field study in the southern and central Baltic Sea,
during cyanobacterial blooms (Loick-Wilde et al., 2019). This was caused by decomposing of
the otherwise unpalatable filamentous cyanobacteria, and an associated increase of the
bacteria, nanoflagellates and ciliates (Hogfors et al., 2014). Warming may also increase
zooplankton grazing on medium-large-sized algae, which could contribute to a change
towards smaller-sized phytoplankton species (Klauschies et al., 2012; Paul et al., 2015). It is
therefore possible that the dominant traits of zooplankton communities will change if climate-
induced warming and reduced salinity trends continue. It has also been suggested, from
experimental (mesocosm) evidence, that warming speeds up the growth of copepods but
leaves phytoplankton unaffected, which shortens the time lag between phyto- and
zooplankton. This may lead to a larger and earlier zooplankton peak and increase the
possibility of zooplankton controlling phytoplankton, which may lead to a reduced
phytoplankton biomass under warm temperature (Paul et al., 2016).
Sufficient supply of essential compounds such as amino acids (AA) produced by
phytoplankton and cyanobacteria is essential for the growth and productivity of zooplankton
grazers. A field study performed in the Baltic Proper shows that, during a warm summer,
thermophilic rotifers and cladocerans (e.g. *Bosmina* spp.) acquired ample AA through filter
feeding on the abundant diazotrophic cyanobacteria, whereas the temperate copepods (e.g.
copepods *Temora longicornis* and *Pseudocalanus* spp.) avoided the warm surface layer and
acquired AA mainly through sinking organic matter and/or via grazing on chemoautotroph
based microbial food web in the suboxic zone (Eglite et al., 2018). Mesocosm experiments
have also demonstrated that a high bacterial production can maintain copepod production
(Lefébure et al., 2013), but that increased heterotrophy leads to a decreased fatty acid content
and lower individual weight of copepods (Dahlgren et al., 2011). This may imply that
thermophilic zooplankton species, such as rotifers and certain cladocerans gain more AA than
copepods in a future warmer Baltic Sea.
Little is known on the adaptation capabilities of zooplankton against physicochemical
stress, but some degree of temperature adaptation has been demonstrated experimentally for
the copepod *Eurytemora affinis* (Karlsson and Winder, 2020). Interestingly, the adaptability
was better in populations reared in warm temperatures (≥17°C), which suggests that southern
populations can better cope with increasing temperatures than the northern ones, and that the
adaptation capability of all (surviving) populations may improve with proceeding climate
change.

To sum up, a shift towards smaller size zooplankton and a stronger linkage between

mesozooplankton and the microbial food web is probable in a warmer Baltic Sea. A decline
of certain marine species has also been projected, but this will depend on the future velocity
of salinity decline, patterns of stratification, realized time lag between phyto- and
zooplankton peaks, predation pressure by fish, and on the possible adaptation of zooplankton
species to the subtle changes in salinity.

**4.4. Bacteria and the microbial food web**

Bacteria are key components of the ecosystem, as they decompose organic material, and
serve as food for heterotrophic nanoflagellates and the associated microbial food web. They
affect the nutrient and carbon dynamics of the marine ecosystem and it is therefore possible
that climate impacts on bacteria may radiate to the structure and functioning of the entire
Baltic Sea ecosystem.

The effects of climate induced changes in environmental factors to pelagic bacteria and

the other components of the microbial food web have been studied experimentally. The
effects of projected ocean acidification (OA) on bacteria have been studied alone and also in
combination with with other abiotic variables, such as temperature (OAW) and salinity
(OAS). OA alone had a limited impact on spring bloom microbial communities (sampled
from the sea area around the island Öland in the Baltic Proper and kept in 100-liter
mesocosms for 21 days), but when combined with increased temperature, certain bacterial
phylotypes, such as betaproteobacteria, increased. It was suggested that synergistic effects of
increased temperature and acidification selectively promote growth of specific bacterial
populations (Lindh et al., 2013). In the southern Baltic Sea (Kiel Bight) the impact of OA
was studied in 1400-liter indoor mesocosms for 21-24 days. Acidification only affected few
operational taxonomic units (OTUs), such as *Bacteroidetes* `NS3a marine group´, as the
bacterial community mainly responded to temperature and phytoplankton succession.
Depending on studied season and temperature treatment, Cyanobacteria, *Bacteroidetes*,
Alphaproteobacteria and/or Gammaproteobacteria increased under OA (Bergen et al., 2016).

In an OAS experiment (4-liter aquaria, 12 days) using a natural summer

microplanktonic community, the biovolume of heterotrophic bacteria declined when $pCO_2$
was increased (from 380 to 960 µatm) and salinity was decreased (from 6 to 3 psu) (Wulff et
al., 2018). In experiments done in the Baltic Proper (NW Gotland Sea, 25-liter microcosms
for 23 days), where temperature was increased (from 16 to 18-20 °C) and salinity reduced
(from 6.9 to 5.9 PSU), the microbial community showed mixed responses. No conclusive
evidence of direct climate-induced change could be detected (Berner et al., 2018). At reduced
salinity levels, certain Actinobacteria and *Bacteroidetes* OTUs increased, and the
heterotrophic bacteria community resembled communities at high temperature, indicating
synergistic effects of temperature and salinity. Biotic interactions were more dominant than
abiotic ones, however. The largest increase in heterotrophic bacterial biomass was detected
when filamentous cyanobacteria started to decay, regardless of temperature or salinity. It was
suggested that this indirect coupling between heterotrophic bacteria and filamentous
cyanobacteria is more important for bacterial communities than the direct effects of climate
induced changes in temperature or salinity (Berner et al., 2018).

Experimental studies have demonstrated that complex food web responses to climate

change may also arise. In the Quark, the Gulf of Bothnia, increase of dissolved organic matter
(DOM) enhanced respiration and abundance of bacteria, whereas an increase of temperature
(from 12 to 15 °C) induced a decrease of bacteria, probably due to an increase in
bacterivorous flagellates (Nydahl et al., 2013). A complex response to warming was also
demonstrated for different size classes of heterotrophic flagellates (HF). There was a
succession from flagellates feeding on bacteria to omnivorous nanoflagellates preying upon
other HF. This intraguild predation pattern probably dampened the response to experimental
treatments (Moustaka-Gouni et al., 2016). Mesocosm experiments done in the Gulf of
Bothnia area have however demonstrated that increasing dissolved organic carbon (DOC)
enhances bacterial production and leads to a promotion of heterotrophy (Dahlgren et al.,
2011; Andersson et al., 2013). Also mixing depth influences the ratio of heterotrophic to
autotrophic production: with a shallow pycnocline, the autumn plankton community in the
northern Bothnian Sea remained net-autotrophic irrespective of DOC, whereas with increased
mixing depth and with added DOC the system became net-heterotrophic (Båmstedt and
Wikner, 2016).

As for microzooplankton (MZP), the effects of OA and warming seem to be mostly

beneficial. OA does not have a negative effect on MZP, probably because estuarine MZP are
adapted to a large natural variability in $pCO_2$ (Horn et al., 2016). The abundance of the
mixotrophic ciliate *Mesodinium* sp. even increased in mesocosms with OA, because of
increase of its prey and food, e.g. picoeukaryotes, at higher $CO_2$ levels (Lischka et al., 2017).
In addition, warming improved the growth rate of MZP, and their biomass peaked earlier in
warm mesocosm treatments. This led to a reduced time-lag between MZP and phytoplankton
peaks, inducing a better food supply to microzooplankton in warm conditions (Horn et al.,
2016). The same applied to the MZP-copepod link: at low temperatures MZP escaped from
predation by slower growing copepods, whereas at higher temperatures especially small-sized
ciliates were more strongly controlled by copepod predation.

To sum up, different components of the microbial food web show very variable

responses to climate induced changes in temperature, salinity, and pH. Bacteria growth is
generally favoured by increasing temperature, but mixed effects are common, and indirect
processes affecting decay and availability of organic matter, and abundances of species
predating on bacteria, are also important. This highlights the importance of considering the
effects of abiotic factors and the delicate indirect food web effects on the dynamics of the
microbial food web, and the pelagic ecosystem in general.

**4.5. Macroalgae and vascular plants**

Long-term changes in Baltic Sea macroalgae and charophytes have mostly been explained by
combined or synergistic effects of changes in salinity, wind exposure, nutrient availability
and water transparency (Gubelit, 2015; Blindow et al., 2016; Eveleens Maarse et al., 2020;
Rinne and Salovius-Laurén, 2020), as well as biotic interactions (Korpinen et al., 2007).

For the brown alga bladderwrack *Fucus* spp., light availability, which may also be

affected by partly climate-driven changes in eutrophication, affects their local coverage
(Lappalainen et al., 2019). A documented long-term decrease of water transparency in 1936
to 2017 has reduced favourable sea floor areas for *Fucus* spp. by 45% (Sahla et al., 2020),
and resulted in a halving of the depth range of *F. vesiculosus* in the Åland Islands (Eveleens
Maarse et al., 2020). For many shallow coastal ecosystems of the Baltic Sea, it has been
concluded that eutrophication is the most important pressure affecting the ecosystem
structure and functioning (Olsson et al., 2015). This is plausible, because of the strong
influence of anthropogenic nutrient loading in coastal areas (Vigouroux et al., 2021),
especially those that are prone to hypoxia due to complex topography (Virtanen et al.,
2018a), and which often are affected by internal loading of phosphorus from the sediment
(Puttonen et al., 2014; Puttonen et al., 2016).
The effects of anthropogenic eutrophication on macroalgae may however be amplified
or counteracted by climate induced changes in environmental parameters. Such interactions
are reviewed below.
The direct effects of climate induced changes in temperature, salinity and ocean
acidification (OA) on bladderwrack *Fucus vesiculosus* have been investigated by a number of
experimental studies. OA appears to have a relatively small effect on macroalgae (Al-Janabi
et al., 2016a; Wahl et al., 2019), while temperature effects may be significant. The impacts of
increasing temperature are not linear, however. Growth or photosynthesis is not impaired
under temperatures of 15 to 17.5 °C but at extreme temperatures, simulating heat waves of 27
to 29 °C, photosynthesis declines, growth ceases and necrosis starts (Graiff et al., 2015;
Takolander et al., 2017b). Necrosis is also enhanced by low salinity (4 PSU) (Takolander et
al., 2017b), and under very low salinity (2.5 PSU) the sexual reproduction of *F. vesiculosus*
ceases (Rothäusler et al., 2018; Rothäusler et al., 2019).
The timing of temperature stress is however important for the damage experienced by
algae. Experiments done with bladderwrack *Fucus vesiculosus* (in 1500-L tanks in Kiel, the
western Baltic Sea) showed that in the early and late summer warming of 1 to 5 °C above
ambient had mostly beneficial effects on bladderwrack, whereas in midsummer a similar
warming had negative effects (Wahl et al., 2021). During midsummer, the heat waves
surpassed the physiological tolerance limits of the species, with detrimental effects on
growth.
Ocean acidification combined with warming (OAW) may also act in concert with
hypoxia, in areas where upwellings bring hypoxic water close to the surface. In a three-day
experiment simulating an upwelling event, hypoxic water caused severe mortality of *Fucus*
germlings if they were already stressed by OAW (Al-Janabi et al., 2016b).
Climate induced decline in salinity may affect communities via its direct effect on the
physiology of individual populations and species. A retreat towards the south/south-west has
been predicted for marine species such as bladderwrack and eelgrass, and for species
affiliated to them (Vuorinen et al., 2015). Species distribution modelling studies have
suggested that this mainly salinity-induced decrease of bladderwrack will cause habitat
fragmentation with large effects on the biodiversity and ecosystem functioning of the shallow
water communities of the northern Baltic Sea (Takolander et al., 2017a; Jonsson et al., 2018;
Kotta et al., 2019).
It is not certain to what degree *Fucus vesiculosus* can adapt to the anticipated changes.
It has been suggested that Baltic marine species have, due to local adaptation, isolation and
genetic endemism, diminished potential for adaptation and therefore, an increased risk of
local extinction (Johannesson et al., 2011). E.g., *F. vesiculosus* has long generation time and
relatively restricted dispersal, and therefore the dispersal rate of locally adapted genotypes
may not keep pace with the projected velocity of salinity decline (Jonsson et al., 2018).
However, a growing body of evidence from experimental studies shows that *F. vesiculosus*
has phenotypic plasticity and tolerance against salinity change (Rothäusler et al., 2018; Rugiu
et al., 2018a, b), and genetic studies show that different sibling groups of *F. vesiculosus* have
different responses to environmental change, including OAW (Al-Janabi et al., 2016a; Al-
Janabi et al., 2016b). There may also be population-specific responses to different stressors,
especially if populations are genetically isolated. In a study performed in the Danish Straits,
certain populations of *F. vesiculosus* were only slightly affected by a salinity decline, while
others displayed clearer responses; one population even showed severe stress symptoms and
stopped growing (Kinnby et al., 2020).
It has also been shown that *Fucus radicans*, an endemic congener of *F. vesiculosus*,
which is tolerant to low salinity, might be able to occupy the niche of *F. vesiculosus* in the
northernmost Baltic if salinity declines (Rugiu et al., 2018a). If *F. radicans* can replace the
ecological functions of its congener, its increase may potentially delay or modify the most
drastic consequences of climate change on the invertebrate and fish species dependent on
bladderwrack belts. Changes in species interactions involved in climate induced
environmental changes are however very difficult to project. Some studies project a decrease
of grazers of *Fucus* spp. in the northern areas (Kotta et al., 2019), while others predict an
increase (Leidenberger et al., 2015).
Similar experiments on climate change effects as done with bladderwrack have also
been made with other macroalgae and certain vascular plants. In field mesocosm
experiments, OA increased the growth of the opportunistic green alga *Ulva intestinalis* in the
Kõiguste Bay, Gulf of Riga, Estonia (Pajusalu et al., 2013; Pajusalu et al., 2016). This
suggests that OA could favour such fast-growing species and induce an overgrowth of *Fucus*
sp. by annual green algae.
Salinity decline is projected to decrease the distributional ranges of the marine eelgrass
*Zostera marina* and the red alga *Furcellaria lumbricalis* (Torn et al., 2020) The changing
environment poses an evolutionary risk for populations that live close to the limits of their
geographical ranges (Johannesson et al., 2011), including *Z. marina* (Billingham et al., 2003).
Indeed, mesocosm studies have indicated that, while OA has little effect on the eelgrass
*Zostera marina* (Pajusalu et al., 2015), they suffer from heatwaves in summer (Ehlers et al.,
2008) and elevated temperatures in winter-spring period (Sawall et al., 2021). On the other
hand, the viability of eelgrass beds also strongly depends on water clarity. A study performed
for the southernmost Baltic Sea coupled biogeochemical and species distribution modelling
to assess how projected wind fields, hydrodynamic conditions and nutrient abatement
scenarios affect the distribution of eelgrass *Z. marina* in the future (2062-2066). It was
concluded that nutrient reductions that fulfil the Baltic Sea Action Plan of the Helsinki
Commission (HELCOM BSAP) will lead to an expansion of eelgrass coverage, despite
potentially harmful effects on eelgrass distribution caused by the climate change (Bobsien et
al., 2021).

Certain species may be favoured by the projected climate change. Lowering of salinity

generally favours vascular plants originating from freshwater, and temperature increase
favours thermophilic species, such as charophytes (Torn et al., 2020). In mesocosm studies
made in Kõiguste Bay, photosynthesis of charophytes (*Chara aspera*, *C. tomentosa* and *C.
horrida*) increased under high $pCO_2$ treatments (Pajusalu et al., 2015), which suggests that
they may be favoured by ocean acidification.

To sum up, recent studies suggest that changes in species composition of macroalgae

and vascular plants are likely, due to temperature, pH, and salinity changes. Climate change,
in conjunction with other environmental changes (especially eutrophication)  may also
influence carbon storage in both macroalgae and vascular plants in the Baltic Sea (Röhr et al.,
2016; Takolander et al., 2017a; Jonsson et al., 2018; Salo et al., 2020; Bobsien et al., 2021). It
has been projected that macroalgae will decline in hard bottoms and vascular plants increase
in the more sheltered soft bottom areas (Torn et al., 2020). Because algae and plants mostly
occupy different habitats, the possible increase of vascular plants or charophytes cannot
counteract the negative effects of the disappearance of macroalgae from hard bottom areas.
Consequently also the invertebrates, fish and birds benefiting from habitats formed by
macroalgae will suffer from the climate change.

As with other species groups, projecting the fate of macroalgae and vascular plants is

challenging. This is caused by the uncertainties in projections concerning salinity and
stratification (Lehmann et al., 2022), discrepancy on which physicochemical factors
determine the distribution of invertebrates, unknown adaptation capabilities of algae and
plants, and by uncertainties concerning future trophic interactions within macroalgae and
vascular plant communities.

**4.6. Benthic invertebrates**

Soft bottom benthic communities are dependent on several hydrographic and biogeochemical
variables, and parameters that change with climatic variations have been shown to drive the
long-term progression of zoobenthic communities (Weigel et al., 2015; Rousi et al., 2019;
Ehrnsten, 2020). In the SW coast of Finland, a drastic community change took place, with
amphipods being replaced by Baltic clam *Limecola balthica* and the non-indigenous
polychaete *Marenzelleria* spp. This major community change was explained by an increase in
near-bottom temperature and by fluctuations in salinity and oxygen (Rousi et al., 2013). In
the Åland Islands (northern Baltic Sea), zoobenthos variations in 1983-2012 were associated
with salinity decline, and environmentally driven shifts in the links between zoobenthos and
benthic-feeding fish assemblages were recorded (Snickars et al., 2015). Long-term climate-
induced shifts in zoobenthos and other trophic levels have also been described in various
parts of the Baltic Sea (Törnroos et al., 2019; Forsblom et al., 2021). In all these cases,
interactions between the physico-chemical climate-affected parameters and secondary
impacts (mainly eutrophication and/or hypoxia) have been identified.
Many marine invertebrates will directly and indirectly suffer from decreasing salinity.
In experiments simulating projected changes in temperature and salinity, the survival of the
isopod *Idotea balthica* decreased, albeit with differences between and within regions (Rugiu
et al., 2018c). Also, effects of warming on invertebrates are often non-linear, with stress
effects being manifested after a certain threshold. In experiments, respiration and growth of
the isopod *Idotea balthica* first increased until 20°C and then decreased at 25°C (Ito et al.,
2019). Heat waves, which have been projected to increase in frequency (Meier et al., 2019a),
may therefore pose a severe threat to sublittoral invertebrates (Pansch et al., 2018). Different
species show different responses to single and sequential heat waves, however, resulting in a
change in community structure. E.g., the bivalve *Limecola balthica* suffered from repeated
heat waves, whereas the sessile amphipod *Corophium* sp. benefited from them. The
polychaete *Polydora cornuta* seemed to get acclimated to heat waves when they were
repeated, showing some signs of adaptation. In general, heat waves favoured crawling or
burrowing predators and suspension feeders, while the abundance of detritivores decreased,
suggesting a climate-induced change in dominant zoobenthic traits (Pansch et al., 2018).
Ocean acidification has various effects on benthic invertebrates. The size and time to
settlement of pelagic larvae of the Baltic clam *Limecola/Macoma balthica* increased in
mesocosms (in the western Gulf of Finland) with OA, suggesting a developmental delay
(Jansson et al., 2016), while OA had no effects on larvae of the bay barnacle *Amphibalanus*
*improvisus* originating from Kiel Fjord, southern Baltic Sea (Pansch et al., 2012). Short (12
h) or long-term (2 week) exposures to OA did not have significant effects on the isopod
*Saduria entomon* either (Jakubowska et al., 2013). Controversial results were obtained for the
isopod *Idotea balthica*, from three different sea areas: North Sea, Kattegat, and the Baltic
Sea. The populations from the more oceanic and saline habitats were not significantly
affected by OA, while the Baltic Sea population showed 100 % mortality (Wood et al., 2014).
It was suggested that the Baltic *I. balthica* had higher levels of oxidative stress, and the
combined stress became lethal to them.
Several modelling studies have suggested that climate-induced changes in temperature,
salinity and eutrophication, affecting oxygen levels and food availability for benthos, drive
the development of benthic communities and their biomass in the future (Ehrnsten et al.,
2019a; Ehrnsten et al., 2019b). A physiological fauna model linked to a 3D coupled
hydrodynamic–ecological model projected that, in areas previously burdened by hypoxia,
benthic biomass will increase (until year 2100) by up to 200 % after re-oxygenating bottom
waters, whereas in permanently oxygenated areas the macrofauna biomass will decrease by
35 %, due to lowered food supply to the benthic ecosystem (Timmermann et al., 2012). In
another modelling study, zoobenthic production decreased in the coastal zones, and gradually
also in the more offshore areas, with increasing temperature and declining salinity and bottom
oxygen, regardless of the nutrient load scenarios (Weigel et al., 2015). The fate of zoobenthos
also depends on human intervention, i.e., success of nutrient reduction schemes. For instance,
it has been projected that, if the HELCOM BSAP will be implemented, the biomass of
benthic animals, and hence food for benthic-eating fish, will first increase and then decrease
(Ehrnsten et al., 2020).
There are very few modelling studies focusing on invertebrates inhabiting shallower
hard bottom habitats. One study, where experimental work and species distribution modelling
were combined, projected a decline of the isopod *Idothea balthica* in the future, mainly due
to the salinity-induced decline of its host macroalgae, *Fucus vesiculosus* (Kotta et al., 2019).
Another study reached quite different conclusions. Species distribution models combined
with oceanographic-biogeochemical scenarios for 2050 projected an increase in habitat
suitability for *Idotea balthica* and *I. chelipes*, and concluded that changes in temperature and
ice cover will be more important determinants for these species than changes in salinity
(Leidenberger et al., 2015).
One modelling study also investigated how *Saduria entomon*, a cold-water crustacean
that occupies brackish and limnic parts of the Baltic Sea, will be affected by climate change
and eutrophication (Gogina et al., 2020). The applied models project a net increase (and some
local declines) for *S. entomon*, and conclude that the positive effects of declining salinity will
override the effects of the two nutrient load scenarios (business as usual and BSAP). The
success of this species in the future warmer conditions is also facilitated by its good tolerance
for temperature changes.
It is notable that hypoxia, which is a key factor affecting zoobenthos, is by no means
limited to the deep basins of the Baltic Sea (Conley et al., 2011). Especially the archipelagos
of the northern Baltic Sea are, due to their complex topography and limited water exchange,
prone to hypoxia (Virtanen et al., 2018a). Increasing sea surface temperature will strengthen
stratification and enhance mineralization of organic matter by microbes, which may increase
the release of phosphorus from sediments (Puttonen et al., 2016) and lead to a "vicious circle
of eutrophication" (Vahtera et al., 2007). The sheltered archipelago areas and enclosed bays
may therefore become "climate change hotspots" (Queiros et al., 2021), where also
zoobenthic communities are most drastically changed.
To sum up, zoobenthic communities are affected by all environmental parameters that
are projected to change with climate change, i.e., temperature, salinity, pH and oxygen, as
well as benthic-pelagic coupling. However, the effects are not unidirectional and several
processes may amplify or counteract the possible changes. The magnitude of the future
salinity decline is unclear, and other factors, such as decreasing ice cover and changes in
future wind conditions (of which no consensus exists) may also affect nutrient and oxygen
dynamics of the Baltic Sea. Also, there may be feedback effects on sediment oxygen levels,
as different benthic species have different bioirrigation activities (Norkko et al., 2012; Guy-
Haim et al., 2018). Such processes, that are dependent of traits of a few species, may be of
particular importance in low-diversity systems such as the northern Baltic Sea (Gladstone-
Gallagher et al., 2021).

**4.7. Non-indigenous invertebrates**

It is often suggested that global climate change favours invasions of non-indigenous species
(NIS) worldwide (Jones and Cheung, 2015). This is plausible, because increase of
temperature will open new niches and induce a poleward shift of the ranges of species
inhabiting tropical and temperate sea areas. In the Baltic Sea, it has been shown that non-
native species typically occur in areas characterized by high temperatures, reduced salinity,
high proportion of soft seabed, and decreased wave exposure, whereas most native species
display an opposite pattern (Jänes et al., 2017). This suggest that the former areas are more
prone to climate induced range expansion of non-native species than the latter. This is
consistent with the hypothesis of climate change hotspots, which suggests that some coastal
areas may be more susceptible to effects of climate change than others (Queiros et al., 2021).

Modelled scenarios of temperature and salinity have been used to project how changes

in the abiotic environment could affect NIS already present in the Baltic Sea. One modelling
study suggests an increase of Ponto-Caspian cladocerans in the pelagic community and an
increase in dreissenid bivalves, amphipods and mysids in the coastal benthic areas of the
northern Baltic Sea until 2100 (Holopainen et al., 2016).

To sum up, the global climate change induces many environmental changes that may

favour establishment of NIS in the Baltic Sea. However, attribution of the observed
establishments to the climate change is difficult. It has even been claimed that there is no
conclusive evidence that NIS will gain significant advantage from environmental alterations
caused by climate change (Henseler et al., 2021). Stochastic processes related to maritime
transport and other types of human activities are obviously important for the chances of NIS
to be introduced and established into a given sea area. Long-term surveys, and comparisons
with areas where NIS have not been established, are needed to distinguish climate-related
effects from other ecosystem-level drivers (Bailey et al., 2020).

**4.8. Fish**

Fish populations in the Baltic Sea are influenced by various environmental and anthropogenic
factors, including nutrition, predation, habitat destruction, and fisheries, but also by climatic
variations.

Sprat probably benefits from global climate change, because increasing spring and

summer temperatures have in empirical studies been observed to increase survival of sprat
eggs and larvae (Voss et al., 2012) and in modelling studies to increase productivity and
biomass of sprat (Voss et al., 2011b; Mackenzie et al., 2012; Niiranen et al., 2013).

For herring the results are more variable. The growth rate of herring larvae is positively

affected by temperature (Hakala et al., 2003), but weight-at-age and stock biomass of herring
adults has in several studies been linked to the availability of food, mainly determined by the
abundance of marine copepods and competition with sprat (Flinkman et al., 1998; Möllmann
et al., 2003; Casini et al., 2011; Heikinheimo, 2011; Otto et al., 2014b). In modelling studies
both increase (Bartolino et al., 2014) and a short-term decrease (until 1950) (Niiranen et al.,
2013) of herring populations have been projected.

Both herring and sprat populations have probably benefited from the eutrophication

during the 1950s to 1980s (Eero et al., 2016), during the same period as the Baltic Sea
eutrophication status changed from good to poor (Andersen et al., 2017; Murray et al., 2019).
Since then, sprat biomass has varied independently of nutrient dynamics, and has been more
strongly affected by climatic variation and top-down control, i.e. cod predation and fisheries
(Eero et al., 2016).

Based on experimental and modelling studies, future climatic variations may affect

Baltic cod through their effects on water temperature, salinity, oxygen, and pH, as well as
nutrients, which indirectly affect both the availability and quality of food (Limburg and
Casini, 2019; Möllmann et al., 2021). The responses of cod larvae to ocean acidification
(OA), have been studied experimentally, also in combination with warming (OAW). In some
studies, no effects of OA or OAW on hatching, survival or development rates of cod larvae
were found (Frommel et al., 2013), while in others mortality of cod larvae doubled when they
were treated with high end projections of OA (based on RCP8.5). When the projected
increase of mortality was included into a stock-recruitment model, recruitment of western
Baltic cod declined to only 8 % of the baseline recruitment (Stiasny et al., 2016), suggesting a
dramatic effect of OA on cod populations.

A thorough review including long-term data and modelling demonstrated how

predation, fishing, eutrophication, and climate have sequentially affected eastern Baltic cod
during the past century (Eero et al., 2011). In the early decades of the 20th century, cod
reproduction was successful but seal predation and food availability kept the size of cod stock
at a moderate level. From the 1940s, fishing replaced seal predation in controlling cod
population, whereas the slowly increasing eutrophication had a minor positive influence on
cod spawning stock biomass in 1950s to 1970s. In the late 1970s, a series of large saline
inflows increased the salinity of the Baltic Sea and kept oxygen conditions in the deep basins
favourable for cod. Consequently, reproduction peaked in 1978-1982 and, as also fishing
pressure was temporarily low, the spawning stock biomass increased to a record-breaking
level of ca. 700,000 tonnes in 1980-1984 (Eero et al., 2011). After this peak period, cod stock
started to decline, due to a drastic reduction of the 'cod reproductive volume' (RV), water
layer sufficiently saline and oxic for survival of cod eggs and larvae. The decline of RV was
associated to a stagnation period with low oxygen, caused by a combination of anthropogenic
eutrophication and climate-induced paucity of major saline inflows. Since then, the
productivity of cod stocks has remained low (Eero et al., 2020), and also the average
maximum length of cod individuals has been constantly declining (Orio et al., 2021). The
reason for low growth may have been the low availability of both benthic and pelagic food
(Neuenfeldt et al., 2020). Alternatively, a long-term exposure to low oxygen conditions may
affect body chemistry (Limburg and Casini, 2019) and decrease digestion rate and food
consumption of cod (Brander, 2020). The physiological hypothesis is strengthened by the
observed increase in depth distribution of cod and consequent dwelling of cod in low oxygen
water (Casini et al., 2021).
Several studies project low abundances of cod towards the end of the century, due to
the climate and eutrophication induced decrease of RV (Eero et al., 2011; Gårdmark et al.,
2013; Niiranen et al., 2013; Eero et al., 2020; Wåhlström et al., 2020). It has also been
speculated that seal predation could contribute to keeping cod stocks low. However, although
seal predation can cause damage to cod fisheries in coastal areas (Blomquist and Waldo,
2021), it has been concluded that the increased seal predation is a less important factor for the
future size of fish stocks in the Baltic Sea than climate, eutrophication and fisheries
(Mackenzie et al., 2011; Tomczak et al., 2021).
There is some disagreement on the effect of fisheries on cod stocks in the future.
Earlier studies suggested that fisheries limitations may well enable stock recovery even in a
'cod-hostile' environment (Cardinale and Svedäng, 2011; Heikinheimo, 2011). Certain recent
modelling studies have however been less optimistic, and projected that cod productivity will
remain low, due to the large impact of environmental drivers, especially oxygen and
availability of food (Eero et al., 2020). For the western Baltic cod (inhabiting the Danish
straits and the Arkona Sea) it has even been suggested that cod is now beyond a tipping point,
with severe ecological, economic, and social consequences. At a critical moment, fisheries
management failed to fully consider the changed environmental conditions, and climatic
factors now prevent the recovery of cod stocks (Möllmann et al., 2021).
Increasing seawater temperature has also made it possible for certain warm water
Atlantic species, such as anchovy (Alheit et al., 2012) and sole and turbot (Sparrevohn et al.,
2013) to occur more abundantly in Kattegat and the southernmost Baltic Sea. Such north- and
eastward migrations of these warm-water species may be caused by both global climate
change and by variations in the Northern Hemisphere temperature Anomalies (NHA), North
Atlantic Oscillation (NAO), the Atlantic Multidecadal Oscillation (AMO), as well as
contraction of the subpolar gyre (Alheit et al., 2012; Sparrevohn et al., 2013).
As for coastal freshwater fish, the distribution of pikeperch (*Sander lucioperca*)
expanded towards north along the coasts of the Bothnian Sea, apparently due to the warming
of waters (Pekcan-Hekim et al., 2011). For many coastal piscivores (perch, pike, pike-perch)
and cyprinids, eutrophication status of coastal waters is however a more important factor for
distribution than climatic variation (Snickars et al., 2015; Bergström et al., 2016). A long-
term study covering four decades (1970s to 2010s), made at different coastal areas of the
Baltic Sea, illustrated that it is hard to disentangle the effects of abiotic factors from biotic
interactions affecting fish and their benthic food-sources (Törnroos et al., 2019).
To sum up, temperature, salinity, oxygen and pH have a big impact on Baltic fish
recruitment and growth and, as all these variables respond to climatic variations, it seems
evident that fish communities in the Baltic Sea will undergo changes, with the open sea
ecosystem remaining clupeid dominated, and certain freshwater fish increasing in coastal
areas (Reusch et al., 2018; Stenseth et al., 2020; Möllmann et al., 2021). Together with other
environmental changes, especially eutrophication, changes in fish populations may lead to
altered food web dynamics (Eero et al., 2021), necessitating ecosystem-based management of
fisheries and socio-ecological adaptation (Woods et al., 2021).


**5.  Climate change and ecosystem structure and function**

The Baltic Sea ecosystem is impacted by climate induced changes in the physical and
biogeochemical environment in various ways. Climatic changes affect species and
populations directly and indirectly, also impacting micro-evolution of species and having
synergistic effects on other environmental drivers such as eutrophication and hypoxia
(Wikner and Andersson, 2012; Niiranen et al., 2013; Ehrnsten et al., 2020; Pecuchet et al.,
2020; Schmidt et al., 2020). In synergy, these impacts have already boosted the emergence of
'novelty' in the system and profoundly altered pathways of energy (Ammar et al., 2021). This
development will probably continue, at least if the environmental conditions of the Baltic Sea
will continue to change as projected by modelling studies. Below, recent findings regarding
climate impacts on structure and functioning on the Baltic Sea ecosystem are summarized.

**5.1. Projections of primary production and eutrophication**

For the global ocean it has been projected that climate change will decrease both primary and
secondary production because of intensified stratification and decreased availability of
nutrients in the surface layer (Blanchard et al., 2012; IPCC, 2019). The effects of climate
change on the Baltic Sea ecosystem may however be different, because of the special
hydrographical characteristics, peculiar communities, strong seasonal cycle, and the strong
dependency of the Baltic Sea of both its watershed and the adjacent North Sea.
In the Baltic Sea, changes in ice conditions, water temperature, density stratification,
and especially supply of nutrients through rivers and from the sediment, affect the nutrient
dynamics and primary productivity in both coastal areas and the open sea. Different species
however respond in different ways to changes in the environmental parameters, and both
increases and decreases in primary production have been reported and projected along with
climate induced changes in the environment.
Climate change will most probably mean milder winters and if soils remain thawed,
more nutrients will leak from the terrestrial areas into the freshwater system. The nutrient
load into the sea will probably increase, especially in the northern Baltic Sea where
precipitation is probably increasing the most (Lessin et al., 2014; Huttunen et al., 2015;
Christensen et al., 2022), but also in the southern Baltic Sea (Voss et al., 2011a). It has also
been projected that the total phosphorus loading (from terrestrial areas of Finland) will
increase relatively more than that of nitrogen (Huttunen et al., 2015) and, together with the
internal loading of phosphorus from sediments (Lessin et al., 2014; Stigebrandt et al., 2014;
Stigebrandt and Anderson, 2020), phosphorus availability to primary producers may increase.
If the N:P ratio of the surface layer will decline, the spring bloom will decline and more
excess phosphate will be available for the summer cyanobacteria communities after the
spring bloom (Lessin et al., 2014). This hypothesis concerns especially the Baltic Proper and
the Gulf of Finland, perhaps also the southern Bothnian Sea.
In the central Baltic Sea, increased spring water temperature causes, together with
increased irradiation and enhanced wind-induced mixing of the surface-layer, an earlier but
less intense spring bloom. In summer, in contrast, increase of temperature is coupled with
increased thermal stratification, which is projected to favour production of cyanobacteria
(Meier et al., 2011a; Neumann et al., 2012; Chust et al., 2014; Andersson et al., 2015).
Intensified blooms of cyanobacteria are expected especially if hypoxia will prevail and
internal loading will decrease the N:P ratio (Meier et al., 2011b; Funkey et al., 2014; Lessin
et al., 2014). If the biomass of diatzotrophic cyanobacteria will increase, nitrogen fixation
could also increase, further contributing to the decrease in the N:P ratio (Lessin et al., 2014).
Several modelling studies project an increase in total phytoplankton concentration
(chlorophyll, in mg m$^{-3}$), until the end of the century, with the increase manifested especially
in summer (Meier et al., 2012a; Meier et al., 2012b; Lessin et al., 2014; Skogen et al., 2014;
Ryabchenko et al., 2016). As hypoxia and associated internal loading of phosphorus will
probably be enforced by global warming (Meier et al., 2019b; Tomczak et al., 2021), it has
even been suggested that this "vicious circle of eutrophication" (Vahtera et al., 2007), will
prevent the success of nutrient abatement measures, unless internal loading of phosphorus
will be reduced (Gustafsson et al., 2012; Stigebrandt and Anderson, 2020).
Nutrient abatement may however counteract climate effects. For instance in Kattegat in
mid 1990s, reduction of nutrient loading led to a shift from a highly eutrophic state,
characterized by small phytoplankton species and low water transparency, to an improved
state, with a larger share of diatoms, decreased phytoplankton biomass and increase of water
transparency (Lindegren et al., 2012). An opposing trend has taken place in the Bothnian Sea.
Because of the lack of halocline and lower anthropogenic nutrient loading, the Bothnian Sea
has this far remained in a relatively good condition. However, since year 2000 also the
Bothnian Sea has shown symptoms of eutrophication (Kuosa et al., 2017), and also open sea
cyanobacteria blooms have in recent years become more common, due to  a "leaking" of
phosphorus rich water from the central Baltic Sea through the Åland Sea (Rolff and Elfwing,
2015; Ahlgren et al., 2017). The connection of this process to climate change is not certain.
Rather, the severe hypoxia of the central Baltic Sea has brought the anoxic layer so close to
the sill separating the Baltic Proper from the Åland Sea that flow of nutrient rich water across
the Åland Sea is at times possible. Whether or not the proceeding climate change will amplify
the ongoing eutrophication of the Bothnian Sea remains to be seen, but if temperature
stratification will increase and N:P ratio continues to decline, this will create conditions that
are favourable for cyanobacteria blooms also in this relatively pristine sea area.
Several recent modelling studies conclude that nutrient abatement according to
HELCOM BSAP will in the long run counteract the climate induced increase in nutrient
loading and lead to decreased eutrophication (Meier et al., 2018; Ehrnsten et al., 2019b;
Murray et al., 2019; Pihlainen et al., 2020). Based on oceanographic-biogeochemical
modelling, it has also been suggested that hypoxia will eventually diminish (Meier et al.,
2021) and that extreme cyanobacteria blooms will no longer occur in the future, if nutrient
loadings will be lowered according to BSAP, despite the proceeding climate change (Meier et
al., 2019a).

To sum up, the fate of the level of primary production and level of eutrophication will

depend on various intertwined factors and processes, and on development of both climate and

the society. Changes in primary production will impact interactions between the main trophic

levels, i.e., phytoplankton, detritus and zoobenthos as well as detritivores, benthivores,

grazers, zooplanktivores and piscivores (Kiljunen et al., 2020; Kortsch et al., 2021).

**5.2. Trophic efficiency and consequences to the secondary production**

Recycling and build-up of carbon within the ecosystem determines the overall productivity

and biomass of different trophic levels. Several studies suggest fundamental changes in

trophic dynamics, and eventually in the pathways of carbon in the Baltic Sea.

A climate and nutrient load driven model reconstruction of the Baltic Sea state from

1850 to 2006 suggest that the shift from spring to summer primary production is

accompanied by an intensification of pelagic recycling of organic matter (Gustafsson et al.,

2012). In mesocosm studies warming accelerated (southern Baltic Sea) phytoplankton spring

bloom and increased carbon specific primary productivity (Sommer and Lewandowska,

2011; Sommer et al., 2012; Paul et al., 2016). The total phytoplankton biomass decreased,

because increased stratification decreased nutrient flux to the surface layer, however

(Lewandowska et al., 2012; Lewandowska et al., 2014). Furthermore, in stratified conditions

the relative importance of pathways of carbon through the microbial food web increased

because copepods switched to feed more on ciliates instead of phytoplankton. Decrease of

ciliates in turn increased amount of heterotrophic nanoflagellates grazing on bacteria.

Decrease of bacteria may reduce remineralization and thus decrease availability of nutrients
for phytoplankton (Lewandowska et al., 2014). On the other hand, decreasing of bacteria
would also decrease  competition for nutrients between bacteria and phytoplankton, which
could counteract the negative effects of diminishing remineralisation on phytoplankton.

It has also been projected that, in addition to nutrients, the flow of dissolved organic

matter (DOM) into the Baltic Sea will increase in the future climate (Voss et al., 2011a;
Stråååt et al., 2018). Precipitation will increase especially in the northern areas and, by using
long-term time series from 1994 to 2006, it was shown that climate change has increased
discharge of terrestrial DOM into the middle part of the Gulf of Bothnia. This provided
additional substrate for bacteria, which maintained bacterial biomass production despite
reduced phytoplankton production (Wikner and Andersson, 2012). This suggests that
increased humic-rich river inflow may counteract climate change induced eutrophication in
the coastal waters (Andersson et al., 2013).
Experimental studies have also demonstrated increased microbial activity and biomass
with increasing DOM and temperature (Ducklow et al., 2010), although different bacteria
taxa respond differently to the simultaneous increase of DOM and temperature (Lindh et al.,
2015). Increase of DOM and bacteria may be detrimental to primary production as bacteria
compete for nutrients with phytoplankton, and as the brownification of water reduces light
availability. Consequently, the carbon flow shifts towards microbial heterotrophy, which may
induce a decrease in both phytoplankton productivity and biomass and lead to a promotion of
the microbial food web and other heterotrophic organisms (Wikner and Andersson, 2012;
Andersson et al., 2013). Especially if stratification increases, cycling of carbon through the
microbial food web increases pelagic recycling and may also decrease vertical flux of organic
matter to zoobenthos (Ehrnsten et al., 2020).
It has been suggested that climate change may also decrease fish productivity. In areas
where climate change increases the supply of allochtonous DOM into the system, and where
increasing stratification reduces the transport of nutrients from deeper waters, phytoplankton
production may decline and the trophic pathways from bacteria and flagellates through
ciliates to copepods may strengthen (Aberle et al., 2015). When the system shifts towards
heterotrophy, the food web efficiency declines (Båmstedt and Wikner, 2016), and if
zooplankton also becomes dominated by smaller sized plankton (Dahlgren et al., 2011;
Suikkanen et al., 2013; Jansson et al., 2020), there will be less suitable food available for
planktivorous fish. If also sedimentation of organic matter will be reduced, zoobenthos
production will decrease and there will be less food for benthic-eating fish. Eventually the
total fish production may decrease.
Results of experimental studies have not equivocally confirmed this hypothesis. A
study performed in a large biotest area artificially heated by the cooling waters of the
Forsmark nuclear power plant, southern Bothnian Sea, found that warming of water may lead
to increased species turnover, and in decreased compositional stability of diatom, macrophyte
and invertebrate communities (Hillebrand et al., 2010). Certain mesocosm studies, simulating
effects of climate change in the pelagic ecosystem, have also found that the production and
biomass of both copepods and fish (three-spined sticklebacks) can remain high, because the
positive effects of increasing temperature and increasing availability of DOC override the
negative effects of decreasing food web efficiency on copepod production (Lefébure et al.,
2013).

Furthermore, many Baltic Sea copepods are omnivorous and can opportunistically

switch between suspension feeding on flagellates and raptorial feeding on ciliates (Kiørboe et
al., 1996). Such a flexible feeding strategy stabilizes the system and can sustain copepod
production even under lower phytoplankton production. This flexibility, and the fact that
heterotrophic production increases with high DOC availability, suggests that fish production
may be supported even also when relatively more carbon flows through the microbial food
web (Lefébure et al., 2013).

To sum up, a reorganisation of pathways of carbon is possible in the Baltic Sea due to

the climate change. The system is complex, however, due to several counter- and interacting
processes and large uncertainties in key processes, such as stratification and nutrient loads
from land and the sediments (Meier et al., 2019c; Saraiva et al., 2019), and both increases and
decreases of secondary producers have been demonstrated in field, experimental and
modelling studies. The complexity of the system has been highlighted by a thorough review
which illustrated how changes in benthic-pelagic coupling may induce ecosystem-wide
consequences, via increasing sedimentation of organic matter inducing hypoxic conditions
and internal loading of nutrients (Griffiths et al., 2017).

**5.3. Food web interactions in the sublittoral ecosystem**

If the climate change induces an increase in allochtonous nutrient loads, consequences can be
expected in the communities of algae and vascular plants in the shallow photic zone. The
shallow water food webs based on macroalgae and seagrasses may also be affected by the
indirect effects of climate change, mediated through interactions between algae and their
grazers.

The effects of late summer heatwaves on algae and invertebrates living amongst

bladderwrack *Fucus vesiculosus* have been studied by outdoor mesocosm experiments
(Werner et al., 2016). A heatwave resulted in a collapse of invertebrate grazers, such as
isopods and amphipods, which in turn released grazing on filamentous algae and resulted in
overgrowth of *Fucus* by epiphytic algae. In the autumn and winter, when the biomass of
epiphytes was lower, the process was reversed: warming resulted in intensified grazing on
bladderwrack. Again, a significant reduction of *Fucus* biomass resulted (Werner et al., 2016).
As for the microalgae (diatoms), growing on *Fucus* in spring, temperature effects were
stronger than grazing effects, suggesting a positive overall effect of climate change on
microalgae (Werner and Matthiessen, 2017).

Similar results were obtained in an artificially heated biotest basin (Forsmark nuclear

power plant) in the Gulf of Bothnia, where the biomass of the non-native gastropod grazer
*Potamopyrgus*, gammarids and the snail *Theodoxus* was much higher than in the adjacent
non-heated area. The community shift was mainly driven by direct temperature effects on
invertebrates and by indirect effects of changes in vegetation cover (Salo et al., 2020).
Cascading effects are also possible. In the same biotest basin, perch shifted from feeding on
small fish to gammarid crustaceans, which released grazing pressure from filamentous algae
(Svensson et al., 2017). If similar cascades take place in other coastal sea areas of the Baltic
Sea as well, warming may promote the growth of filamentous algae and contribute to the
decline of bladderwrack.

Decline of bladderwrack will affect other species due to declining availability to habitat

and food (Takolander et al., 2017a; Jonsson et al., 2018; Kotta et al., 2019). Connectivity
between bladderwrack populations as well as organisms inhabiting patches of bladderwrack
may also decline (Jonsson et al., 2020; Virtanen et al., 2020). However, perhaps due to the
complex biotic interactions in  the sublittoral ecosystem, there are very few modelling studies
that have attempted to project the fate of the algal and invertebrate communities inhabiting
the shallow photic zone of the Baltic Sea. Only one study has used a combination of
experimental work and modelling to study the effects of climate change on invertebrates. A
decline of the isopod *Idothea baltica*, the main grazer of *Fucus* spp., was projected due to the
decline of bladderwrack (Kotta et al., 2019).

To sum up, temperature and salinity changes have been projected to affect species

interactions in hard and soft bottoms in the sublittoral zone. Both summer heatwaves and cold
season warming can induce novel trophic interactions that produce eutrophication-like
effects, e.g., overgrowth of bladderwrack by epiphytes, in the photic zone dominated by
macroalgae, even without an increase in nutrient loading. However, as macroalgae are very
much dependent on water clarity, the future level of eutrophication will also affect the fate of
the shallow water communities in the Baltic Sea. The complexity of the system, uncertainty
of the oceanographic projections, and unknown adaptation capabilities of species, make it
challenging to project the future food web interactions in the sublittoral ecosystem.

**5.4. Regime shifts**

In the 1980's a partly climate induced regime shift was recorded with drastic changes in the
central Baltic food web, including phytoplankton, zooplankton and pelagic planktivores and
their main predator, Baltic cod (Möllmann et al., 2009; Lindegren et al., 2010a). In 1980-
2000, a decline in 'reproductive volume' (RV), contributed to the decline of cod population
(Hinrichsen et al., 2011; Casini et al., 2016; Bartolino et al., 2017) and induced cascading
effects on planktivorous fish as well as zooplankton (Casini et al., 2008). The different effects
of temperature and salinity on sprat and cod (see above) also resulted in a spatial mismatch
between these species, which further released sprat from cod predation and contributed to the
increase of sprat stocks in the central Baltic Sea (Eero et al., 2012; Reusch et al., 2018). As
herring is an inferior competitor for food, and food availability per individual declined, the
condition of herring declined (Möllmann et al., 2003; Casini et al., 2010). Transition to a
lower saline Baltic Sea, and associated decline of marine copepods (Hänninen et al., 2015),
also contributed to the observed halving of (3-year old) herring weight-at-age, from 50–70 g
in the late 1970s to 25–30 g in the 2000s (Dippner et al., 2019). The described regime shift
has also been partly questioned, as the descriptions of the shift did not cover the entire food
web (Yletyinen et al., 2016).

A factor that has been less often considered when studying reasons of cod decline is the

interaction with another benthic predator, flounder. Flounder may be both prey for larger cod
and a competitor for the small and juvenile ones. Now that cod size has declined, cod
predation on flounder has decreased, releasing competition for benthic food again. This has
caused more spatial overlap between flounder populations and the remaining small sized cod,
and created more intense competition between flounder and the small sized cod, further
contributing to the decline in body condition of cod (Orio et al., 2020).

Multi-species modelling studies have concluded that both fishing and climate strongly

affects the size of cod stocks. If fishing is intense but climate remains unchanged, cod
declines, but not very dramatically, while if climate change proceeds as projected (according
to the intermediate-high A2 scenario), cod goes extinct in two models out of seven, even with
present low fishing effort (Gårdmark et al., 2013). Different combinations of climate change
and eutrophication scenarios may yield very different outcomes, however. Medium $CO_2$
concentrations (RCP4.5), low nutrients and sustainable fisheries resulted in high biodiversity
and high numbers of cod and flounder, while high emissions (RCP8.5) and high nutrient
loads resulted in diminished biodiversity and high abundance of clupeids, especially sprat
(Bauer et al., 2018; Bauer et al., 2019; Hyytiäinen et al., 2019).

The above studies have mostly considered the ecosystem of the central Baltic. In other

basins, the associated processes and species interactions may be different. E.g., in the
Bothnian Bay, salinity was also a major driver for changes in populations of planktivorous
fish, but the species involved were different. Here the decline of spawning-stock biomass of
herring, observed in 1980-2013, was explained by a simultaneously increased competition
with vendace, a limnic species that had increased with lowering salinity (Pekcan-Hekim et
al., 2016).
In Kattegat, the western Baltic Sea, where the ecosystem is more oceanic than in the
other parts of the Baltic Sea, a regime shift was detected in mid 1990s. Here the shift was
explained by both global climate change, cyclic climate phenomena, and by human
intervention. First, a reduction of anthropogenic nutrient loading led into a shift from a
eutrophic ecosystem state to an ecosystem characterized by decreasing phytoplankton and
zooplankton biomass, dominance by small sized fish in the pelagial, an increase of
macroalgae and filter-feeding molluscs on hard bottoms and other benthic animals in the soft
sediments (Lindegren et al., 2012). Second, the positive phases of NAO and BSI enabled an
inflow of oxygenized water from the North Sea, which improved conditions for zoobenthos,
including the commercially important Norway lobster. A climate induced increase of sea
surface temperatures contributed to the improved flatfish growth and survival in the shallow
nursery areas (Lindegren et al., 2012). Decreasing fishing may also have contributed to the
increase of gadoid and flatfish populations, but its relative importance is difficult to
distinguish from other co-occurring effects.
To sum up, regime shifts are usually a result of several environmental, climatic, and
anthropogenic effects acting synergistically on the entire ecosystem. The climate driven
changes in temperature and salinity have been identified as key drivers for the significant rise
of 'novelty' in both abiotic conditions and biotic assemblages in several basins of the Baltic
Sea (Ammar et al., 2021), but also human contribution, i.e., anthropogenic eutrophication or
its alleviation have also contributed (Reusch et al., 2018). The recent research confirms that
climate change induces multiple direct and indirect effects on species and communities and
affects nutrient and carbon dynamics of the Baltic Sea ecosystem. However, despite the
major structural changes, the overall food web complexity in the central Baltic Sea has
remained surprisingly stable (Yletyinen et al., 2016). The relatively small changes may be
explained by the fact that responses to climate change are not uniform or unidirectional, but
vary from species group to another, within groups, and even between sibling species.
Species-specific responses, many feedbacks, altered trophic pathways, and possibility of
species level adaptation, make projections concerning the state of the ecosystem and trophic
effects challenging.


## 6. Knowledge gaps


The main challenge when analysing effects of climate change on the Baltic Sea is the
possible synergistic effects of climate with other environmental drivers, such as
eutrophication, harmful substances, and introduction of non-indigenous species, which also
may have profound impacts on ecosystems and their functioning (Reusch et al., 2018;
Stenseth et al., 2020; Bonsdorff, 2021). Consequently there are numerous knowledge gaps,
bottlenecks and issues of dissensus that weaken our ability to project the future biological
processes, such as primary and secondary productivity, benthic-pelagic coupling and
hypoxia, interactions between phytoplankton, zooplankton and fish populations, as well as
geographic shifts in macroalgal and invertebrate communities.
Attribution of the observed phenomena to climate change is challenging because of the
collinear, intertwined and interacting processes. Especially difficult is to distinguish the
effects of anthropogenic global climate change from those of quasi-cyclic phenomena, such
as the NAO or BSI, or from other more stochastic variations in climate. This is partly due to
the slow pace of climatic variations and time lags between physical and chemical variations
and ecosystem responses. Quite few studies have investigated a period long enough to cover
any larger number of NAO periods. Especially research into the long-term dynamics of the
food webs is still scarce (Törnroos et al., 2019; Pecuchet et al., 2020; Kortsch et al., 2021).
Field studies have ended up with different conclusions concerning past and present
changes of the environment and the biota, and their causes, depending on time periods and
data scrutinized. For instance, certain studies note that cyanobacteria have increased
(Suikkanen et al., 2013; Kuosa et al., 2017), while others do not find proof for such a
phenomenon (Griffiths et al., 2020; Olofsson et al., 2020). Different periods studied, sparse
sampling, varying species responses, and changes in phenology rather than total biomass,
may explain some of the discrepancies between studies. The tendency of filamentous
cyanobacteria to float during calm weather may also bias our view on the total biomass
cyanobacteria in the sea, especially if low wind periods become more frequent.
Experimental studies are useful in pinpointing causative relationships, but their small
spatial scales, short duration and simple food webs make upscaling of results to natural
systems difficult. Experiments usually only last for a few days or weeks and study one or few
species at a time. Reproducing natural patterns of environmental variability is also
challenging. When mesocosms of hundreds of litres and natural communities are used, it may
be difficult to simulate seasonal processes extending over several life cycles of the studied
organisms. Even the most sophisticated multi-stressor experiments, which use levels of
environmental stressors projected by modelling studies, tend to use constant stress levels.
A few mesocosm studies have exposed the communities to near-natural environmental
conditions and have been able to shed light on the complex dynamics of the Baltic Sea
ecosystem, e.g., the responses of the microbial food web to changes of environmental
variables affected by the climate change. In studies made in the Gulf of Bothnia, bacterial,
phytoplankton and zooplankton production increased with additions of inorganic carbon, and
the systems remained net autotrophic. In contrast, when both nutrients and DOC was
increased, only bacterial and zooplankton production increased, driving the system to net
heterotrophy (Andersson et al., 2013; Båmstedt and Wikner, 2016). Increased heterotrophy
led to a decreased fatty acid content and lower individual weight in the zooplankton
(Dahlgren et al., 2011). With the combined treatment of elevated temperature and terrestrial
nutrient loads, also fish production (of three-spined sticklebacks) increased, with terrestrial
and not autotrophic carbon being the main energy source (Lefébure et al., 2013). The
complex responses indicate that, to provide useful inferences about physiological and
population-level responses of organisms to climate change, experimental work should use full
communities, apply naturalistic exposure regimes, and investigate effects of stress at spatial
and temporal scales appropriate to the species studied (Gunderson et al., 2016).
Ecosystem modelling using coupled oceanographic-biogeochemical models has
advanced greatly in the past 15 years, but significant challenges remain. Projections of sea
surface temperature and ice conditions can be held relatively reliable, but there are still large
uncertainties in projecting salinity, stratification, hypoxia and, hence, the rate of internal
loading (Meier et al., 2022a). Also, natural variability is a larger source of uncertainty in
future projections of hypoxia than previously understood (Meier et al., 2021). Because
salinity, stratification and oxygen strongly affect many Baltic Sea organisms, it is difficult to
project the fate of plankton and benthos communities with certainty. This uncertainty
concerns especially marine species, such as cod, bladderwrack, eelgrass, and blue mussel,
which in many studies have been projected to decrease in the northern basins of the Baltic
Sea. Further, uncertainties are imposed by complex biogeochemical processes in the
terrestrial and freshwater ecosystems, as well as by unknown development of national
economies and farming practices (Huttunen et al., 2015), especially in coastal areas strongly
affected by nutrient loading.

Ecosystem models rarely consider complex biological interactions and feedback

effects, caused, e.g., by multi-species predatory or intraguild relationships. Inclusion of such
effects would require parameterizing the 3D ecosystem models with experiments and results
from multi-species food web models, that operate on the level populations rather than carbon
flows. Also, models cannot at present consider potential adaptation capabilities of species, as
little is known on them. Several recent studies have however pointed out that, e.g.,
macroalgae (Rothäusler et al., 2018; Rugiu et al., 2018a) and zooplankton (Karlsson and
Winder, 2020) have phenotypic plasticity and potential for adaptation against gradual
changes in the abiotic environment.

Food web models offer useful tools for assessing the relative effects of climate,

eutrophication, and other human impacts, including fisheries, on the structure of the Baltic
Sea ecosystem. They could potentially take into account characteristics of species and their
responses to changes in the environment. The current models however mostly concern the
pelagic ecosystems (e.g. cod-sprat-herring-zooplankton food web) and there are major gaps
for key trophic groups, such as macrophytes and macrozoobenthos (Korpinen et al., 2022) as
well as the microbial food web.

3D ecosystem models, food web models and 2D spatial modelling would benefit from

integration. Species distribution models (SDMs) can be produced at a fine spatial scale, even
a few tens of meters (Virtanen et al., 2018b), and in climate change studies they can be
parametrized with 3D model results (Jonsson et al., 2018; Kotta et al., 2019). In the future,
food web models involving relevant coastal taxa could also be used to fill in the missing links
between the large scale (3D) processes and detailed spatial patterns identified by the 2D
models.

Assessing climate effects in a smaller spatial scale would be useful, because shallow

and sheltered bays, lagoons and estuaries may be more susceptible to climate change effects
than deeper offshore areas, and may appear as 'climate change hotspots', where climate
change drives the ecosystem towards a new state (Queiros et al., 2021). The existing coupled
oceanographic-biogeochemical modelling studies however typically have a horizontal
resolution of 1 or 2 nautical miles (ca. 2 or 4 km) and thus cannot easily be used for
projecting local variations in temperature, salinity and stratification within the archipelago or
inside estuaries. A bottleneck for high-resolution 3D models is the poor availability of high-
resolution pan-Baltic bathymetries and forcing data (e.g. wind fields). For the SDMs, in turn,
a major constraint is in many areas the poor availability of detailed species and habitat
mapping data, as well as availability of high-resolution data on benthic substrates.
Considering population level effects on spatial patterns of species would also require
estimation of connectivity between sea areas, a research field that is also under-developed in
the Baltic Sea (Berglund et al., 2012; Jonsson et al., 2020; Virtanen et al., 2020).
Consequently no study this far has considered how the climate change affects microclimatic
patterns in the Baltic Sea, and how different species and habitats may respond to such local
variations.
Due to the above challenges, there are certain discrepancies concerning our view on
the effects of climate change on the structure and function of the Baltic Sea ecosystem. Some
of these issues are highlighted below.
Increased primary production and phytoplankton biomass (measured in chlorophyll a)
have been projected by several modelling studies (Meier et al., 2012a; Skogen et al., 2014;
Ryabchenko et al., 2016). Experimental studies however show that responses of
phytoplankton to climate induced changes in temperature, salinity and pH are variable and
can be modified by simultaneous changes in biogeochemical processes and zooplankton
grazers (Paul et al., 2015; Sommer et al., 2015).  Also, strengthening of stratification and
simultaneous increase in riverine DOM loads may induce a decrease in phytoplankton
production, at least in sea areas where rivers carry large DOM loads (Wikner and Andersson,
2012). Presently it is not clear which of these processes determine primary production in
different sea areas, or whether there are transition areas where the two processes balance each
other, leading to no net change in primary production.
Several ecosystem models also predict an increase of cyanobacteria. As cyanobacteria
blooms are favoured by warm, stabile and conditions and low N:P ratio (Munkes et al.,
2021), increase of vigorous blooms and an increase in nitrogen fixation, could be expected.
However, there are large differences in model projections, due to the unclear relationship
between excess phosphorus and cyanobacteria growth, and the relation between bloom
intensity and nitrogen fixation (Munkes et al., 2021). Cyanobacteria are also not a uniform
group. Some cyanobacteria species benefit from increased temperature and acidification,
whereas others suffer from them (Eichner et al., 2014; Berner et al., 2018; Paul et al., 2018).
Further uncertainty is caused by unknown biological factors, such as stoichiometric elasticity,
impact of viruses and grazers. Thus it is still challenging to project how biomasses of
cyanobacteria and nitrogen fixation will develop in the future.
Recent awareness of marine heatwaves and their potential impacts on the marine
ecosystem has increased our knowledge on how climate change may impact pelagic, benthic,
and littoral communities in the ocean (Pansch et al., 2018; Saha et al., 2020). More studies on
the responses of pelagic and benthic organisms of the Baltic Sea to heat waves would
increase our understanding of the population level consequences of short term variability in
environmental parameters. Research on effects of climate change would also benefit from
methodological diversity. E.g., more extensive use of biochemical and genetic methods, such
as biomarkers (Turja et al., 2014; Turja et al., 2015; Villnäs et al., 2019), stable isotopes
(Voss et al., 2000; Gorokhova et al., 2005; Morkune et al., 2016; Lienart et al., 2021),
compound-specific isotope analyses (Ek et al., 2018; Weber et al., 2021) or metabarcoding
(Leray and Knowlton, 2015; Bucklin et al., 2016; Klunder et al., 2021), as well as
development of remote sensing methods (Huber et al., 2021), could yield novel information
on stress levels experienced by organisms and environmental niches preferred by species.
Such information would allow validation of the biogeochemical models under different
environmental and climate scenarios.
There is some bias in the focus organisms and habitats studied. While experiments on
planktonic organisms and soft bottom animals are relatively abundant, experiments on
macroalgae, vascular plants and invertebrates inhabiting hard bottoms are less abundant, and
studies focusing on the entire food web are scarce. In general, empirical and modelling
studies focusing on climate effects on shallow photic habitats are less abundant than those on
the pelagic and deep benthic habitats (Tedesco et al., 2016). Very few studies have
investigated the shallow water ecosystems holistically, including macro- and microalgae,
invertebrates and fish at the same time. Those that have done so, have revealed complex
interactions and multiple feedbacks between species and ecosystem components (Svensson et
al., 2017; Salo et al., 2020). Also, while there are ample monitoring data on pelagic and deep
benthic communities, similar long-term records are very sporadic for communities associated
with key habitat-forming species such as bladderwrack, eelgrass, blue mussel on hard
bottoms, and vascular plants growing on soft sediments. This lack of empirical data and
subsequent modelling studies hampers our understanding of the long-term responses of
sublittoral communities to climate change.
Furthermore, there is a large body of literature published on sea ice algae and sea ice
ecology in the Baltic Sea (Granskog et al., 2006; Tedesco et al., 2017; Thomas et al., 2017),
and all of them are relevant for studying winter ecology. However, few of them have directly
assessed the effects of climate change on ice ecology in the Baltic Sea. More empirical and
modelling studies including quantitative projections on the effect of diminishing sea ice to
biodiversity and functioning of the Baltic Sea ecosystem in winter and spring would therefore
be desirable.
To sum up, there are still several significant knowledge gaps and issues of dissensus in
our understanding of the effects of climate change on the Baltic Sea ecosystem. To fill these
gaps, the results and conclusions from the experimental work should be better integrated into
the wider empirical and modelling studies of food web dynamics, and more emphasis should
be placed on studying effects of climate change on less studied environments, such as the
microbial food web, sea ice communities, and the sublittoral ecosystem. Such studies would
provide a more comprehensive view of the responses of the pelagic and benthic systems to
climate change in both the open sea and the benthic system, from bacteria to fish (Kortsch et
al., 2021). Also, continuation of both spatial mapping programs and long term ecological
studies will be crucial for validating experimental results and for developing ecosystem
models, advancing our understanding of environmental and meteorological drivers of the
Baltic Sea ecosystem on large spatial and temporal scales.


**7.   Conclusions**

Climate change has an obvious potential to affect entire marine food webs, from coastal to
offshore areas, from shallow to deep, as well as from pelagic to benthic systems. Climate
change can also induce changes in species distributions and proportions, and key nodes and
linkages in the food webs may be altered or lost (Lindegren et al., 2010b; Niiranen et al.,
2013; Leidenberger et al., 2015; Griffiths et al., 2017; Kotta et al., 2019; Gårdmark and Huss,
2020). As many ecosystem services are dependent on the state of the entire ecosystem
(Hyytiäinen et al., 2019), a long-term decline in provision of ecosystem services to humans is
possible. It is therefore indispensable to increase our understanding of the consequences of
climate change on the socio-ecological system of the Baltic Sea and its surrounding marine
regions (Stenseth et al., 2020).
The direct and indirect effects of climate change-related parameters on species,
communities and the ecosystem are summarized in Table 1, based on research done since
2010. While results are variable, some conclusions can be drawn from the evidence this far.
As for the eutrophication status of the Baltic Sea it can be concluded that the ecological
status of the Baltic Sea has not significantly improved despite a decrease in anthropogenic
nutrient loading since the 1980s (Fleming-Lehtinen et al., 2015; Andersen et al., 2017),
largely due to the pervasive internal loading (Murray et al., 2019; Stigebrandt and Anderson,
2020). Success of nutrient abatement largely determines the future state of the Baltic Sea
(Hyytiäinen et al., 2019; Ehrnsten et al., 2020), but climate change may delay, or even
counter the improvement of the ecosystem state (Bonsdorff, 2021).

Climate induced increase of nutrient loading and enhancing of internal loading of

phosphorus have been hypothesized to promote phytoplankton and cyanobacteria production,
and to maintain the 'vicious circle of eutrophication' (Vahtera et al., 2007), and several
modelling studies indeed project an increase in both total phytoplankton biomass and
cyanobacteria blooms in the future (Meier et al., 2011a; Funkey et al., 2014).

Eutrophication process may however be counteracted by various factors. Increase of

DOM flowing via the rivers may decrease both primary and secondary production, at least in
the Gulf of Bothnia (Wikner and Andersson, 2012; Andersson et al., 2013), and certain
cyanobacteria may be negatively affected by increased temperature and ocean acidification
(Paul et al., 2018). Thus, changes in structure and functioning of phytoplankton and
cyanobacteria communities are probable, but the narrative that the global climate change will
inevitably increase phytoplankton biomass and cyanobacteria blooms, and inevitably amplify
the eutrophication of the Baltic Sea, may be too simplistic and needs to be refined by
reconsidering the climate effects on food web processes and nutrient and carbon dynamics.

Also for the deep benthic communities, climate change effects are not straightforward.

If salinity declines, the most marine species will suffer, but according to the latest analyses
undisputable evidence is lacking for a future decline in the salinity of the Baltic Sea
(Lehmann et al., 2022; Meier et al., 2022b). Improvement of oxygen conditions may first
promote higher zoobenthos biomasses but, eventually, increasing stratification will weaken
benthic-pelagic coupling and reduce food availability for benthic organisms. If also nutrient
abatement proceeds favourably, biomass of zoobenthos will start to decline (Ehrnsten et al.,

2020).

In the shallower photic benthic systems, nutrient increase probably enhances

eutrophication, and, if salinity also declines, habitat-forming marine species, such as
bladderwrack, eelgrass and blue mussel, probably decline in the northern Baltic Sea
(Vuorinen et al., 2015; Jonsson et al., 2018; Kotta et al., 2019). As both eutrophication and
increasing temperature favour filamentous algae, continued major changes in the sublittoral
communities can be expected, including negative effects of such algal aggregations (Arroyo
and Bonsdorff, 2016). Of particular concern is the potential loss from rocky substrates of the
habitat forming bladderwrack and red macroalgae. Freshwater vascular plants will be
favoured by freshening of the Baltic Sea, but they cannot replace the marine macroalgae on
rocky sublittoral, because they only grow on soft substrates. On the other hand, salinity
projections are still uncertain (Lehmann et al., 2022), and even if salinity declined, *Fucus*
*vesiculosus* may be able to adapt to salinity changes (Rothäusler et al., 2018).

As for fish, responses also depend on species. Salinity decline and hypoxia increase

will most probably have negative consequences on cod stocks (Gårdmark et al., 2013),
whereas the increasing temperature has been projected to favour sprat (Mackenzie et al.,
2012) and certain coastal fish (Bergström et al., 2016). Again, as projections for salinity,
stratification and oxygen levels are uncertain, the future fate of fish populations cannot be
projected with certainty.

The global climate change induces many environmental changes that may favour

establishment of NIS in the Baltic Sea. Opportunistic and thermophilic species occupying
soft sediments are the most probable winners. It is notable that it is extremely difficult to
eradicate a marine NIS after it has found a suitable niche in the Baltic Sea. As the effects of
NIS on both the ecosystem and the society are usually negative, their spreading should be
prevented already before they enter the Baltic Sea, by effectively eradicating NIS from ballast
waters of ships and other possible vectors.

Climate change is obviously not the only factor determining the fate of the Baltic Sea in

the future. Several modelling studies have concluded that nutrient reductions will be a
stronger driver for ecosystem functions in the Baltic Sea than climate change (Friedland et
al., 2012; Niiranen et al., 2013; Ehrnsten et al., 2019b; Pihlainen et al., 2020; Meier et al.,
2021). In moderate nutrient loading scenarios also climate change will play a role, but under
full implementation of BSAP, the environmental state of the Baltic Sea is projected to
become significantly improved and hypoxia reduced by the end of the century (Meier et al.,
2018; Saraiva et al., 2018, 2019; Meier et al., 2021). Despite the many uncertainties
concerning the effects of climate and eutrophication on the state of the Baltic Sea (Munkes et
al., 2021), it can be stated that continued abatement of anthropogenic nutrient loading,
combined with sustainable fisheries, seem to be the most reliable, albeit slow, measures to
solve the grand challenges of the Baltic Sea (Meier et al., 2018; Murray et al., 2019).

Several studies have focused on studying the effects of climate change on the future

state of the Baltic Sea, and especially the ecosystem modelling studies already provide
valuable results that are directly usable in decision making concerning mitigation of
eutrophication under climate change. In contrast, studies concerning effects of climate change
on biodiversity of the Baltic Sea are lagging behind, and are hampered by model uncertainties
(e.g. for salinity) and by the current inability of models to consider the complex interactions
between species and multiple feedbacks between trophic levels. Especially long-term and
modelling studies focusing on shallow photic environments, which harbour the highest
biodiversity in the Baltic Sea, are sparse. This is a major drawback in a situation where all
major environmental policies, including UN Convention on Biological Diversity and EU
Biodiversity Strategy for 2030, urge for halting the ongoing biodiversity loss. To designate
effective measures to safeguard biodiversity, including a climate smart expansion of the
protected area network, a better understanding of the effects of climate change on the
sublittoral ecosystem is urgently needed.
Knowledge of the mechanisms and processes governing Baltic Sea ecosystem under
climate change have recently accumulated and already provide information that can be used
to design adaptation tools and mitigation measures for the Baltic Sea (Reusch et al., 2018). It
is necessary to continue studying the Baltic Sea as a socio-ecological system, responding to
both environmental and societal changes (Bauer et al., 2018; Bauer et al., 2019; Hyytiäinen et
al., 2019), and to continue the dialogue with human society, in order to attune to the future
changes ultimately driven by the Ocean itself (Stenseth et al., 2020).


**Author contributions.** MV prepared the manuscript with contributions from EB.

**Competing interests.** The authors declare that they have no conflict of interest.

**Acknowledgements.** The work of MV has been financed by the projects SmartSea
(Academy of Finland, Strategic Research Council, grant numbers 292985 and 314225) and
FutureMARES (EU Horizon 2020 grant No. 869300). The Åbo Akademi Foundation is
thanked for financial support for EB, and this contribution is part of the Åbo Akademi
University Strategic Profile *The Sea* (EB). We thank the Baltic Earth secretariat for inviting
us to write this review and three anonymous reviewers for constructive comments
significantly improving earlier versions of the manuscript.


Table 1. Summary of research findings and conclusions on the anticipated effects of climate
change (CC) effects in the Baltic Sea for selected variables. The table only shows studies
published in 2011-2021 and a part of studies referred to in the text are not included. For
earlier studies, see Dippner et al. (2008) and Viitasalo et al. (2015). Observations,
experimental simulations, or modelled projections: T = temperature increase; S = salinity
decline; $TSO_2$ = temperature increase, salinity decline and/or oxygen decline; A =
acidification; AT = acidification and temperature increase; AS = acidification increase and
salinity decline; DOM = dissolved organic matter. EXP = experimental manipulations
/microcosms; MES = experimental manipulations /mesocosms; LTS = Long-term studies;
MOD = modelling studies; FIE = Field data. Empty fields indicate knowledge gaps.

| Taxonomic group | T | S | TSO₂ | A | AT & AS | Other changes in physico-chemical environment | Interactions between trophic levels |
|---|---|---|---|---|---|---|---|
| **Bacterial communit-ies** | EXP: Bacteria community changes under T (Bergen et al. 2016) | EXP: Drastic change in bacterial communities (Wulff et al. 2018) | EXP: Mixed responses to TS change in microbial community (Wulff et al. 2018) | EXP: Limited impact of A on bacteria (Bergen et al. 2016; Lindh et al. 2013) | EXP: Community change with AT (Lindh et al. 2013); EXP: Biovolume of bacterial communities decline with AS (Wulff et al. 2018) | MES: Different responses for increase of DOM and T in different bacteria taxa (Lindh et al. 2015); MES: Bacteria increase with DOC addition (Andersson et al. 2013) | EXP: T induced a decline in bacteria, due to increase of flagellates (Nydahl et al. 2013); EXP: Bacteria increase caused by decaying cyano-bacteria (Berner et al. 2018) |
| **Phyto-plankton** | LTS: Prolonged growing season under T (Kahru et al. 2016; Wasmund et al. 2019). Earlier and longer spring bloom (Sommer et al. 2012; Groetsch et al. 2016; | EXP: Growth rates of *A. ostenfeldii* declined at lowered S (Kremp et a. 2016) Toxicity of *A. ostenfeldii* may increase or decrease, depending on strain, under S | LTS: Eutrophicat-ion effects modified by climate-induced variations in T and S (Hällfors et al. 2013; Olofsson et al. 2020) and by Baltic Sea Index (Griffiths et al. 2020) | MES: Autumn phyto-plankton biomass increased Sommer et al. 2015); EXP: No/minor effects on community composition, fatty acids or biovolumes of | MES: Autumn phyto-plankton biomass increase with AT (Sommer et al. 2015); EXP: Growth and saxitoxing concentrat-ion of *Alexandrium ostenfeldii* increases | LTS: Shift from diatoms to dino-flagellates due to changes in sunshine, wind and ice conditions (Klais et al.2011, 2013;Hällfor s et al. 2013; Spilling et al. 2013; Kuosa et al. | MES: Phytoplankt on increases with increasing inorganic nutrients but not when also DOC is added (Andersson et al. 2013). MES: Warming increases zooplankton grazing on |

| | | | | | | | |
|---|---|---|---|---|---|---|---|
| | Wasmund et al. 2016) EXP: Growth of dino-flagellate *Alexandrium ostenfeldii* decreased under T (Kremp et al. 2016) EXP: Toxicity of *A. ostenfeldii* may increase or decrease, depending on strain, under T (Kremp et al. 2016) Germination of *A. ostenfeldii* resting cysts is unaffected by T (Jerney et al. 2019) | (Kremp et al. 2016) Germination of *A. ostenfeldii* resting cysts is unaffected by T (Jerney et al. 2019) | | phytoplankt on (Paul et al. 2015; Bermudez et al.2016; Olofsson et al. 2019) | with AT (Kremp et al. 2012) | 2017; Hjerne et al. 2019) MOD: Increased phyto-plankton biomass caused by increase in nutrient availability (Meier et al. 2012a,b; Skogen et al. 2014; Ryabchenko et al. 2016); MOD: CC and nutrient reduction lead to a shift from pelagic to benthic primary production (Lindegren et al. 2012) | medium-sized algae which releases smaller algae from predation (Paul et al. 2015); EXP: Effects of AT modified by diminishing of grazing by copepods (Paul et al. 2016) |
| **Cyano-bacteria** | EXP: Earlier peak but lower biomass of cyano-bacteria (Berner et al. 2018); LTS: Increase of cyano-bacteria blooms in summer (Suikkanen et al. 2013); EXP: Toxicity of *Dolicho-spermum* sp. increases with T (Brutemark et al. 2015; Wulff et al. 2018) | LTS: community change caused by S decline in the Gulf of Bothnia (Kuosa et al. 2017); EXP: Toxicity of *Dolichosper mum* sp. increases at low salinity (3-6 psu) (Wulff et al. 2018) | | EXP: Production of single-celled cyano-bacterium *Cyanothece* increases and that of filamentous *Nodularia* decreases under A (Eichner et al. 2014); EXP: Decline of cyano-bacteria may induce a decline of nitrogen fixation (Eichner et al. 2014; Berner et al. 2018) | MES: AT has a negative impact on *Nodularia* biomass (Paul et al. 2018); EXP: Only T affects biovolume and photo-synthetic activity of *Nodularia* and *Aphanizo-menon* (Karlberg & Wulff 2013) | MOD: Cyano-bacteria blooms will increase in the warmer and more stratified future (Meier et al. 2011a,b; Neumann et al. 2012; Chust et al. 2014; Funkey et al. 2014; Lessin et al. 2014; Andersson et al. 2015) | LTS: Shift to cyano-bacteria dominance also attributed to changes in eutrophicat-ion and top-down pressure (Suikkanen et al. 2013) |
| **Microzoo-plankton** | MES: Growth rate | | | MES: No effect on | EXP: Community | | MES: Positive |

| | | | | | | | |
|---|---|---|---|---|---|---|---|
| | of microzoo-plankton increased (Horn et al. 2016) | | | microzoo-plankton (Horn et al. 2016) | change with AT (Lindh et al. 2013); EXP: Biovolume of cilates decline with AS (Wulff et al. 2018) | | effect from A on the mixotrophic ciliate *Myrionecta* (*Mesodinium*) due to increase in food availability (Lischka et al. 2017) |
| **Mesozoo-plankton** | EXP: Decrease in copepod egg viability and nauplii development under T (Vehmaa et al. 2013); Decrease in copepod adult body size and survival (Vehmaa et al. 2013; Garzke et al. 2015); LTS: T favours cladocerans and rotifers (Jansson et al. 2020); EXP: Southern populations of copepod *Eurytemora affinis* can adapt to T (Karlsson & Winder 2020) | EXP: Respiration of copepod *Acartia longiremis* increases and feeding rate decreases at S below 7 psu (Dutz & Christensen 2018) | LTS: Decline of marine copepods due to S (Suikkanen et al. 2013; Hänninen et al. 2015); LTS: Increase of brackish copepods due to S and T (Mäkinen et al. 2017) | MES: A-iduced decline in body size of adult copepods (Vehmaa et al. 2016); | | | MOD: Surface-dwelling copepods are favoured by T-induced increase in food (Otto et al. 2014a); MES: T induces a grazer-driven change to smaller-sized phytoplankton (Klauschies et al. 2012; Paul et al. 2015); MES: At T copepods control micro-zooplankton (Horn et al. 2016); MES: T strengthens hetero-trophic pathways of carbon through protozoo-plankton to copepods (Aberle et al. 2015) and induces a switch from bottom-up to top-down |

| | | | | | | | |
|---|---|---|---|---|---|---|---|
| | | | | | | | control (Paul et al. 2016); MES: Growth of cladocerans increases under A because of increase in food (Lischka et al. 2017) MES: Mesozooplankton production is maintained both with autotrophic and hetero-trophic production (Andersson et al. 2013; Lefébure et al. 2013) |
| **Macroalgae** | EXP: At heat wave temperat-ures, photo-synthesis declines, growth ceases and necrosis starts in bladder-wrack (Graiff et al. 2017; Takolander et al. 2017b); MES: T is beneficial for *Fucus* in early and late summer, but harmful in mid-summer (Wahl et al. 2021) | EXP: Sexual reproduction of bladder-wrack ceases at S (Rothäusler et al. 2018, 2019); MOD: Bladder-wrack distribution will be restricted in the Baltic Sea (Vuorinen et al. 2015; Takolander et al. 2017a; Jonsson et al. 2018; Kotta et al. 2019); MOD: Red alga *Furcellaria* distribution will be restricted (Torn et al. 2020); | | EXP: Generally small effects on macroalgae (Al-Janabi et al. 2016a; Wahl et al. 2019); EXP: Increase in growth of green alga *Ulva intestinalis* due to A (Pajusalu et al. 2013, 2016) | EXP: A-induced necrosis in bladder-wrack is worsened by S (Takolander et al. 2017b); EXP: Upwelling of hypoxic water causes mortality of bladder-wrack germlings under AT Al-Janabi et al. 2016b) | | MES: In spring, T induces overgrowth of bladder-wrack by epiphytic diatoms (Werner & Matthiessen 2017); In summer, heatwave collapses grazers and results in overgrowth of bladder-wrack by filametous algae; in winter, T enhances grazing by invertebr-ates, resulting in decline of bladder-wrack (Werner et al. 2016) |

| | | | | | | |
|---|---|---|---|---|---|---|
| | | EXP: Populations of *F. vesiculosus* show different responses to S (Kinnby et al. 2020) | | | | |
| **Vascular plants** | MOD: Charophyte distribution increases under T (Torn et al. 2020); MES: Springtime heatwaves cause high mortality in eelgrass (Sawall et al. 2021) | MOD: Eelgrass distribution will be restricted by S (Torn et al. 2021) | | EXP: No effect of A on eelgrass *Zostera marina* (Pajusalu et al. 2015) | MOD: Eelgrass distribution will be retained if nutrient abatement is implement-ed, despite CC effects (Bobsien et al. 2021) | |
| **Benthic animals** | EXP: Non-linear response to T in isopod *Idothea balthica* (Ito et al. 2019); EXP: Heat waves induce a shift in community structure (Pansch et al. 2018); LTS: T increase induces a higher biomass ofgammarids and snails (Salo et al. 2020) | LTS: Salinity decline affected zoobenthos variations in Åland Islands (Snickars et al. 2015) | LTS: Long-term changes in physico-chemical parameters drive the variations in zoobenthos (Weigel et al.2015; Rousi et al. 2019; Törnroos et al. 2019; Ehrnsten et al. 2020; Forsblom et al. 2021); LTS: Replace-ment of amphipods by Baltic clam and *Marenzell-eria* sp. explained by TSO (Rousi et al. 2013); EXP: Survival of isopod *Idothea Baltica* decreases | EXP: Develop-ment of Baltic clam larvae slows down under A (Jansson et al. 2016); EXP: No effects of A on barnacle larvae (Pansch et al. 2012); EXP: No effects of A on isopod *Saduria entomon* (Jaku-bowska et al. 2013); EXP: No effect of A on isopod *Idothea balthica* in Kattegat but strong effects in the other parts of the Baltic Sea (Wood et al. 2014) | | MOD: Climate-induced changes in physical and biogeo-chemical parameters will modify the response of zoobenthos to availability of food and oxygen (Timmer-mann et al. 2012; Ehrnsten et al. 2019a,b) | MOD: Abundance of isopod *Idothea baltica* will decline due to salinity-induced decline in bladder-wrack (Kotta et al. 2019) |

| | | | | | | | |
|---|---|---|---|---|---|---|---|
| | | | (Rugiu et al. 2018c); MOD: Biomass of Saduria entomon increases due to S (Gogina et al. 2020) | | | | |
| **Non-indigenous invertebrates** | FIE: T induced higher biomass of gastropod *Potamopyrgus* (Salo et al. 2020) | | MOD: Ponto-Caspian bivalves, amphipods and mysids will increase under TS in the coastal benthic areas (Holopainen et al. 2016); FIE: NIS establish in areas with high T and low S (Jänes et al. 2017) | | | | |
| **Fish** | LTS: Sprat has benefited from T (Voss et al. 2011; MacKenzie et al. 2012; Eero et al. 2016); LTS: Warm water Atlantic species (e.g. anchovy, sole and turbot) occur in the western Baltic (Alheit 2012; Sparrevohn et al. 2013); MOD: Sprat productivity will increase with T (Voss et al. 2011; MacKenzie | LT: S and associated decline of marine copepods induced a halving of herring weight-at-age (Dippner et al. 2019); LTS/MOD: Different effects of T and S on sprat and cod cause a spatial mismatch between these species (Eero et al. 2012); LTS: Decline in S intensified resource competition between herring and | MOD: Cod reproductive volume will diminish towards the end of the century due to TSO (Niiranen et al. 2013; Wåhlström et al. 2020); FIELD, EXP & MOD: Digestion, food consumption, growth and maximum length of cod declines in low O conditions (Limburg et al. 2019; Brander et al. 2020; | EXP: No effect of A on cod larvae (Frommel et al. 2013); EXP: Mortality of cod larvae doubles when treated with RCP8.5 scenarios (Stiasny et al. 2016) | EXP: No effect on cod larvae with AT (Frommel et al. 2013) | LTS/MOD: cod declined due to the climate- and human-induced decrease of 'reproductive volume' (Gårdmark et al., 2013; Niiranen et al., 2013; Wåhlström et al., 2020). | MES: Stickleback production is maintained with elevated temperature and increased DOC loads (Lefébure et al 2013); MOD: Climate-induced decoupling of benthic feeding fish from their food source (Törnroos et al. 2019); MOD: Herring stocks decrease in short term Niiranen et a. 2013); FIE: Perch |

| | | | | | | |
|---|---|---|---|---|---|---|
| | et al. 2012; Pansch et al. 2012); MOD: Herring stocks will increase due to T (Bartolino et al., 2014); LTS: Pike-perch more abundant in the northern-most Baltic Sea due to T (Peckan-Hekim et al.2011) | vendace in the Bothnian Bay (Pekcan-Hekim et al. 2012) | Orio et al. 2021) | | | shift from feeding on small fish to gammarids, which releases grazing from filamentous algae (Svensson et al. 2017); LTS/MOD: Partly climate induced decline in cod stock caused a cascading effect on sprat, herring and zooplankton (Hinrichsen et al. 2011; Casini et al. 2016; Bartolino et al. 2017) |

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
