# Peer review of "Global climate change and the Baltic Sea ecosystem: direct and"

_Earth System Dynamics, 2021_

## Referee Comment (RC2)

Review " Global climate change and the Baltic Sea ecosystem: direct and indirect effects on species, communities and ecosystem functioning" by Markku Viitasalo and Erik Bonsdorff

**General comments**

The review provides a valuable overview of past decades scientific studies with relation to climate change projection and the Baltic Sea.

I miss data synthesis in terms of figures and statistical tests on cross-experimental and cross-ecosystem data, proving significance of made conclusions.

More attempts to weight the importance of different factors would make some scenarios to be presented as more likely than others. Several sections is now a list of different outcomes with seemingly similar probability to occur.

A more critical view on the ability to prove climate effects would enhance the scientific value of the manuscript. The same is true for lack of understanding of adaptive and evolutionary processes for the outcome of projected climate change.

**Detailed comments**

r. 20 Effects of climate would explicitly require statistically significant changes attributed to climate factors. Since this type of data are scarce, scientific "evidence on effect of climate " is unlikely to be found. I suggest a rephrasing.

r. 22 Please specify "responses" of what effectors? By which type of species?

r. 25 "will improve " is ambiguous. Increase or decrease cyanobacterial blooms? Reduced blooms would be an improvement.

r.26 The impact of allochthonous carbon is primarily influenced by the specific loading of organic matter. Not the latitude.

r. 29 Influence of organic matter is primarily hampering photosynthetic production. That cannot be counteracted by the proposed food chain. Please remove or adhere hypothesis better to current knowledge.

r. 44 To uncertainties the adaptation and even evolution of organisms in most trophic levels driven by changed climate should be pointed out. This is not possible to study in short term experiments or modelling.

r.60 The shortcoming of not covering meteorological definition of climate change should be mentioned (i.e., significant differences between 30 year periods). One may even question of it is meaningful to make scientific conclusions of climate effects on chemistry and biology.

r. 88 OAW is defined as abbreviation but OA used below. Please harmonize.

r. 104 Please present the duration of those experiment in relation to organism generation time and discuss its influence on the conclusions that can be made.

r. 109 This could primarily be associated with weather changes given the periods investigated (i.e., mainly within a 30-year period).

r. 188-190 This firm conclusion would merit from a prestation of a strong relationship between eutrophication and "shallow coastal water areas". Please specify what quantities that is used to indicate both factors and the strength of the statistical relationship.

r. 195-196 The presented ranges of temperatures investigated does not appear to include the natural variation observed of the annual cycle. Please comment.

r.221 Good that also adaptation is discussed here. Please also include in the introduction.

r. 255-258 This is one of several examples where direct or indirect effect by climate change on biota is not part of the conclusion (cf. title of the MS). The statement is just a list of factors influencing the organisms today and is assumed to do so in the future, however, without proposing the net outcome of this (i.e. the effect).

r. 267-269 Do you mean that projected climate driven temperature may lead to a rapid increase in hypoxia? increase in temperature? Please rephrase accordingly if so. The conclusion that potential phosphorous release alone will cause eutrophication I premature. Projected enhancement of precipitation and river discharge of organic matter may counteract this by reducing light irradiance to the water column.

r. 305 Please specify what analysis you refer to? The modelling?

r. 331-335 The main sentence and the subordinate cluse appear contradictory. If the factors are difficult to disentangle, how can you then derive significant climate factors? Please clarify and rephrase.

r. 501 Or could it be referred to as a shift in weather conditions?

r. 571 As pointed out above few if any studies including biological variables cover at least 2 climate periods. Most also lack coverage of adaptive and evolutionary processes. This should be recognized and the statements rephrased accordingly.

r. 575 Please correct and shorten the sentence. Message is unclear.

r.583 "…will promote cyanbacterial blooms…". "Improve" is ambiguous.

r. 585 The dominating effect of reducing photosynthesis is overlooked (reduced light irradiance and intensified competition for the limiting nutrient with bacterioplankton). Please include and rephrase. Again, the proposed food chain cannot counteract this. The sentences are also close to repetition of what is said in the abstract. Consider replacing by complementing text.

r. 604. I am sceptic that cyanobacterial bloom would be markedly reduced as they are also found in sediment records representing pre-industrial conditions. Consider rephrasing sentence.

Table 1. Effects by increased precipitation and discharge of organic matter is overlooked. This primarily influence phytoplankton carbon dioxide fixation but also bacterioplankton and other parts of the food web. This is demonstrated both in long-term field data and controlled mesocosm experiments.

---

## Referee Comment (RC3)

Review 1 of the manuscript "Global climate change and the Baltic Sea ecosystem: direct and indirect effects on species, communities and ecosystem functioning" by Markku Viitasalo1 and Erik Bonsdorff submitted to Earth System Dynamics Discussion (https://doi.org/10.5194/esd-2021-73).

**ESD specific reviewer statements**

1. Does the paper address relevant scientific questions within the scope of ESD?

   Yes

2. Does the paper present novel concepts, ideas, tools, or data?

Not yet so much. There are new tools (or methods) to better bridge gaps in cross-compartment community structure analyses (e.g. by fuzzy coding or metabarcoding). But there are also new methods (namely compound specific isotope analyses of amino acid nitrogen, CSIA) that bridge gaps in cross-compartment functional diversity analyses more directly and which, different to other "multi trait" approaches, include significant reduction of complexity by addressing functional groups (according to Tilman 2001: phototrophs, mixotrophs, heterotrophs, herbivores, different levels of carnivores,) rather than individual traits or taxonomic groups. CSIA allows for the direct measurement of the mean trophic position of a field sample as one key cross-compartment functional trait (according to Tilman 2001) and of the dominant inorganic nitrogen source used for growth from a field sample as second key cross-compartment functional trait (Tilman 2001) for end-to-end analyses (e.g., physics to fish to human sectors, in sensu Peck et al., 2018) of food webs. Both traits can directly be determined from a single field sample independent from the compartment it comes from (e.g. mixed phytoplankton, mixed mesozooplankton, bivalves, herring, cod, seagulls) and can directly be used as "common currency" across all compartments (e.g., physics to fish to human sectors, in sensu Peck et al., 2018) to calibrate and validate current biogeochemical models. For example, no biogeochemical model yet accounts for the mass and energy loss for fish that must be related to the shift in mean trophic position of mesozooplankton from herbivory (TP of 2) to carnivory (TP of 3) during cyanobacterial blooms in the central Baltic Sea (see specific comment #) and which can help to explain the loss in apex predators in the Baltic Sea.

3. Are substantial conclusions reached?

   Yes, but they are sometimes inconsistent with parts in the earlier text. For example, I still don´t know if the research up to now points to $N_2$-fixing, unpalatable cyanobacteria as "winner" of cc or not. So conclusions should be somewhat refined.

4. Are the scientific methods and assumptions valid and clearly outlined?

   Generally yes. Yet, I feel like important empirical field based studies are still missing in this review as marked in the specific comments.

5. Are the results sufficient to support the interpretations and conclusions?

See point 3.

6. Is the description of experiments and calculations sufficiently complete and precise to allow their reproduction by fellow scientists (traceability of results)?

Pls explain how you found and chose your reviewed literature (e.g. did you use google scholar, did you visit the webites of major Baltic Sea research institutes for most recent publications etc.?).

7. Do the authors give proper credit to related work and clearly indicate their own new/original contribution?

Not applicable for an invited review article, I think.

8. Does the title clearly reflect the contents of the paper?

Authors should definitely add a definition of ecosystem functioning into your introduction or as a glossary, e.g. Ecosystem Functioning (Tilman 2001): The rate, level, or temporal dynamics of one or more ecosystem processes like primary production or nutrient gain or loss.

9. Does the abstract provide a concise and complete summary?

I will suggest a major revision and there might be changes in the abstract necessary due to that. For example, what about the internal P storage in the sediments that gets released under anoxic conditions? How will that influence the projection of future cyanobacterial blooms in the Baltic Sea?

10. Is the overall presentation well structured and clear?

The authors could more systematically show, which direct and indirect effects have been addressed by which kind of research. Generally, I feel that new knowledge based on empirical research did not always find its way into here, if the "titel" did not include certain key words. See also reply to 6.

11. Is the language fluent and precise?

There are some minor spelling and grammar errors that need to be corrected for.

12. Are mathematical formulae, symbols, abbreviations, and units correctly defined and used?

Not applicable.

13. Should any parts of the paper (text, formulae, figures, tables) be clarified, reduced, combined, or eliminated?

The structure could be sharpened a little bit maybe. For example, I find it more intuitive to start with phototrophs like phytoplankton and cyanobacteria rather than with "Microbial communities". Also, cyanobacteria is a very broad group. Maybe palatable cyanobacteria should be differentiated from unpalatable ones, as well as those that do fix nitrogen from those that don´t fix nitrogen but only profit from leacking out of diazotroph nitrogen from the N2-fixing ones. More in the revised pdf attached.

14. Are the number and quality of references appropriate?

There are important papers missing at the moment, I think. Pls see the revised pdf for details.

15. Is the amount and quality of supplementary material appropriate?

Not applicable.

**General comment**

I enjoyed reading and reviewing this manuscript and I think it's worth publishing. That's said, I think that there are still many improvements possible to make it even better. For example, the authors did not find yet a good balance between a "too detailed" revision of a study and a "to brief" review of a study. In many places, I have a quarrel with the statements of many studies, which are often generalized beyond recognition (anytime I request: "Pls be more specific"). Maybe the authors can add some details here and there, namely where mechanisms are mentioned but which are hardly explained in sufficient detail (pointed out below in the specific comments).

The scientific community currently is struggling to find "the right" definition of functional diversity and other terms and how to investigate functional diversity. I think it's worth crediting the different approaches and the dissent. Some methods allow for an indirect approach to study functional diversity (those including taxa), others a direct approach (those including only functional groups). A review is a chance to point out methodological improvements over the last years and the review would gain relevance if some cross-compartment approaches are included here (e.g. specific comment #79).

**Specific comments**
1) p 1, ll. 3: add a definition of ecosystem functioning into your introduction or as a glossary, e.g. Ecosystem Functioning (Tilman 2001): The rate, level, or temporal dynamics of one or more ecosystem processes like primary production or nutrient gain or loss

2) p. 4, ll. 60-64: After Tilman (2001) the definition of ecosystem functioning (EF) is: The rate, level, or temporal dynamics of one or more ecosystem processes like primary production or nutrient gain or loss.

My question: How do you define EF in this review and which processes do you include in your review and which do you not include and why (e.g. are there other reviews out to refer to like "Wannicke, N., Frey, C., Law, C. S., & Voss, M. (2018). The response of the marine nitrogen cycle to ocean acidification. Global Change Biology, 24(11), 5031-5043.")? Pls add this information into your text.

Pls add a side note with definitions for the most prominent terms in your review, e.g.
-climate change
-ecosystem functioning
- trophic dynamics etc.

3) p. 4, ll. 68: wording→ „...more light onto (not into) the complex.."

4) p. 4, ll 67-69: I understand that there is an upcomming review on the cc projections associated with this review. Still, for the "stand alone status" of this review it would be very helpful, to specify here in more detail the projected "abiotic" changes that possibly are most important for the biology (maybe less the atmospheric forcing behind them) including namely salinity, temperature, stratification and oxygen as well as OA, nitrate and phosphate levels, the latter from both, rivers and the anoxic sediments. Important to include: How certain or uncertain can we be about them (e.g. in line 537-539)?

5) p. 4, ll. 70: add: „the" before year

6) p. 4, ll. 71: add „field based" before "responses"

7) p. 4, ll. 75: „food web dynamics", what do you mean by this? Pls specifiy.

8) p. 4, ll. 75: add „both directly in the field as well as in experimental studies." At the end of the sentence.

9) p. 4, ll. 76: What kind of modeling studies, pls specify.

10) p. 4, ll. 85: Why don´t you start with the autotrophic communities instead of the heterotrophic microbial community in Chapter 2?
Phototrophs form the base of the food web also for the heterotrophic microbial community that you seem to mainly refer to in Chapter 2.1.
Also, you should define, which organisms you mean with "microbial community", e.g. only heterotrophs?
What about marine viruses and fungi?

11) p. 4, ll. 87: add "(OA)" after ocean acidification

12) p. 4, ll. 90: How did community change? Be more specific.

13) p. 4, ll 91: Which OTUs? Be more specific.

14) p. 4, ll. 92: How did the microbial community respond? Be more specific.

15) p. 5, ll. 94: Give range also for CO2 as for Sal and Temp.

16) p. 5, ll. 99 delete bracket before "Berner"

17) p. 5, ll. 99 delete extra dot after et al.

18) p. 5, ll. 100-101 Changed drastically to what? Be more specific. What does a "high temperature community" look like?

19) p. 5, ll. 102-103 Which increase? Unclear where you refer to here.

20) p. 5, l. 104 (end of paragraph) What about the effect of other abiotic variables like stratification, oxygen, nitrate and phosphate on heterotophic bacteria, viruses and fungi?
Möller, L., Kreikemeyer, B., Gerdts, G., Jost, G., & Labrenz, M. (2021). Fish as a winter reservoir for Vibrio spp. in the southern Baltic Sea coast. Journal of Marine Systems, 221, 103574.
Rojas-Jimenez, K., Rieck, A., Wurzbacher, C., Jürgens, K., Labrenz, M., & Grossart, H. P. (2019). A salinity threshold separating fungal communities in the Baltic Sea. Frontiers in microbiology, 10, 680.

21) p. 5, ll. 106 Cyanobaceria is a wide field, namely in the BS. The ecology can be very different. Some are palatable (unicells also called picocyanobacteria seem to be palatabel for mesozooplankton), some are not (the large, filamentouse ones are hardly grazed directly, right?). So pls refine what you mean by cyanobacteria, and why they may (or may not) be problematic.

22) p. 5, ll. 109 Be more specific: not few but last 30? years?

23) p. 5, l. 110 Although with a clear gab between spring and summer blooms, right?

24) p. 5,, ll. 112-116 Is that true for all basins of the BS?

25) p. 5, l. 115 add summarized by:
Spilling, K., Olli, K., Lehtoranta, J., Kremp, A., Tedesco, L., Tamelander, T., ... & Tamminen, T. (2018). Shifting diatom—dinoflagellate dominance during spring bloom in the Baltic Sea and its potential effects on biogeochemical cycling. Frontiers in Marine Science, 5, 327.

Consider adding: Paul, A. J., Sommer, U., Paul, C., & Riebesell, U. (2018). Baltic Sea diazotrophic cyanobacterium is negatively affected by acidification and warming. Marine Ecology Progress Series, 598, 49-60.

26) p. 5, l. 117 Careful: There are hardly Cyanobacteria in the western BS. Pls add the basins that you refer to as this is unclear from the titel of your review, which includes the whole BS, not just the Baltic Proper and adjacent gulfs.

27) p. 5, l. 118 add: "comprising of diatoms" after "spring bloom"

28) p. 5, l. 118 add: "comprising of mainly unpalatable cyanobacteria" after "August"

29) p. 5, ll. 117-121 Here the text is unclear where you refer to the spring and summer blooms, respectively. The top down pressure probably refers to the spring bloom, as cyanobacteria are hardly grazed directly, right? This does not become clear in the text at the moment. Pls rephrase for clarity, e.g., by adding "on the diatom blooms in spring"

30) p. 5, ll. 120-121 You leave out the most interesting info here: change from which phytoplankton group to which other phytoplankton group?

31) p. 5, ll. 122-125 Consider moving into section 3.1. Climate change and primary production in the pelagial

32) p. 5, ll. 122-123 Again: be more specific, which changes in pelagic PP do you mean?

33) Nummer: 20 l. 124 Be more specific: Which ecosystem wide consequences do you mean?

34) p. 5, l. 125
What exactly do you mean with "food web dynamics"?

35) p. 5, l. 126 And with "climate" you mean what again?

36) p. 5, l. 125  add BS literature:
Kiljunen, M., Peltonen, H., Lehtiniemi, M., Uusitalo, L., Sinisalo, T., Norkko, J., ... & Karjalainen, J. (2020). Benthic-pelagic coupling and trophic relationships in northern Baltic Sea food webs. Limnology and Oceanography, 65(8), 1706-1722

37) p. 5, l. 124 Again you talk about changes without specifying which phytoplankton groups were replaced by which other groups. Pls specify.

38) p. 5, l. 127 switch community and biomass as so far you have mainly talked about community

39) p. 5, l. 128 Give an example for dominant species, pls.

40) p. 5, l. 128 Dominating in density (aka abundance) or biomass?

41) p. 5, l. 129 Add something like „...leading to a switch from group x to group y." after effect. Pls be more specific.

42) p. 6, l. 131 Pls define BSI

43) p. 6, l. 132 Pls, explain which change you mean, e.g. be more specific. Otherwise the reader has no clue of what quality the changes are that you review about.

44) p. 6, l. 132 Density or biomass (e.g. cell-carbon) wise community changes?

45) p. 6, l. 138 Do you mean phytoplankton or cyanobateria or both?

46) p. 6, ll. 140-141 How does that mechanism work? Pls explain in more detail. Really, evidence? Or rather indication? What kind of evidence do you refer to? Also, do you mean predicted or projected climate change?

47) p. 6, ll. 142-145 Consider moving into section 3.1. Climate change and primary production in the pelagial

48) p. 6, ll. 143-145 Pls revise sentence for correct grammar.

49) p. 6, ll. 143-145 Add detailed info, e.g.: "..increases by xy% compared to the control of ambient conditions."

50) p. 6, l. 143 Add „the" before „water"

51) p. 6, l. 148 Wording: „decrease in" or "release in grazing pressure from", your choice.

52) p. 6, ll. 150-154 Long sentence. Pls break up in shorter ones for better readability.

53) p. 6, l. 150 Have you defined at the beginning of your review, what you mean anytime you say CC? If not, pls add. If CC includes different variables in different parts of the text, I think you need to specify, which variables you refer to in each case.

54) p. 6, l. 151 add
Jerney, J., S. Suikkanen, E. Lindehoff and A. Kremp (2019). Future temperature and salinity do not exert selection pressure on cyst germination of a toxic phytoplankton species. Ecol. Evol. 9: 4443-4451, doi: 10.1002/ece3.5009

55) p. 6, ll. 150-154 convoluted sentence, pls revise for clarity.

56) p. 6, ll. 154-155 I don´t thing cyanobacteria per se are a problem. Unicells should not be a problem, right? Yet they ae cyanobacteria. The problem are large, unpalatable Cyanobacteria, which should be specified here and elsewhere.

57) p. 6, l. 155 Why another? Which is the first competitive advantage to begin with that you seem to refer to?

58) p. 6, ll. 154-155
This is contradictory to what is stated in ll. 603-605.

59) p. 6, l. 157 Consider the field study: Eglite, E., Wodarg, D., Dutz, J., Wasmund, N., Nausch, G., Liskow, I., et al. (2018). Strategies of amino acid supply in mesozooplankton during cyanobacteria blooms: A stable nitrogen isotope approach. Ecosphere, 9, e02135. https://doi.org/10.1002/ecs2.2135

60) p. 6, ll. 159-161 Like whom? Pls add species.

61) p. 6, l. 160 add "copepod" after small-sized

62) p. 6, l. 160: delete the comma after „small sized"

63) p. 6, l. 161 Give examples of marine taxa.

64) p. 6, l. 162 Give examples for brackish-water taxa.

65) p. 6, l. 164 add the following study to explain at least one underlying mechanism:
Dutz, J., & Christensen, A. M. (2018). Broad plasticity in the salinity tolerance of a marine copepod species, Acartia longiremis, in the Baltic
Sea. Journal of Plankton Research, 40(3), 342–355. https://doi.org/10.1093/plankt/fby013

66) p. 7, l. 167: Small scale impacts like what? Pls be more specific.

67) p. 7, l. 174: In which way are cladocerans and rotifers different functional groups e.g. after the definition of Tilman (2001)? They are different taxonomic groups but as stated it is not clear why they would represent a functional group. E.g. which specific function do they represent? Also: "Shift" from whom? Herbivory, omnivory or carnivory are functional groups. Do you mean eventually a shift from omnivorouse copepods to herbivorouse cladocerans and rotifers? I´m confused, pls clarify. Again, a definition how you define "functional group", "functional diversity" etc is urgently needed in this review.
Again, I suggest definitions of terms as given in the glossary of "Tilman, D. (2001). Functional diversity. Encyclopedia of biodiversity, 3(1), 109-120."

68) p. 11, l. 337
What about food quality and transfer efficiency of mass and energy?

69) p. 11, l. 338: After „fish" add:

Limburg, K. E., & Casini, M. (2019). Otolith chemistry indicates recent worsened Baltic cod condition is linked to hypoxia exposure. Biology letters, 15(12), 20190352.
Möllmann C, Cormon X, Funk S, Otto SA, Schmidt J, Schwermer H, Sguotti C, Voss R, Quaas M (2021): Tipping point realized in cod fishery, Nature Scientific Reports, DOI: https://doi.org/10.1038/s41598-021-93843-z

70) p. 13, l. 390: Consider adding the following study to determine trophic efficiency in field samples, because it includes examples also from the Oeland upwelling and larger Baltic Proper in the Fig. 6 :

Weber, S. C., Loick-Wilde, N., Montoya, J. P., Bach, M., Doan-Nhu, H., Subramaniam, A., ... & Voss, M. (2021). Environmental regulation of the nitrogen supply, mean trophic position, and trophic enrichment of mesozooplankton in the Mekong River plume and southern South China Sea. Journal of Geophysical Research: Oceans, 126(8), e2020JC017110. namely Chapter:"4.4. Ecosystem-Specific Trophic Enrichment in Mesozooplankton"

71) p. 13, l. 390: trophic efficiency: Pls define in the text or glossary

72) p. 13, l. 390 At the moment it does not become clear why some processes are reviewed and others are missing. E.g. what do we know about key processes like nitrification, denitrification, $N_2$ fixation in a future Baltic Sea?

Suggested References from whichs´results at least some clues may be deduced:

Bartl, I., Hellemann, D., Rabouille, C., Schulz, K., Tallberg, P., Hietanen, S., & Voss, M. (2019). Particulate organic matter controls benthic microbial N retention and N removal in contrasting estuaries of the Baltic Sea. Biogeosciences, 16(18), 3543-3564.

Allin, A., Schernewski, G., Friedland, R., Neumann, T., & Radtke, H. (2017). Climate change effects on denitrification and associated avoidance costs in three Baltic river basin-coastal sea systems. Journal of Coastal Conservation, 21(4), 561-569.

Asmala, E., Carstensen, J., Conley, D. J., Slomp, C. P., Stadmark, J., & Voss, M. (2017). Efficiency of the coastal filter: Nitrogen and phosphorus removal in the Baltic Sea. Limnology and Oceanography, 62(S1), S222-S238.

Hellemann, D., Tallberg, P., Bartl, I., Voss, M., & Hietanen, S. (2017). Denitrification in an oligotrophic estuary: a delayed sink for riverine nitrate. Marine Ecology Progress Series, 583, 63-80.

Olofsson, M., Klawonn, I., & Karlson, B. (2021). Nitrogen fixation estimates for the Baltic Sea indicate high rates for the previously overlooked Bothnian Sea. Ambio, 50(1), 203-214.

Loick-Wilde, N., Weber, S. C., Eglite, E., Liskow, I., Schulz-Bull, D., Wasmund, N., ... & Montoya, J. P. (2018). De novo amino acid synthesis and turnover during N2 fixation. Limnology and Oceanography, 63(3), 1076-1092.

73) p. 13, l. 412-415: This is interesting! Pls explain the mechanism behind this at least briefly.

74)

75) p. 13, l. 413: Add „switches" after „system"

76) p. 13, ll. 415-418: I don´t understand this. If the food web bases on heterotrophy rather than photoautotrophy, doesn´t that imply that less mass and energy is transfered to fish since the lower food chain is elongated (based on heterotrophs rather than autotrophs) leading to a higher trophic position in mesozooplankton (e.g. carnivorouse zoops instead of herbivorouse zoops, e.g. as documented Loick-Wilde et al. 2019)? Then e.g. eastern Baltic cod would also have a higher TP in such an, at times, heterotrophy based system, compared to a lower TP in western Baltic cod due to a a phototrophy based system, right? Pls clarify.

77) p. 15, ll. 461-467: I disagree with "warming induces a switch from a bottom-up controlled to a mainly top-down controlled system, which may result in increased zooplankton abundance and reduced phytoplankton biomass under warm temperature". What consequences do you think it has for higher trophic levels like fish or sea birds if mesozooplankton switches from herbivory to carnivory due to increasing densities of unpalatable cyanobacteria in a future Baltic Sea? According to a simple biogeochemical model (see Figure below from T&T 2011), the decrease in mass and energy that is available for TPs above the mesozooplankton compartment should be massiv, shouldn´t it? Pls discuss in a larger context.

[Figure]

(Thurman & Trujillo, Fig. 14.22)

Figure 14.22 shows the passage of energy between trophic levels through an entire ecosystem, from the solar energy assimilated by the autotrophic plankton through all trophic levels to

piscivorous humans. From the assimilated chemical energy, a large fraction is converted by respiration into kinetic energy for sustaining life or is lost as heat, and what remains remains is available for growth and reproduction. Thus, only about 10% of the ingested by herbivores is available for the next trophic level. Since energy is lost at each trophic level, it takes thousands of smaller marine marine organisms to produce a single fish that can be so easily consumed during a meal! Source: Trujillo, A. P., & Thurman, H. V. (2011). Essentials of oceanography (10th edition). Pearson Education. pp. 551.

78) p. 15, ll. 461-467: Add the following field based study about the environmental regulations of a switch from herbivory to carnivory in mesozooplankton in the Baltic Sea in summer in this paragraph:
Loick-Wilde, N., Fernandez-Urruzola, I., Eglite, E., Liskow, I., Nausch, M., Schulz-Bull, D., et al. (2019). Stratification, nitrogen fixation, and cyanobacterial bloom stage regulate the planktonic food web structure. Global Change Biology, 25(3), 794–810. https://doi.org/10.1111/gcb.14546

79) p. 17, l. 535
There is also a significant knowledge gap about the chances of new methodological approaches. So how about including methodological improvements that allow for a significant reduction in trait complexity while considering intraspecific variations in biological samples and specifically allow for the calibration and validation of current biogeochemical models?

Using compound-specific isotope analyses (CSIA) of amino acid nitrogen, it is now possible to measure a continuous trophic position in any biological compartment (as opposed to discrete trophic levels) based on a single field sample, which integrates the assimilation of mass from all the trophic pathways leading to a top predator from different field locations. With this information, we can take the next step of relating the effective TPs e.g. of zooplankton to the environmental conditions measured in situ (Loick-Wilde et al., 2019), providing much needed insights into the mechanisms driving shifts in TP.

The strength of CSIA lies in providing information on both TP and N sources from a single organism/sample, which is achieved with a simple comparison of the $\delta^{15}N$ values of glutamic acid (Glu) and phenylalanine (Phe) amino acids (McClelland & Montoya, 2002; Mompean et al., 2016). While Glu is enriched in $^{15}N$ by ∼8.0‰ per trophic transfer (Chikaraishi et al., 2009), the $\delta^{15}N$ of Phe remains nearly unchanged when the amino acid (AA) is transferred through the food web and thus reflects the isotopic composition of the primary producers (N-source measure, Chikaraishi et al., 2010). This approach largely eliminates potential sources of error in TP estimates associated with temporal and physiological decoupling between a consumer and its diet, and has been refined and confirmed in numerous field- and lab-based trophic studies over the last decade (reviewed by Glibert et al., 2019 and Ohkouchi et al., 2017).

The CSIA based N source identification and mean trophic position from cross-compartment analyses can directly be used to calibrate and validate current biogeochemical models and

allow for an end-to-end quantification e.g. of N inputs from $N_2$ fixation into apex predators like cod or sea birds.

80) p. 17, l. 535: Why do you refer only to salinity and stratification, what about the other abiotic variables like temperature, oxygen, OA, nitrate or phosphate inputs (either through rivers or from the anoxic sediments)?

81) p. 19, l. 604: The following review is less certain about a future decrease in cyanobacteria, pls discuss more controversial: Munkes, B., Löptien, U., & Dietze, H. (2021). Cyanobacteria blooms in the Baltic Sea: a review of models and facts. Biogeosciences, 18(7), 2347-2378.

What about the internal P storage in the sediments that gets released under anoxic conditions? Stigebrandt, A., Rahm, L., Viktorsson, L., Ödalen, M., Hall, P. O., & Liljebladh, B. (2014). A new phosphorus paradigm for the Baltic proper. Ambio, 43(5), 634-643.

82) p. 19, l. 608: What about empirical field observations/ research? Pls add.

83) p. 20, l. 623: add any missing field and lab studies as pointed out in the text here, too, when applicable.

**Technical corrections**
Just few, part of the specific comments!

---

## Author Comment (AC1)

Manuscript "Global climate change and the Baltic Sea ecosystem: direct and indirect effects on species, communities and ecosystem functioning"
by Markku Viitasalo and Erik Bonsdorff, submitted to Earth System Dynamics.

Author comments for reviewer no. 1 (esd-2021-73-RC1.pdf).

All author replies to reviewer comments in red font.

**Anonymous reviewer #1**

This review by Viitasalo and Bonsdorff offers a timely overview of the scientific evidence for effects of climate change on the ecosystems of the Baltic Sea. Both authors are in a good position to deliver a comprehensive review of this complex field, and Erik Bonsdorff is probably the researcher with the broadest knowledge on Baltic ecosystems. The review covers most ecosystem components except for marine birds and mammals. With 177 included references the coverage is impressive.

The authors give a balanced account of the wide range of studies applying many different approaches including observational, experimental and modelling studies. The emerging picture is complex where ecosystem components show different sensitivity to ongoing and anticipated climate-change effects. This review also highlights the great challenges involved in the interpretation of effects on the ecosystem from studies on the species level considering feedbacks and indirect effects through biological interactions like trophic links, cascading effect and the potential capacity for plastic acclimation and adaptation. The review honestly points out the great difficulty in predicting major changes in ecosystem functions with possible regime shifts. This is also viewed in the perspective of the uncertainties involved in climate model projections of how the climate may change across this century.

This review should be a very useful introduction to the field of climate-change effects on the ecosystem to many researchers as well as in higher education.

The review is well-written and concise. It may be argued that the review is a bit short considering the scope, but this may be an advantage if the aim is to offer a brief summary of the current knowledge together with an extensive collection of relevant literature. I only have a few comments below.

We are thankful for the constructive comments.

**Major Comment**

The only major comment is that the section on "Knowledge gaps" could be more extensive. There are questions how to best approach climate-change effects through experimental studies. There may be a lack of experimental infrastructure of sufficient scale in terms of the ability to control multiple environmental factors, sufficient replication, and not least technical staff to maintain also long-term studies. There is the question of more extensive habitat-mapping and also the development of more advanced Species Distribution Models (partly mentioned), e.g. the inclusion of biological interactions, plasticity and capacity for adaptation. A major knowledge gap (likely deserving a different paper) is also how to interpret the present, rather sprawling, knowledge about climate-change effects into Marine Spatial Planning and conservation efforts, e.g. the design of Marine Protected Areas.

Very valid points, thank you. We have now expanded the chapter Knowledge gaps significantly and also point out methodological challenges.

**Minor Comments**

Page 4, line 63. Climate change may also affect the opportunities for freshwater biota, e.g. vegetation.

Modified as suggested. Edited text: "…potentially affecting the marine and freshwater biota inhabiting the Baltic Sea, as well as the human society"

Page 5, line 112. More sunlight because of less cloudy conditions?

Yes, it is probable that the proximate reason for more sunlight is decreased cloudiness in spring. Hjerne et al. (2019) states: "The less windy and *less cloudy springs* observed after 1990 are linked to a negative trend in the North Atlantic Oscillation Index (p = 0.01, Mann–Kendall) and is most likely only a temporary shift in the long-term climate development." We edit the text and add a notion on the potential attribution to NAO and global climate change, and the uncertainties associated with projections.
New text: "Some studies have attributed these shifts to changes in environmental conditions associated with global change (Groetsch et al., 2016), while others have indicated a connection with the North Atlantic Oscillation (NAO); a decline in the intensity of NAO in the 1990s caused less cloudy conditions (more irradiance), and less windy conditions induced stronger stratification of surface water (Hjerne et al., 2019). Such shifts, if caused by variations in NAO, may be temporary and reversible, whereas shifts caused by global climate change may be more enduring. It has been suggested that, in the future climate, higher temperatures and less ice will cause an earlier bloom of both diatoms and dinoflagellates, with increased dinoflagellate dominance, but this development may be counteracted by increasing windiness and cloudiness, which have been projected by earlier modelling studies (Hjerne et al., 2019). However, more recent models have suggested that, while the winter conditions will most probably become more cloudy and windy, the projections for future conditions in spring and summer are more uncertain (Christensen et al., 2021). Therefore, while it is obvious that the projected warming will induce shifts in the structure of spring phytoplankton communities in the next 60 to 80 years, the exact nature of the changes cannot be projected with certainty."

Page 5, line 114. Mesodinium should be considered mixotrophic (e.g. Stoecker DK, Hansen PJ, Caron DA, Mitra A. 2017. Mixotrophy in the Marine Plankton. Annual Review of Marine Science 9: 311-335).

Modified as suggested. Edited text: "…mixotrophic ciliate *Mesodinium rubrum*…"

Page 5, line 119. A change in N/P ratio? with more P favouring the N-fixing Cyanobacteria?

A good point. We include a notion on the N/P ratio and add a more thorough analysis of the development in the Bothnian Sea, where the N/P ratio has also declined.
New text: "Also, in the Gulf of Bothnia, changes in phytoplankton and cyanobacteria communities have been observed in a study covering period 1979 to 2017. In both Bothnian Sea and Bothnian Bay diatoms have decreased while chrysophytes, prasinophytes, and prymnesiophytes have increased. In the Bothnian sea, also concentration of chlorophyll *a* has increased in summer, along with the increase of the autotrophic ciliate *Mesodinium rubrum* and cyanobacteria blooms. have increased (Kuosa et al., 2017). The observed increase of Cyanobacteria blooms in the Bothnian Sea have been attributed to an increased freshwater flow and, since 2000, to an increased intrusion of more saline Baltic Proper water into the Bothnian Sea. These changes have increased stratification, lowered oxygen conditions and a led to decline in N:P ratio of the Bothnian Sea, which have favoured the development of Cyanobacteria blooms in the area (Rolff and Elfwing, 2015; Ahlgren et al., 2017; Kuosa et al., 2017)."

Page 6, line 140. So what is causing that nutrient reduction? This sentence links poorly to the previous sentence about increased nutrient loading.

Here, "nutrient reduction" refers to measures to reduce anthropogenic nutrient loading. The text is now clarified.
New/edited text: "On the other hand, it has been projected that reduction of anthropogenic nutrient loading according to HELCOM Baltic Sea Action Plan will, in the long run, counteract the increased nutrient loading caused by climate change and lead to decreased eutrophication (Ehrnsten et al., 2019b;

Pihlainen et al., 2020). This has also been seen in Kattegatt, where reduction of nutrient loading led in mid 1990s to a shift from a highly eutrophic state characterized by small phytoplankton species and low water transparency to increasing share of diatoms, decreasing overall phytoplankton biomass and increase of water transparency (Lindegren et al., 2012)."

Page 6, line 155. What competitive advantage? Higher levels of toxins that defend against predation?

Yes, we refer to toxic effects, and this is now clarified and backed up by references.
New/edited text: "As toxins of both dinoflagellates (Sopanen et al., 2011) and cyanobacteria (Karjalainen et al., 2006; Karjalainen et al., 2007; Engström-Öst et al., 2017) can accumulate in Baltic Sea zooplankton and induce lower grazing rates and higher mortality, these studies suggest that toxic dinoflagellates and filamentous unpalatable cyanobacteria may get a competitive advantage against diatoms in a future Baltic Sea."

Page 7, line 173. There is an experimental study "Karlsson K, Winder M. 2020. Adaptation potential of the copepod Eurytemora affinis to a future warmer Baltic Sea, Ecology and Evolution 10: 5135-5151". This Experimental study suggests that copepod populations from warmer environments can at present adapt to a future warmer Baltic Sea, whereas populations from colder areas show reduced adaptation potential to high temperatures.

A good suggestion. Text amended and reference added.
New text: "On the other hand, some capability to temperature adaptation has been demonstrated experimentally for the Baltic Sea copepod *Eurytemora affinis* (Karlsson and Winder, 2020). Interestingly, the adaptability was better in populations reared in warm temperatures ($\geq 17°C$), which suggests that (*i*) southern populations can better cope with increasing temperatures than the northern ones, and (*ii*) the adaptation capability of all (surviving) populations may improve with proceeding climate change."

Page 8, line 216. There is a recent study by "Kinnby A, Jonsson PR, Ortega-Martinez O, Töpel M, Pavia H, Pereyra RT, Johannesson K. 2020. Combining an ecological experiment and a genome scan show Idiosyncratic responses to salinity stress in local populations of a seaweed. Frontiers in Marine Science. 7: 470". This study shows the possible presence of locally adapted populations of Fucus vesiculosus in the Baltic with different tolerance to salinity and with different genetic backgrounds.

A good suggestion. Text amended and reference added.
New text: "Especially if populations are genetically separated, they may have very different adaptation capabilities. In a study performed in the Danish Straits, certain populations of *Fucus vesiculosus* were only slightly affected by a salinity decline, while others displayed clear responses, and one population showed severe stress symptoms and stopped growing (Kinnby et al., 2020)."

Page 8, line 230. A typo: "algae" should read "alga".

Corrected.

Page 9, line 239. It may be pointed out that Zostera in the Baltic proper may consist of some few clones making the total genetic diversity low with less capacity for adaptation to a changing environment, although it has been found that somatic mutations may increase overall diversity (Yu et al. 2020. Nature Ecology & Evolution 4: 952).

Good point. Text amended with a notion to low genetic differentiation in Baltic *Zostera*.
Edited text: "The rapidly changing marine environment in the Baltic Sea however pose an evolutionary risk, especially for populations with specific adaptations, such as relicts, which may be at risk for local extinctions (Johannesson et al., 2011), and for populations that live close to the limits of their geographical ranges and may have low genetic differentiation, such as eelgrass *Zostera marina* (Billingham et al., 2003). Indeed, mesocosm studies have indicated that Baltic Sea eelgrass populations suffer from heatwaves in summer (Ehlers et al., 2008) and elevated temperatures in winter-spring period (Sawall et al., 2021)."

Page 9, line 241. A particular concern is the potential loss of marine, canopy-forming macroalgae (Fucus, Furcellaria). There is here no freshwater vascular plants that can replace that type of vegetation on hard substrata.

Important point. New text (in Conclusions): "Of particular concern is the potential loss from rocky substrates of marine canopy-forming brown and red macroalgae. Freshwater vascular plants only grow on soft substrates and therefore they cannot replace the ecological functions of marine macroalgae, even though they would be favoured by freshening of the surface waters."

Page 9, line 273. In the paper "Meier et al. 2020. Future projections of record-breaking sea surface temperature and cyanobacteria bloom events in the Baltic Sea. Ambio 48: 1362-1376", they showed how the frequency of heatwaves may dramatically increase under some climate change scenarios.

Yes, we agree. (The paper is from 2019, though.)
Edited text: "Therefore, heat waves, which have been projected to increase in frequency (Meier et al., 2019), may pose a severe threat to sublittoral invertebrates."

Page 12, line 369. "Ipcc" should read "IPCC".

Corrected.

Page 12, line 375. The previous sentence states a projected increase in stratification, while this sentence refers to enhanced mixing. I guess that this enhanced mixing is caused by more intense wind speeds during the spring when the thermocline is weak. Please, rephrase to avoid confusion here.

The paragraph is now clarified.
Edited text: "Several studies using coupled oceanographic-biogeochemical ecosystem models have projected more phytoplankton and especially cyanobacteria in the warmer future Baltic Sea. In the central Baltic Sea, increased water temperature causes, together with increased irradiation and enhanced wind-induced mixing of the surface-layer, an earlier but less intense spring bloom, while in summer, enhanced thermal stratification favours more intense cyanobacteria blooms (Meier et al., 2011a; Andersson et al., 2015; Neumann et al., 2012; Chust et al., 2014). Intensified blooms of cyanobacteria are expected especially if hypoxia will prevail and internal loading will increase supply of phosphorus from anoxic sediments into the surface layer, decreasing the N:P ratio (Meier et al., 2011b; Funkey et al., 2014). However, the magnitude of changes and their consequences for biogeochemical processes, e.g. for nitrogen fixation, differ greatly between models (Munkes et al., 2021)."

Page 12, line 378. Do you know what is the projected P/N ratio for the external loading?

Good point. We do not find direct references, but from Huttunen et al. 2015, their Figs 6 and 7, something can be deduced.
New text: "Phytoplankton communities will also be affected by anthropogenic nutrient loading. It has been projected that, due to climate change, the total phosphorus loading into the Finnish sea area will increase relatively more than that of nitrogen (Huttunen et al., 2015), suggesting a decreased N:P ratio of the external loading. The future nutrient ratios of the external loading are, however, hard to be predicted, as they are affected by several factors, including agricultural adaptation, and biogeochemical processes in the soil, lakes and rivers."

Page 13, line 386. Species names should appear in italic.

Corrected.

Page 13, line 386. Note that also Cyanothece (supposed to increase) is a nitrogen fixer.

Good point. New text: "This could however be balanced by potential increase of *Cyanothece*, which is also a nitrogen fixer."

Page 13, line 401. What is the mechanism behind this negative effect on flux? Stratification?

Yes, but also trophic effects. Edited text: "The total phytoplankton biomass however decreased because increased stratification decreased nutrient flux to the surface layer (Lewandowska et al., 2014; Lewandowska et al., 2012). Furthermore, in stratified conditions the relative importance of the microbial loop increased because copepods switched to feed more on ciliates instead of phytoplankton, which probably releases heterotrophic nanoflagellate grazing on bacteria, which may reduce remineralization and decrease availability of nutrients for phytoplankton (Lewandowska et al., 2014)."

Page 13, line 410. What may be the consequence of this shift apart from lower food web efficiency? Lower export to benthic biota?

Yes. Edited text: "This shifts the carbon flow towards microbial heterotrophy (Wikner and Andersson, 2012), which may also decrease vertical flux of organic matter to zoobenthos (Ehrnsten et al., 2020)."

The whole chapter is now thoroughly rewritten to encompass all different aspects of this phenomenon.

Page 14, line 449. A detail: Myrionecta is regarded as a junior synonym to Mesodinium (and not the other way around). Also, Mesodinium rubrum is now considered a complex of several species.

Yes, we now use the form *Mesodinium* sp. (although Lischka et al. 2017 use *M. rubra*).
Edited text: "…and the abundance of the mixotrophic ciliate *Mesodinium* sp. even increased in mesocosms with OA,…"

Page 14, line 450. Note that Dinophyta (e.g. Dinophysis acuminata) is a PREDATOR on M. rubrum.

We agree and omit "Dinophyta". Edited text: "…the abundance of the mixotrophic ciliate *Mesodinium* sp. even increased in mesocosms with OA, because of increase of its food, e.g. picoeukaryotes, at higher $CO_2$ levels…"

Page 17, line 542. The word "through" should be omitted.

Corrected.

Page 17, line 558. A bottle-neck for high-resolution 3D circulation models is the availability of high-resolution pan-Baltic bathymetries, and forcing data (e.g. wind fields).
For species distribution models (SDM) a major constraint is the poor habitat mapping in many areas (with exceptions in Finland and Estonia). There is also a need for the inclusion of biological interactions (e.g. predator-prey) into SDMs.

Good suggestions. New text: "A bottleneck for high-resolution 3D circulation models is however the availability of high-resolution pan-Baltic bathymetries and forcing data (e.g. wind fields). For pan-Baltic species distribution models (SDMs), in turn, a major constraint is – in many areas – the poor availability of detailed species and habitat mapping data."
New text: "Statistical models, both 3D ecosystem models and 2D SDMs, rarely include biological interactions into the models, nor can they fully incorporate the more complex effects of multi-species predatory or intraguild relationships. Inclusion of such complex food web effects would require merging of 3D and 2D models with multi-species food web models that operate on the level populations rather than carbon flows (ecosystem models) or species coverages (SDMs)."

Page 18, line 574. The word "While" can be omitted.

Omitted.

Page 18, line 581. Better start this sentence with "However, some common…".

Edited as suggested.

**AC1 references**

[revised manuscript text omitted]

---

## Author Comment (AC3)

Manuscript "Global climate change and the Baltic Sea ecosystem: direct and indirect effects on species, communities and ecosystem functioning"
by Markku Viitasalo and Erik Bonsdorff, submitted to Earth System Dynamics.

Author comments for reviewer no. 3 (esd-2021-73-RC3-supplement.pdf).

All author replies to reviewer comments in red font.

**Anonymous reviewer #3**

**ESD specific reviewer statements**

1. **Does the paper address relevant scientific questions within the scope of ESD?**

   Yes.

2. **Does the paper present novel concepts, ideas, tools, or data?**

   Not yet so much. There are new tools (or methods) to better bridge gaps in cross-compartment community structure analyses (e.g. by fuzzy coding or metabarcoding). But there are also new methods (namely compound specific isotope analyses of amino acid nitrogen, CSIA) that bridge gaps in cross-compartment functional diversity analyses more directly and which, different to other "multi trait" approaches, include significant reduction of complexity by addressing functional groups (according to Tilman 2001: phototrophs, mixotrophs, heterotrophs, herbivores, different levels of carnivores,) rather than individual traits or taxonomic groups. CSIA allows for the direct measurement of the mean trophic position of a field sample as one key cross-compartment functional trait (according to Tilman 2001) and of the dominant inorganic nitrogen source used for growth from a field sample as second key cross-compartment functional trait (Tilman 2001) for end-to-end analyses (e.g., physics to fish to human sectors, in sensu Peck et al., 2018) of food webs. Both traits can directly be determined from a single field sample independent from the compartment it comes from (e.g. mixed phytoplankton, mixed mesozooplankton, bivalves, herring, cod, seagulls) and can directly be used as "common currency" across all compartments (e.g., physics to fish to human sectors, in sensu Peck et al., 2018) to calibrate and validate current biogeochemical models. For example, no biogeochemical model yet accounts for the mass and energy loss for fish that must be related to the shift in mean trophic position of mesozooplankton from herbivory (TP of 2) to carnivory (TP of 3) during cyanobacterial blooms in the central Baltic Sea (see specific comment #) and which can help to explain the loss in apex predators in the Baltic Sea.

   Thank You for the critical comments, we have now revised the ms. thoroughly and used a more critical approach. For instance, we have included more scrutiny on the reviewed papers and their methods, included assessment of consensus and dissensus in Conclusions, and also added new text on the potential of novel methods in Knowledge gaps (see response to the Specific comment no. 79).

3. **Are substantial conclusions reached?**

   Yes, but they are sometimes inconsistent with parts in the earlier text. For example, I still don´t know if the research up to now points to N2-fixing, unpalatable cyanobacteria as "winner" of cc or not. So conclusions should be somewhat refined.

   We agree that projecting structure and functioning of the pelagic ecosystem with 3D modelling in the distant future (e.g., for the year 2100) is challenging and involves many assumptions. In our review we note that several 3D modelling studies project (in a BAU scenario) an increase of Cyanobacteria in a warmer and more stratified Baltic Sea, but in the Knowledge gaps and

Conclusions try to highlight the associated uncertainties by noting the dissensus between modelling studies and certain short-term monoculture and mesocosm studies. We emphasize that also the modelling studies propose that successful reduction of nutrient loading from land (according to HELCOM BSAP) would outweigh the effects of proceeding climate change and decrease the cyanobacteria biomass, or at least diminish the occurrence of very large blooms.

**4. Are the scientific methods and assumptions valid and clearly outlined?**

Generally yes. Yet, I feel like important empirical field based studies are still missing in this review as marked in the specific comments.

Thank you for the comment. We have added the suggested references, as pointed out below.

**5. Are the results sufficient to support the interpretations and conclusions?**

See point 3.

Response in point 3.

**6. Is the description of experiments and calculations sufficiently complete and precise to allow their reproduction by fellow scientists (traceability of results)?**

Pls explain how you found and chose your reviewed literature (e.g. did you use google scholar, did you visit the webites of major Baltic Sea research institutes for most recent publications etc.?).

A new Chapter 2, Definitions and review methods, is now included in the ms. The main method was Web of Science and we only included papers published in 2010-2021 (the period since BACC II; see Chapter 22 for a more detailed explanation of the review method).

We did not specifically look for institute web pages. We want to emphasize that this review is not a full systematic review of all research done on climate change effects on the Baltic Sea . Rather, we highlight the variety of field, experimental and modelling studies on this subject and summarise what can be concluded from the evidence published since 2010.

**7. Do the authors give proper credit to related work and clearly indicate their own new/original contribution?**

Not applicable for an invited review article, I think.

Ok.

**8. Does the title clearly reflect the contents of the paper?**

Authors should definitely add a definition of ecosystem functioning into your introduction or as a glossary, e.g. Ecosystem Functioning (Tilman 2001): The rate, level, or temporal dynamics of one or more ecosystem processes like primary production or nutrient gain or loss.

A new Chapter 2, Definitions and review methods is now included and the definitions are given there. We apply Tilman's (2001) definition of ecosystem functioning.

**9. Does the abstract provide a concise and complete summary?**

I will suggest a major revision and there might be changes in the abstract necessary due to that. For example, what about the internal P storage in the sediments that gets released under anoxic conditions? How will that influence the projection of future cyanobacterial blooms in the Baltic Sea?

Abstract is now revised and adjusted according to the suggestions provided. A notion of the importance of internal loading of P is also there.

**10. Is the overall presentation well structured and clear?**

The authors could more systematically show, which direct and indirect effects have been addressed by which kind of research. Generally, I feel that new knowledge based on empirical research did not always find its way into here, if the "titel" did not include certain key words. See also reply to 6.

We have now assessed the potential direct and indirect climate related effects on species and ecosystem functioning, and also made an attempt to the attribution to either global (anthropogenic) climate change and more natural variations in climate related parameters. We have added missing papers as suggested.

**11. Is the language fluent and precise?**

There are some minor spelling and grammar errors that need to be corrected for.

Language errors corrected as suggested.

**12. Are mathematical formulae, symbols, abbreviations, and units correctly defined and used?**

Not applicable.

Ok.

**13. Should any parts of the paper (text, formulae, figures, tables) be clarified, reduced, combined, or eliminated?**

The structure could be sharpened a little bit maybe. For example, I find it more intuitive to start with phototrophs like phytoplankton and cyanobacteria rather than with "Microbial communities". Also, cyanobacteria is a very broad group. Maybe palatable cyanobacteria should be differentiated from unpalatable ones, as well as those that do fix nitrogen from those that don´t fix nitrogen but only profit from leaking out of diazotroph nitrogen from the N2-fixing ones. More in the revised pdf attached.

Thank you for the comment, we have now restructured the text. The order is now: (1) Phytoplankton (2) Cyanobacteria (now separated from "Phytoplankton and Cyanobacteria"), (3) Zooplankton, etc. The chapter Bacteria (instead of "Microbial communities") is now presented last, after Fish, before the chapter Climate change and ecosystem structure and function.
    We decided to move relevant parts of the chapter "Climate change and primary production" to the chapters "Phytoplankton" and "Cyanobacteria", as there was much overlap between these three chapters.
    We have also clarified the distinctions between palatable and less palatable cyanobacteria, and explained the $N_2$-dynamics as presented in the responses to detailed comments, below.

**14. Are the number and quality of references appropriate?**

There are important papers missing at the moment, I think. Pls see the revised pdf for details.

Suggested papers, and some others, added.

**15. Is the amount and quality of supplementary material appropriate?**

Not applicable.

Ok.

**General comment**

I enjoyed reading and reviewing this manuscript and I think it's worth publishing. That's said, I think that there are still many improvements possible to make it even better. For example, the authors did not find yet a good balance between a "too detailed" revision of a study and a "too brief" review of a study. In many places, I have a quarrel with the statements of many studies, which are often generalized beyond recognition (anytime I request: "Pls be more specific"). Maybe the authors can add some details here and there, namely where mechanisms are mentioned but which are hardly explained in sufficient detail (pointed out below in the specific comments).

The scientific community currently is struggling to find "the right" definition of functional diversity and other terms and how to investigate functional diversity. I think it's worth crediting the different approaches and the dissent. Some methods allow for an indirect approach to study functional diversity (those including taxa), others a direct approach (those including only functional groups). A review is a chance to point out methodological improvements over the last years and the review would gain relevance if some crosscompartment approaches are included here (e.g. specific comment #79).

> Thank you for the critical comment. We have now added much more explanation in parts where the low specificity was pointed out. We have, e.g., explained biogeochemical mechanisms behind the presented hypotheses and added some methodological details, such as duration of experiments and size of micro- and mesocosms, to highlight the time and space scales. Suggested papers, and some others, are added.
>
> We have written a new chapter 2, Definitions and review methods, where the key definitions used are explained.
>
> We have included, in the chapter Knowledge gaps, a commentary on the potential usage of novel methods in climate change research in the Baltic Sea.

**Specific comments**

1) p 1, ll. 3: add a definition of ecosystem functioning into your introduction or as a glossary, e.g. Ecosystem Functioning (Tilman 2001): The rate, level, or temporal dynamics of one or more ecosystem processes like primary production or nutrient gain or loss

   > A new full chapter 2, Definitions and review methods, has been written after the Introduction. The definitions are explained there.

2) p. 4, ll. 60-64: After Tilman (2001) the definition of ecosystem functioning (EF) is: The rate, level, or temporal dynamics of one or more ecosystem processes like primary production or nutrient gain or loss.

   My question: How do you define EF in this review and which processes do you include in your review and which do you not include and why (e.g. are there other reviews out to refer to like "Wannicke, N., Frey, C., Law, C. S., & Voss, M. (2018). The response of the marine nitrogen cycle to ocean acidification. Global Change Biology, 24(11), 5031-5043.")? Pls add this information into your text.

   Pls add a side note with definitions for the most prominent terms in your review, e.g.
   -climate change
   -ecosystem functioning
   - trophic dynamics etc.

   > A new full chapter 2, Definitions and review methods, has been written after the Introduction. The approach and definitions are explained there.

3)   p. 4, ll. 68: wording "...more light onto (not into) the complex..."

Corrected.

4)   p. 4, ll 67-69: I understand that there is an upcoming review on the cc projections associated with this review. Still, for the "stand alone status" of this review it would be very helpful, to specify here in more detail the projected "abiotic" changes that possibly are most important for the biology (maybe less the atmospheric forcing behind them) including namely salinity, temperature, stratification and oxygen as well as OA, nitrate and phosphate levels, the latter from both rivers and the anoxic sediments. Important to include: How certain or uncertain can we be about them (e.g. in line 537-539)?

We have revised and expanded the Introduction thoroughly.
    We have also included in Knowledge gaps more analysis of the consensus and dissensus, and uncertainties, and in Conclusions added more text on the possible attribution of the found responses to the global climate change and climatic variations in general.
    Note also that there will be a separate review on Biogeochemistry of the Baltic Sea, so we only review biogeochemical processes if they are directly linked to the species and communities, or ecosystem functioning.
    New text (in Introduction): "We primarily review studies that shed light to the effects of climate change on the Baltic Sea species, populations and communities, and the ecosystem function, by studying parameters which are assumed to change due to climate change, such as water temperature, salinity, oxygen, pH and nutrients. We also include studies that show past changes in communities, and those which study more indirect processes related to changes in biogeochemistry and food web, if these changes were attributed to parameters that have been shown to fluctuate with climatic variations. Most of these studies have in their Introduction referred to global climate change, and in their Discussion interpreted the results in the context of future climate change. Conclusions on the role of climate change on shaping the structure and functioning of the Baltic Sea ecosystem are then drawn from the existing field, experimental and modelling evidence."

5)   p. 4, ll. 70: add: „the" before year

Corrected.

6)   p. 4, ll. 71: add „field based" before "responses"

Corrected.

7)   p. 4, ll. 75: „food web dynamics", what do you mean by this? Pls specify.

Edited text: "Third, the complex effects of climate change on the interactions between trophic groups, such as phytoplankton, cyanobacteria, zooplankton and fish, as well as algae or vascular plants and invertebrates inhabiting or grazing on them are analysed based on field and experimental studies where trophic interactions have been investigated."

8)   p. 4, ll. 75: add „both directly in the field as well as in experimental studies." At the end of the sentence.

Added as suggested.

9)   p. 4, ll. 76: What kind of modeling studies, pls specify.

Edited text: "Finally, a number of modelling studies, mostly based on coupled oceanographic-biogeochemical models, sometimes also including the main biotic components of the open sea

ecosystem, i.e., phytoplankton, cyanobacteria, zooplankton and planktivorous and piscivorous fish, as well as benthos, are reviewed."

10) p. 4, ll. 85: Why don´t you start with the autotrophic communities instead of the heterotrophic microbial community in Chapter 2?
Phototrophs form the base of the food web also for the heterotrophic microbial community that you seem to mainly refer to in Chapter 2.1.
Also, you should define, which organisms you mean with "microbial community", e.g. only heterotrophs?
What about marine viruses and fungi?

A valid comment. We now start with "Phytoplankton" and continue with "Cyanobacteria" (separated from the original chapter "Phytoplankton and Cyanobacteria").
In the original chapter "Microbial communities" we referred to experimental studies investigating bacteria (not viruses or fungi), so we change the name of the Chapter to "Bacteria", and move this chapter later in the paper (after the chapter Fish).

11) p. 4, ll. 87: add "(OA)" after ocean acidification

Added as suggested.

12) p. 4, ll. 90: How did community change? Be more specific.

Edited text: "OA alone had a limited impact, but when combined with increased temperature, certain bacterial phylotypes, such as betaproterobacteria, increased. The authors suggest that synergistic effects of increased temperature and acidification may selectively promote growth of specific bacterial populations."

13) p. 4, ll 91: Which OTUs? Be more specific.

Edited text: "In the southern Baltic Sea (Kiel Bight) the impact of OA was also limited to few operational taxonomic units (OTUs), such as *Bacteroidetes* `NS3a marine group´, as the bacterial community mainly responded to temperature and phytoplankton succession."

14) p. 4, ll. 92: How did the microbial community respond? Be more specific.

New/edited text: "…the bacterial community mainly responded to temperature and phytoplankton succession. Depending on studied season and temperature treatment, Cyanobacteria, *Bacteroidetes*, Alphaproterobacteria and/or Gammaproterobacteria increased."

15) p. 5, ll. 94: Give range also for CO2 as for Sal and Temp.

Edited text: "In experiments using a natural summer microplanktonic community, where $CO_2$ was increased (from 380 to 960 µatm) and salinity decreased (from 6 to 3 psu),…"

16) p. 5, ll. 99 delete bracket before "Berner"

Unfortunately, the EndNote program forces this format. If the paper will be accepted, we will correct this at copy-editing stage.

17) p. 5, ll. 99 delete extra dot after et al.

There is no extra dot. It is a comma.

18) p. 5, ll. 100-101 Changed drastically to what? Be more specific. What does a "high temperature community" look like?

New/edited text: "At reduced salinity levels, certain Actinobacteria and Bacteroidetes OTUs increased, and the heterotrophic bacteria community resembled communities at high temperature, indicating synergistic effects of temperature and salinity."

19) p. 5, ll. 102-103 Which increase? Unclear where you refer to here.

New/edited text: "Biotic interactions were more dominant than abiotic ones, however. The largest increase in heterotrophic bacterial biomass was detected when filamentous cyanobacteria started to decay, regardless of temperature or salinity. It was suggested that this indirect coupling between heterotrophic bacteria and filamentous cyanobacteria is more important for bacterial communities than the direct effects of temperature or salinity."

20) p. 5, l. 104 (end of paragraph) What about the effect of other abiotic variables like stratification, oxygen, nitrate and phosphate on heterotrophic bacteria, viruses and fungi?
Möller, L., Kreikemeyer, B., Gerdts, G., Jost, G., & Labrenz, M. (2021). Fish as a winter reservoir for Vibrio spp. in the southern Baltic Sea coast. Journal of Marine Systems, 221, 103574.
Rojas-Jimenez, K., Rieck, A., Wurzbacher, C., Jürgens, K., Labrenz, M., & Grossart, H. P. (2019). A salinity threshold separating fungal communities in the Baltic Sea. Frontiers in microbiology, 10, 680.

New/edited text: "This highlights the importance of considering both abiotic and nutrient effects and the more indirect food web effects, i.e. predation, on microbial communities."
    The papers of Möller et al. (2021) and Rojas-Jimenez et al. (2019) are interesting but are not specifically referring to climate change, so we chose not to include them.

21) p. 5, ll. 106 Cyanobacteria is a wide field, namely in the BS. The ecology can be very different. Some are palatable (unicells also called picocyanobacteria seem to be palatable for mesozooplankton), some are not (the large, filamentous ones are hardly grazed directly, right?). So pls refine what you mean by cyanobacteria, and why they may (or may not) be problematic.

We are mainly reviewing papers that study filamentous, bloom-forming cyanobacteria, and make this clear in relevant places, especially when considering zooplankton grazing.
    To make the text better structured, we have separated text on cyanobacteria from the chapters "Phytoplankton and cyanobacteria", and "Climate change effects on primary producers", to a new chapter "Cyanobacteria".

22) p. 5, ll. 109 Be more specific: not few but last 30? years?
23) p. 5, l. 110 Although with a clear gap between spring and summer blooms, right?
24) p. 5, ll. 112-116 Is that true for all basins of the BS?

Excellent points. As an answer to comments 22-24 we have thoroughly edited the text in this point.
    New/edited text: "The growing season of phytoplankton has significantly prolonged with warming temperatures during the past few decades. A satellite-based study suggested that the period with chlorophyll-a of at least 3 mg m$^{-3}$ doubled, from 110 days in 1998 to 220 days in 2013 (Kahru et al., 2016). Another study using phytoplankton sampling data from the shallow Bay of Mecklenburg, western Baltic Sea, confirmed that the phytoplankton growing season, which in 1988-1992 on average endured from March to August, now (2014-2017) extends from February to December (Wasmund et al., 2019), albeit with a longer gap between the spring and late summer peaks. This prolongation was tentatively explained by increased sunshine in spring and higher temperature in the autumn, inducing changes in various factors, such as species composition and settling rates of phytoplankton, remineralization of organic matter by bacteria, and grazing rates by zooplankton (Wasmund et al., 2019). Although studies from all Baltic Sea basins do not exist, it is probable that similar prolongation, caused by changes in radiation and temperature, have also taken place elsewhere."

25)    p. 5, l. 115 add summarized by: Spilling, K., Olli, K., Lehtoranta, J., Kremp, A., Tedesco, L., Tamelander, T., ... & Tamminen, T. (2018). Shifting diatom—dinoflagellate dominance during spring bloom in the Baltic Sea and its potential effects on biogeochemical cycling. Frontiers in Marine Science, 5, 327.

   Reference added.

   Consider adding: Paul, A. J., Sommer, U., Paul, C., & Riebesell, U. (2018). Baltic Sea diazotrophic cyanobacterium is negatively affected by acidification and warming. Marine Ecology Progress Series, 598, 49-60.

   Reference added, and a notion made on the discrepancy between modelling and mesocosm studies regarding potential increase of Cyanobacteria.
   New text: "It is also notable that, in mesocosm studies, an increase of $pCO_2$ (from 360 to 2030 µatm) coupled with an increase in water temperature (from 16.6 to 22.4 °C) had a *negative* impact on the biomass of the diatzotrophic cyanobacteria *Nodularia spumigena* (in 1400-L mesocosms, 28 days) (Paul et al., 2018). This result contradicts the modelling studies that suggest that the increased stratification, together with potentially increasing remineralization of organic matter and release of phosphorus from the anoxic sediments, will increase cyanobacteria blooms (Meier et al., 2011a; Andersson et al., 2015; Neumann et al., 2012; Chust et al., 2014)."

26)    p. 5, l. 117 Careful: There are hardly Cyanobacteria in the western BS. Pls add the basins that you refer to as this is unclear from the title of your review, which includes the whole BS, not just the Baltic Proper and adjacent gulfs.

   A valid comment. We have added notions on the study areas in several places in the ms.
   New/edited text (in the chapter Phytoplankton): "In the northern Baltic Proper, Åland Sea and the Gulf of Finland, the biomasses of Chrysophyceae, Prymnesiophyceae and Cyanophyceae have increased (Suikkanen et al., 2013), and the phytoplankton biomass maximum, which in the 1980's was in spring and mainly consisted of diatoms, is now in July-August and is dominated by filamentous cyanobacteria."

27)    p. 5, l. 118 add: "comprising of diatoms" after "spring bloom"
28)    p. 5, l. 118 add: "comprising of mainly unpalatable cyanobacteria" after "August"

   Edited text: "and the phytoplankton biomass maximum, which in the 1980's was in spring and mainly consisted of diatoms, is now in July-August and is dominated by filamentous cyanobacteria."

29)    p. 5, ll. 117-121 Here the text is unclear where you refer to the spring and summer blooms, respectively. The top down pressure probably refers to the spring bloom, as cyanobacteria are hardly grazed directly, right? This does not become clear in the text at the moment. Pls rephrase for clarity, e.g., by adding "on the diatom blooms in spring"

   Edited text: "This shift has been explained by a complex interaction between warming, eutrophication and increased top-down pressure on species of the spring bloom, as well as changes in DIN:DIP ratio in summer (Suikkanen et al., 2013)."

30)    p. 5, ll. 120-121 You leave out the most interesting info here: change from which phytoplankton group to which other phytoplankton group?

   New text (in the chapter Cyanobacteria): "Also, in the Gulf of Bothnia, changes in phytoplankton and cyanobacteria communities have been observed. In the Bothnian sea, concentration of chlorophyll *a* has increased in summer, along with the increase of cyanobacteria and the mixotrophic ciliate *Mesodinium rubrum* (Kuosa et al., 2017)."

31) p. 5, ll. 122-125 Consider moving into section 3.1. Climate change and primary production in the pelagial

Thank you for the suggestion for restructuring the text. Sentence moved as suggested. We have however collated most text concerning cyanobacteria under a new chapter "Cyanobacteria", to avoid overlap between chapters "Phytoplankton and cyanobacteria" and "Climate change and primary production".

32) p. 5, ll. 122-123 Again: be more specific, which changes in pelagic PP do you mean?

New/edited text (in chapter "Projections of primary producers", formerly called Climate change and primary productivity"): "Experimental (mesocosm) evidence also supports findings that climate change induced warming up of water and changes in light conditions will drive changes in the pelagic primary producers, by accelerating spring bloom, inducing a decline in peak biomass and favouring small size cells, either directly or via increased grazing by copepods (Sommer et al., 2012)."

33) Nummer: 20 l. 124 Be more specific: Which ecosystem wide consequences do you mean?

New/edited text (in chapter "Projections of primary producers"): "A thorough review illustrating benthic-pelagic coupling shows ecosystem-wide consequences of altered pelagic primary production, e.g. via increasing sedimentation of organic matter and consequent remineralization, inducing hypoxic conditions both in the deep basins and in the shallower archipelago areas (Griffiths et al., 2017), probably also inducing internal loading of phosphorus from sediments (Puttonen et al., 2014; Stigebrandt et al., 2014)…"

34) p. 5, l. 125 What exactly do you mean with "food web dynamics"?

Edited text: "…and impacting abundances and interactions between main trophic levels, e.g. phytoplankton, detritus and zoobenthos as well as detritivores, benthivores phytoplanktivores, zooplanktivores and piscivores (Kortsch et al., 2021)."

35) p. 5, l. 126 And with "climate" you mean what again?

We use it in the same meaning as the papers cited in our ms. We have added a separate Chapter 2 – "Definitions and review methods", where we explain our own definitions.
   Edited text: "There is, however, a discrepancy on the relative effects of eutrophication and climate change in explaining past variations in phytoplankton communities and biomass."

36) p. 5, l. 125 add BS literature: Kiljunen, M., Peltonen, H., Lehtiniemi, M., Uusitalo, L., Sinisalo, T., Norkko, J., ... & Karjalainen, J. (2020). Benthic-pelagic coupling and trophic relationships in northern Baltic Sea food webs. Limnology and Oceanography, 65(8), 1706-1722

Reference added.

37) p. 5, l. 124 Again you talk about changes without specifying which phytoplankton groups were replaced by which other groups. Pls specify.

On line 124 we did not refer to specific phytoplankton groups. Edited as explained in replies to comment numbers 33 and 34.

38) p. 5, l. 127 switch community and biomass as so far you have mainly talked about community

Edited as suggested.

39) p. 5, l. 128 Give an example for dominant species, pls.

New text: "For instance, spring phytoplankton biomass increased in the Baltic Proper and decreased in the Belt Sea area, both areas showing oscillations between communities dominated by diatoms or dinoflagellates (Wasmund et al., 2011)."

40) p. 5, l. 128 Dominating in density (aka abundance) or biomass?

We refer to biomass. See answer to Comment no. 39.

41) p. 5, l. 129 Add something like „...leading to a switch from group x to group y." after effect. Pls be more specific.

We agree that this paragraph was not specific enough, and we have now completely rewritten this chapter.
New/edited text: "A study comparing historic phytoplankton communities from 1903-1911 with the present ones (1993-2005) in the northern Baltic Proper and the Gulf of Finland observed an undefined "period effect", characterized by a decline of diatoms and increase of dinoflagellates, that was not well explained by the available environmental variables (temperature, salinity and climatological data). Although data on biogeochemical parameters was not available for the period 1903-1911, the authors interpreted the observed community change as evidence of the direct and/or indirect influence of eutrophication (Hällfors et al., 2013).
A fifteen-year study (2000-2014) using FerryBox observations, covering the area between Helsinki (Gulf of Finland) and Travemünde (Mecklenburg Bight), confirmed that spring bloom intensity was mainly determined by winter nutrient concentration, while bloom timing and duration co-varied with meteorological conditions. The authors conclude that the bloom magnitude has been affected by the reduction of nutrient loading from land, while bloom phenology can also be modified by global climate change affecting seasonal oceanographic and biogeochemical processes (Groetsch et al., 2016). It has also been noted that the trends in certain groups, like cryptophytes, may be affected by anomalies in the Baltic Sea Index, a regional climate index similar to NAO, although a mechanistic explanation for the relationship could not be found (Griffiths et al., 2020)."

42) p. 6, l. 131 Pls define BSI

New text: "It has also been noted that the trends in certain groups, like cryptophytes, may be affected by anomalies in the Baltic Sea Index, a regional climate index similar to NAO, although a mechanistic explanation for the relationship could not be found (Griffiths et al., 2020)."

43) p. 6, l. 132 Pls, explain which change you mean, e.g. be more specific. Otherwise the reader has no clue of what quality the changes are that you review about.
44) p. 6, l. 132 Density or biomass (e.g. cell-carbon) wise community changes?

Edited text: "Other studies did not find any explanation for the observed changes in the biovolumes of different taxa, e.g. decrease of diatoms and increase of certain dinoflagellate taxa, and concluded that phytoplankton community in the Baltic Sea is not in a steady state (Olli et al., 2011),..."

45) p. 6, l. 138 Do you mean phytoplankton or cyanobacteria or both?

The cited study does not make this distinction.
Edited text: "...and together with increased internal loading of nutrients (Stigebrandt et al., 2014), several modelling studies project an increase in total phytoplankton concentration (in mgChl m$^{-3}$) until the end of the century (Meier et al., 2012a; Meier et al., 2012b; Skogen et al., 2014; Ryabchenko et al., 2016)."

46) p. 6, ll. 140-141 How does that mechanism work? Pls explain in more detail. Really, evidence? Or rather indication? What kind of evidence do you refer to? Also, do you mean predicted or projected climate change?

We have now edited the text thoroughly and explained the process affecting phytoplankton more carefully.

We move the rest of the description of the Kattegat regime shift to the Chapter Climate change and regime shifts.

New/edited text (in this chapter): "This has also been seen in Kattegatt, where reduction of nutrient loading led in mid 1990s to a shift from a highly eutrophic state characterized by small phytoplankton species and low water transparency to increasing share of diatoms, decreasing overall phytoplankton biomass and increase of water transparency (Lindegren et al., 2012). The improving oxygen conditions and increase of water temperature, induced by cyclic climatic variations (positive NAO) and gradual warming of climate, respectively, mainly affected the benthic ecosystem (Lindegren et al., 2012)."

New text (in Climate change and regime shifts): "Also in Kattegat, the western Baltic Sea, where the ecosystem is more oceanic than in the Baltic proper and the northern Baltic Sea, a drastic regime shift was detected in mid 1990s. First, a drastic reduction of nutrient loading, led into a shift from a highly eutrophic, pelagic ecosystem state to an ecosystem characterized by decreasing overall phytoplankton and meso- and microzooplankton biomass, dominance by small sized fish in the pelagial, an increase of macroalgae and filter-feeding molluscs in the hard bottoms and other benthic animals in the soft sediments (Lindegren et al., 2012). Second, climate variability, i.e., positive NAO and Baltic Sea index (a regional climate index), has increased inflow of well oxygenized water from the North Sea into the area, improving conditions for zoobenthos, including, e.g., populations of the commercially important Norway lobster. Further, increase of sea surface temperatures, possibly induced by global climate change, probably has contributed to the improved flatfish growth and survival in the shallow nursery areas (Lindegren et al., 2012). Decreasing fishing may also have been contributed to the increase of gadoids and flatfish, but its relative importance is difficult to distinguish from other co-occurring effects. However, it is obvious that regime shifts are often a result of several environmental, climatic and anthropogenic effects acting synergistically on the entire ecosystem."

47) p. 6, ll. 142-145 Consider moving into section 3.1. Climate change and primary production in the pelagial

Thank you for the comment. We notice now that there is much overlap between chapters "Phytoplankton and cyanobacteria" and "Climate change and primary production in the pelagial". We decided to move part of the contents of the latter chapter to separate chapters "Phytoplankton" and "Cyanobacteria". These two chapters are now rewritten, to give a more coherent picture concerning pelagic primary producers.

48) p. 6, ll. 143-145 Pls revise sentence for correct grammar.

Grammar corrected.

49) p. 6, ll. 143-145 Add detailed info, e.g.: "..increases by xy% compared to the control of ambient conditions."

Edited text: "E.g. the biomass of southern Baltic autumn phytoplankton (kept in 1400-L indoor mesocosms for 21 days) increased when $pCO_2$ was increased from 439 ppm to 1040 ppm, also under warm conditions (Sommer et al., 2015)."

50) p. 6, l. 143 Add „the" before „water"

Added as suggested.

51) p. 6, l. 148 Wording: „decrease in" or "release in grazing pressure from", your choice.

Wording changed to: "caused by an associated decrease of grazing by copepod nauplii."

52) p. 6, ll. 150-154 Long sentence. Pls break up in shorter ones for better readability.

   Sentence split in two.

53) p. 6, l. 150 Have you defined at the beginning of your review, what you mean anytime you say CC? If not, pls add. If CC includes different variables in different parts of the text, I think you need to specify, which variables you refer to in each case.

   A valid comment. We have now added a chapter 2 – Definitions and review methods after the Introduction. We explain all main definitions there.

54) p. 6, l. 151 add Jerney, J., S. Suikkanen, E. Lindehoff and A. Kremp (2019). Future temperature and salinity do not exert selection pressure on cyst germination of a toxic phytoplankton species. Ecol. Evol. 9: 4443-4451, doi: 10.1002/ece3.5009

   Reference added.

55) p. 6, ll. 150-154 convoluted sentence, pls revise for clarity.

   Sentence split in two (cf. comment no. 52)

56) p. 6, ll. 154-155 I don´t think cyanobacteria per se are a problem. Unicells should not be a problem, right? Yet they are cyanobacteria. The problem are large, unpalatable Cyanobacteria, which should be specified here and elsewhere.

   A valid point. New/edited text: "As toxins of both dinoflagellates (Sopanen et al., 2011) and cyanobacteria (Karjalainen et al., 2006; Karjalainen et al., 2007; Engström-Öst et al., 2017) can accumulate in Baltic Sea zooplankton and induce lower grazing rates and higher mortality, these studies suggest that toxic dinoflagellates and filamentous unpalatable cyanobacteria may get a competitive advantage against diatoms in a future Baltic Sea."

57) p. 6, l. 155 Why another? Which is the first competitive advantage to begin with that you seem to refer to?

   Phrase "yet another" is deleted.

58) p. 6, ll. 154-155 This is contradictory to what is stated in ll. 603-605.

   A valid point. Therefore, we delete "yet another" (see response to comment no. 57). The sentence on lines 603-605 refers to studies that note that reducing nutrient loading according to BSAP would decrease the intensity of cyanobacteria blooms in the future. These are two counteracting effects, but it is obvious that modelling studies cannot presently take into account complex biological interactions between species. This is now highlighted as one of the knowledge gaps.

59) p. 6, l. 157 Consider the field study: Eglite, E., Wodarg, D., Dutz, J., Wasmund, N., Nausch, G., Liskow, I., et al. (2018). Strategies of amino acid supply in mesozooplankton during cyanobacteria blooms: A stable nitrogen isotope approach. Ecosphere, 9, e02135. https://doi.org/10.1002/ecs2.2135

   Thank you for the suggestion, reference added.
   New text: "Sufficient supply of essential compounds such as amino acids (AA) produced by phytoplankton and cyanobacteria is essential for the growth and productivity of zooplankton grazers. It has been suggested that this supply may change if the sear surface temperature increases. A field study performed in the Baltic Proper explored the natural abundances of AA in particulate organic matter and mesozooplankton (Eglite et al., 2018). The results show that, during a warm summer, thermophilic rotifers and cladocerans (e.g. *Bosmina* spp.) acquired ample AA through filter feeding on the abundant diazotrophic cyanobacteria, whereas the temperate copepods (e.g. copepods *Temora longicornis* and *Pseudocalanus* spp.) avoided the warm surface

layer and acquired AA mainly through sinking organic matter and/or via grazing on chemoautotroph based microbial food web in the suboxic zone. This may imply that thermophilic zooplankton species, such as rotifers and certain cladocerans gain more AA than copepods in a future warmer and more stratified Baltic Sea."

60) p. 6, ll. 159-161 Like whom? Pls add species.

Species added. New/edited text: "Several studies have confirmed that during the 1980s and 1990s marine copepod species (e.g., *Pseudocalanus* spp. and *Temora longicornis*) declined, while euryhaline and limnetic, smaller-sized copepod species (*Acartia* spp. And *Eurytemora* spp.) increased in abundance (Hänninen et al., 2015; Suikkanen et al., 2013)."

61) p. 6, l. 160 add "copepod" after small-sized

Added as suggested.

62) p. 6, l. 160: delete the comma after „small sized"

Deleted as suggested.

63) p. 6, l. 161 Give examples of marine taxa.
64) p. 6, l. 162 Give examples for brackish-water taxa.

Examples given. See reply to comment no. 60.

65) p. 6, l. 164 add the following study to explain at least one underlying mechanism: Dutz, J., & Christensen, A. M. (2018). Broad plasticity in the salinity tolerance of a marine copepod species, Acartia longiremis, in the Baltic Sea. Journal of Plankton Research, 40(3), 342–355. https://doi.org/10.1093/plankt/fby013

Thank you for the suggestion. Reference added.
   New text: "Also, it has been experimentally shown that close to the physiological tolerance limit for salinity, respiration of copepods (*Acartia longiremis*) increases and feeding rate decreases (in 610 ml bottles, 24 h experiments) , indicating a disruption of the energetic balance under low salinity (Dutz and Christensen, 2018)."

66) p. 7, l. 167: Small scale impacts like what? Pls be more specific.

Edited text: "Environmental impacts on the physiology of the more sensitive species may however affect their reproductive success, and thus influence both populations and communities (Möller et al., 2015)."

67) p. 7, l. 174: In which way are cladocerans and rotifers different functional groups e.g. after the definition of Tilman (2001)? They are different taxonomic groups but as stated it is not clear why they would represent a functional group. E.g. which specific function do they represent? Also: "Shift" from whom? Herbivory, omnivory or carnivory are functional groups. Do you mean eventually a shift from omnivorous copepods to herbivorous cladocerans and rotifers? I´m confused, pls clarify. Again, a definition how you define "functional group", "functional diversity" etc is urgently needed in this review. Again, I suggest definitions of terms as given in the glossary of "Tilman, D. (2001). Functional diversity. Encyclopedia of biodiversity, 3(1), 109-120."

A valid comment. We here refer to filter-feeding vs. raptorially feeding species, not herbivorous or carnivorous. We made an attempt to clarify the processes better.
   New/edited text: "Changes in zooplankton functional groups, such as a shift from raptorially and suspension-feeding copepods and cladocerans to a dominance by filter-feeding rotifers and cladocerans, have been shown as a result of warming (Suikkanen et al., 2013; Jansson et al., 2020). Also, a switch from predominantly herbivorous feeding to carnivory (feeding on ciliates)

has been observed in a field study in the southern and central Baltic Sea, during cyanobacterial blooms (Loick-Wilde et al., 2019), probably supported by decomposing of the otherwise unpalatable filamentous cyanobacteria, and an associated increase of the heterotrophic pathways of energy (Hogfors et al., 2014). It is possible that the functions of zooplankton community will change as climate-induced warming and reduced salinity continues."

68) p. 11, l. 337 What about food quality and transfer efficiency of mass and energy?

Food quality added. We consider transfer efficiency in the chapter "Nutrient cycling, benthic pelagic coupling and trophic efficiency".
Edited text: "Future climatic variations may affect fish in the Baltic Sea through their effects on water temperature, salinity, oxygen and pH, as well as nutrients, which indirectly affect availability and quality of food for fish."

69) p. 11, l. 338: After „fish" add: Limburg, K. E., & Casini, M. (2019). Otolith chemistry indicates recent worsened Baltic cod condition is linked to hypoxia exposure. Biology letters, 15(12), 20190352. Möllmann C, Cormon X, Funk S, Otto SA, Schmidt J, Schwermer H, Sguotti C, Voss R, Quaas M (2021): Tipping point realized in cod fishery, Nature Scientific Reports, DOI: https://doi.org/10.1038/s41598-021-93843-z

Thank you for the suggestion. References added.

70) p. 13, l. 390: Consider adding the following study to determine trophic efficiency in field samples, because it includes examples also from the Oeland upwelling and larger Baltic Proper in the Fig. 6 : Weber, S. C., Loick-Wilde, N., Montoya, J. P., Bach, M., Doan-Nhu, H., Subramaniam, A., ... & Voss, M. (2021). Environmental regulation of the nitrogen supply, mean trophic position, and trophic enrichment of mesozooplankton in the Mekong River plume and southern South China Sea. Journal of Geophysical Research: Oceans, 126(8), e2020JC017110. namely Chapter:"4.4. Ecosystem-Specific Trophic Enrichment in Mesozooplankton"

Thank you for the suggestion. We however want to concentrate citing papers that have a focus on the Baltic Sea, and therefore prefer not to include this study.

71) p. 13, l. 390: trophic efficiency: Pls define in the text or glossary

Defined in chapter 2 – Definitions and review methods.

72) p. 13, l. 390 At the moment it does not become clear why some processes are reviewed and others are missing. E.g. what do we know about key processes like nitrification, denitrification, N2 fixation in a future Baltic Sea?

Suggested References from whichs´results at least some clues may be deduced:
Bartl, I., Hellemann, D., Rabouille, C., Schulz, K., Tallberg, P., Hietanen, S., & Voss, M. (2019). Particulate organic matter controls benthic microbial N retention and N removal in contrasting estuaries of the Baltic Sea. Biogeosciences, 16(18), 3543-3564.
Allin, A., Schernewski, G., Friedland, R., Neumann, T., & Radtke, H. (2017). Climate change effects on denitrification and associated avoidance costs in three Baltic river basin-coastal sea systems. Journal of Coastal Conservation, 21(4), 561-569.
Asmala, E., Carstensen, J., Conley, D. J., Slomp, C. P., Stadmark, J., & Voss, M. (2017). Efficiency of the coastal filter: Nitrogen and phosphorus removal in the Baltic Sea. Limnology and Oceanography, 62(S1), S222-S238.
Hellemann, D., Tallberg, P., Bartl, I., Voss, M., & Hietanen, S. (2017). Denitrification in an oligotrophic estuary: a delayed sink for riverine nitrate. Marine Ecology Progress Series, 583, 63-80.
Olofsson, M., Klawonn, I., & Karlson, B. (2021). Nitrogen fixation estimates for the Baltic Sea indicate high rates for the previously overlooked Bothnian Sea. Ambio, 50(1), 203-214.

Loick-Wilde, N., Weber, S. C., Eglite, E., Liskow, I., Schulz-Bull, D., Wasmund, N., ... & Montoya, J. P. (2018). De novo amino acid synthesis and turnover during N2 fixation. Limnology and Oceanography, 63(3), 1076-1092.

Thank you for the suggestions. We have not reviewed biogeochemical processes thoroughly, because in the ESD BEAR Special Issue, there is a separate manuscript, Kulinski et al., on biogeochemistry of the Baltic Sea https://esd.copernicus.org/preprints/esd-2021-33/esd-2021-33.pdf : Almost all the suggested papers are reviewed in that paper. In our review we have highlighted mainly papers that connect biogeochemical processes directly to biota. We therefore choose not to cite the suggested papers in our review.

73) p. 13, l. 412-415: This is interesting! Pls explain the mechanism behind this at least briefly.

New/edited text: "It has also been suggested that climate change may decrease fish productivity, especially in the northernmost Baltic Sea, because, when the system shifts towards heterotrophy, the food web efficiency declines, due to competition for nutrients between bacteria and phytoplankton, and the phytoplankton production decreases. This creates less food for zooplankton and planktivorous fish, and also decreases sedimentation of organic matter, reducing benthic production and diminishing food availability for benthic-eating fish. Eventually the fish production may decrease (Berglund et al., 2007; Wikner and Andersson, 2012)."

74) [*No comment under this number.*]

75) p. 13, l. 413: Add „switches" after „system"

Corrected, albeit with the word "shifts".

76) p. 13, ll. 415-418: I don´t understand this. If the food web bases on heterotrophy rather than photoautotrophy, doesn´t that imply that less mass and energy is transfered to fish since the lower food chain is elongated (based on heterotrophs rather than autotrophs) leading to a higher trophic position in mesozooplankton (e.g. carnivorouse zoops instead of herbivorouse zoops, e.g. as documented Loick-Wilde et al. 2019)? Then e.g. eastern Baltic cod would also have a higher TP in such an, at times, heterotrophy based system, compared to a lower TP in western Baltic cod due to a a phototrophy based system, right? Pls clarify.

We agree that the wording was not clear enough. We have now revised the text to better explain the rationale behind the idea of microbial loop sustaining high zooplankton and fish production, and the associated uncertainties.
New/edited text: "Certain mesocosm studies simulating effects of climate change have however found that the production and biomass of copepods can remain high, even when they feed upon the longer bacteria-flagellate-ciliate food chain, because the positive effects of increasing temperature on copepod production override the negative effects of decreasing food web efficiency (Lefebure et al., 2013). Furthermore, many Baltic Sea copepods are omnivorous and can opportunistically switch between suspension feeding on flagellates and raptorial feeding on ciliates (Kiorboe et al., 1996). This creates an ´intraguild´ relationship between the three trophic levels, flagellates, ciliates and copepods (Gismervik and Andersen, 1997), which stabilizes the system and can sustain copepod production even under lower phytoplankton production. If copepod production remains high, also fish production may be supported also when the system shifts to heterotrophic production (Lefebure et al., 2013). On the other hand, a study performed in a large biotest area artificially heated by the cooling waters of the Forsmark Nuclear power plant, southern Bothnian Sea, found that warming of water may lead to increased species turnover, and in decreased compositional stability of diatom, macrophyte and invertebrate communities (Hillebrand et al., 2010). As it is challenging to incorporate such complex interactions in 3D ecosystem models, the consequences of climate change on trophic efficiency and future fish production under different scenarios and in different sea areas remain unsecure."

77) p. 15, ll. 461-467: I disagree with "warming induces a switch from a bottom-up controlled to a mainly top-down controlled system, which may result in increased zooplankton abundance and reduced phytoplankton biomass under warm temperature". What consequences do you think it has for higher trophic levels like fish or sea birds if mesozooplankton switches from herbivory to carnivory due to increasing densities of unpalatable cyanobacteria in a future Baltic Sea? According to a simple biogeochemical model (see Figure below from T&T 2011), the decrease in mass and energy that is available for TPs above the mesozooplankton compartment should be massive, shouldn´t it? Pls discuss in a larger context.

[Figure]

(Thurman & Trujillo, Fig. 14.22)

Figure 14.22 shows the passage of energy between trophic levels through an entire ecosystem, from the solar energy assimilated by the autotrophic plankton through all trophic levels to piscivorous humans. From the assimilated chemical energy, a large fraction is converted by respiration into kinetic energy for sustaining life or is lost as heat, and what remains is available for growth and reproduction. Thus, only about 10% of the ingested by herbivores is available for the next trophic level. Since energy is lost at each trophic level, it takes thousands of smaller marine organisms to produce a single fish that can be so easily consumed during a meal! Source: Trujillo, A. P., & Thurman, H. V. (2011). Essentials of oceanography (10th edition). Pearson Education. pp. 551.

We agree that the wording was not clear, and we have now rewritten this part to better explain the suggestions of the cited studies. In this chapter we avoid judging the validity of published studies, but try to review their rationale as best as we can.

New/edited text: "In areas where climate change increases the supply of allochtonous DOM into the system, and warms up the sea surface temperature, strengthening the stratification and reducing the availability of nutrients from deeper waters, phytoplankton production may decline and the trophic pathways from bacteria and flagellates through ciliates to copepods may strengthen (Aberle et al., 2015). It has also been suggested, from experimental (mesocosm) evidence, that warming speeds up the growth of copepods but leaves phytoplankton unaffected, which shortens the time lag between phyto- and zooplankton. This may lead to a larger and earlier zooplankton peak and increase the possibility of zooplankton controlling phytoplankton, which may lead to a reduced phytoplankton biomass under warm temperature (Paul et al., 2016). Such results highlight the importance of considering food web effects (both bottom-up and top-down) on the pelagic ecosystem under climate change."

78) p. 15, ll. 461-467: Add the following field based study about the environmental regulations of a switch from herbivory to carnivory in mesozooplankton in the Baltic Sea in summer in this paragraph: Loick-Wilde, N., Fernandez-Urruzola, I., Eglite, E., Liskow, I., Nausch, M., Schulz-Bull, D., et al. (2019). Stratification, nitrogen fixation, and cyanobacterial bloom stage regulate the

planktonic food web structure. Global Change Biology, 25(3), 794–810. https://doi. org/10.1111/gcb.14546.

> Reference added. New/edited text: "Changes in zooplankton functional groups, such as a shift from raptorially and suspension-feeding copepods and cladocerans to a dominance by filter-feeding rotifers and cladocerans, have been shown as a result of warming (Suikkanen et al., 2013; Jansson et al., 2020). Also, a switch from predominantly herbivorous feeding to carnivory (feeding on ciliates) has been observed in a field study in the southern and central Baltic Sea, during cyanobacterial blooms (Loick-Wilde et al., 2019), probably supported by decomposing of the otherwise unpalatable filamentous cyanobacteria, and an associated increase of the heterotrophic pathways of energy (Hogfors et al., 2014)."

79) p. 17, l. 535: There is also a significant knowledge gap about the chances of new methodological approaches. So how about including methodological improvements that allow for a significant reduction in trait complexity while considering intraspecific variations in biological samples and specifically allow for the calibration and validation of current biogeochemical models?

Using compound-specific isotope analyses (CSIA) of amino acid nitrogen, it is now possible to measure a continuous trophic position in any biological compartment (as opposed to discrete trophic levels) based on a single field sample, which integrates the assimilation of mass from all the trophic pathways leading to a top predator from different field locations. With this information, we can take the next step of relating the effective TPs e.g. of zooplankton to the environmental conditions measured in situ (Loick-Wilde et al., 2019), providing much needed insights into the mechanisms driving shifts in TP.

The strength of CSIA lies in providing information on both TP and N sources from a single organism/sample, which is achieved with a simple comparison of the $\delta$ 15N values of glutamic acid (Glu) and phenylalanine (Phe) amino acids (McClelland & Montoya, 2002; Mompean et al., 2016). While Glu is enriched in 15N by ~8.0‰ per trophic transfer (Chikaraishi et al., 2009), the $\delta$ 15N of Phe remains nearly unchanged when the amino acid (AA) is transferred through the food web and thus reflects the isotopic composition of the primary producers (Nsource measure, Chikaraishi et al., 2010). This approach largely eliminates potential sources of error in TP estimates associated with temporal and physiological decoupling between a consumer and its diet, and has been refined and confirmed in numerous field- and lab-based trophic studies over the last decade (reviewed by Glibert et al., 2019 and Ohkouchi et al., 2017).

The CSIA based N source identification and mean trophic position from cross-compartment analyses can directly be used to calibrate and validate current biogeochemical models and allow for an end-to-end quantification e.g. of N inputs from N2 fixation into apex predators like cod or sea birds.

> A good point and we appreciate the suggestion. We have added a notion on novel methods in Knowledge gaps.
> New text: "Also, more extensive use of biochemical and genetic methods, such as biomarkers (Turja et al., 2014; Turja et al., 2015; Villnäs et al., 2019), stable isotopes (Voss et al., 2000; Gorokhova et al., 2005; Morkune et al., 2016; Lienart et al., 2021), compound-specific isotope analyses (Ek et al., 2018; Weber et al., 2021) or metabarcoding (Leray and Knowlton, 2015; Bucklin et al., 2016; Klunder et al., 2021) could yield novel information on community structure, stress levels experienced by organisms, and of the trophic position of various taxa, under environmental change. Such information would allow validation of current biogeochemical models under different environmental scenarios, including climate change."

80) p. 17, l. 535: Why do you refer only to salinity and stratification, what about the other abiotic variables like temperature, oxygen, OA, nitrate or phosphate inputs (either through rivers or from the anoxic sediments)?

> A valid comment. New/edited text: "Projections of sea surface temperature and ice conditions are held relatively reliable, but, despite more than two decades of 3D modelling, there are still large

uncertainties in projecting certain basic physical parameters like salinity level as well as stratification under different climate forcings. Consequently, it is also difficult to project all parameters affected by stratification, especially oxygen levels and release of nutrients from the sediments, during different periods, at different depths and in different sea areas. Also, long term variation in external loading of nutrients from rivers depends, in addition to the magnitude of anthropogenic loading, also biogeochemical processes in the soil and in lakes and rivers. All these uncertainties weaken our ability to project marine biological processes, from pelagic primary and secondary productivity and benthic-pelagic coupling to zooplankton and fish populations, and to geographic shifts in macroalgal communities and invertebrates inhabiting the photic zone."

81) p. 19, l. 604: The following review is less certain about a future decrease in cyanobacteria, pls discuss more controversial: Munkes, B., Löptien, U., & Dietze, H. (2021). Cyanobacteria blooms in the Baltic Sea: a review of models and facts. Biogeosciences, 18(7), 2347-2378.

What about the internal P storage in the sediments that gets released under anoxic conditions? Stigebrandt, A., Rahm, L., Viktorsson, L., Ödalen, M., Hall, P. O., & Liljebladh, B. (2014). A new phosphorus paradigm for the Baltic proper. Ambio, 43(5), 634-643.

Thank you for the comment. The reference is added to the chapter on Cyanobacteria. Citations on Stigebrandt et al. (2014) are inserted in several places where internal loading is dealt with.
New/edited text: Intensified blooms of cyanobacteria are expected especially if hypoxia will prevail and internal loading will increase supply of phosphorus from anoxic sediments into the surface layer, decreasing the N:P ratio (Meier et al., 2011b; Funkey et al., 2014). However, the magnitude of changes and their consequences for biogeochemical processes, e.g. for nitrogen fixation, differ greatly between models (Munkes et al., 2021)."

82) p. 19, l. 608: What about empirical field observations/ research? Pls add.

Thank You for the suggestion. Edited text: "These studies further highlight the importance of studying the Baltic Sea as a socio-ecological system, responding to both environmental and societal changes (Bauer et al., 2018; Bauer et al., 2019; Hyytiäinen et al., 2019), and it is important to continue efforts combining empirical field studies, experimental studies, modelling and dialogue with human society in order to attune to the changes ultimately driven by the Ocean itself (Stenseth et al., 2020)."

83) p. 20, l. 623: add any missing field and lab studies as pointed out in the text here, too, when applicable.

The Table 1 is updated to include all papers suggested, and some more.

**Technical corrections**

Just few, part of the specific comments!

A few typos corrected and some grammatical improvements made.

---

## Author Comment (AC4)

Manuscript "Global climate change and the Baltic Sea ecosystem: direct and indirect effects on species, communities and ecosystem functioning"
by Markku Viitasalo and Erik Bonsdorff, submitted to Earth System Dynamics.

Author comments for reviewer no. 2 (esd-2021-73-RC2-supplement.pdf).

All author replies to reviewer comments in red font.

**Anonymous reviewer #2**

**General comments**

The review provides a valuable overview of past decades scientific studies with relation to climate change projection and the Baltic Sea.

I miss data synthesis in terms of figures and statistical tests on cross-experimental and cross-ecosystem data, proving significance of made conclusions.

More attempts to weight the importance of different factors would make some scenarios to be presented as more likely than others. Several sections is now a list of different outcomes with seemingly similar probability to occur.

A more critical view on the ability to prove climate effects would enhance the scientific value of the manuscript. The same is true for lack of understanding of adaptive and evolutionary processes for the outcome of projected climate change.

> We find these comments valid and agree that more analysis of the published results will improve the value of the manuscript. We have made a serious attempt for a data synthesis and evaluation of the attribution of the found effects to climate change in light of the evidence published since 2010 (see replies to Detailed comments, below).

**Detailed comments**

r. 20 Effects of climate would explicitly require statistically significant changes attributed to climate factors. Since this type of data are scarce, scientific "evidence on effect of climate" is unlikely to be found. I suggest a rephrasing.

> A valid point. New text: "Studies investigating species-, population- and ecosystem-level effects of abiotic factors that may change due to global climate change, such as temperature, salinity, oxygen, pH, nutrient levels, and the more indirect biogeochemical and food web processes, are reviewed, primarily from published literature after 2010."

r. 22 Please specify "responses" of what effectors? By which type of species?

> Edited text: "The responses to the studied abiotic factors vary within and between taxonomic groups, (microbes, phytoplankton, zooplankton, benthic algae and vascular plants, macrozoobenthos and fish), species, and even between sibling species (as is the case with the brown alga *Fucus vesiculosus*)."

r. 25 "will improve " is ambiguous. Increase or decrease cyanobacterial blooms? Reduced blooms would be an improvement.

> Edited text: "It is likely that the combined effects of increased external nutrient loads, stratification and internal loading will favour formation of cyanobacterial blooms in large parts of the Baltic."

r.26 The impact of allochthonous carbon is primarily influenced by the specific loading of organic matter. Not the latitude.

Edited text: "In areas strongly influenced by allochtonous DOM, such as the northern parts of the Gulf of Bothnia, increasing freshwater runoff may further complicate the process by increasing heterotrophy and by decreasing food web efficiency."

r. 29 Influence of organic matter is primarily hampering photosynthetic production. That cannot be counteracted by the proposed food chain. Please remove or adhere hypothesis better to current knowledge.

Yes, the text was not clear. We edit the text in abstract and in the respective chapter, "Complex food web responses in the microbial loop."

New text (in Abstract): Warming of seawater in spring also speeds up zooplankton production and shortens the time lag between phytoplankton and zooplankton peaks, which may lead to zooplankton controlling phytoplankton and reduced phytoplankton biomass in spring.

r. 44 To uncertainties the adaptation and even evolution of organisms in most trophic levels driven by changed climate should be pointed out. This is not possible to study in short term experiments or modelling.

We agree. New/edited text: "Experimental studies can indicate how species and populations respond to projected levels of abiotic variables, such as temperature, salinity, oxygen or pH, but they cannot show how much species can adapt to slow shifts in the environment, in the time scale of 50 to 100 years. Short-term experiments are also weak in demonstrating the long-term effects of potential changes in the structure of the food web. Experimental work should therefore be better integrated into field and modelling studies of food web dynamics, to get a more comprehensive view of the responses of the pelagic and benthic systems to climate change, from bacteria to fish."

r.60 The shortcoming of not covering meteorological definition of climate change should be mentioned (i.e., significant differences between 30 year periods). One may even question of it is meaningful to make scientific conclusions of climate effects on chemistry and biology.

Yes, we agree, and we have now added an entire new chapter 2 – Definitions and review methods in the beginning of the ms. We hope this will help the reader to better follow the reasoning between climate issues raised. We also better define where we speak of NAO and other climate fluctuations, and when of global climate change. As suggested in the general Comments by the reviewer, we have tried to clarify this distinction throughout the ms., and also increased our analysis of attribution of the observed effects to either "natural" variations in climate and the global climate change

We do think that it is meaningful to review published literature on responses on *climate related parameters*, i.e., parameters that have been projected to change due to global climate change, at different spatial and temporal scales. By reviewing past changes in pelagic and benthic communities, and short-term experiments, we consider effects of short/medium term changes in such parameters.

r. 88 OAW is defined as abbreviation but OA used below. Please harmonize.

Edited text: "The effects of projected ocean acidification (OA) on microbes have been studied alone, and together with other abiotic variables, such as temperature and salinity."

r. 104 Please present the duration of those experiments in relation to organism generation time and discuss its influence on the conclusions that can be made.

A valid comment. We have now added notions of the sizes if the micro- and mesocosms, as well as duration of the experiments. They vary from 12 to 1400 litres, and from 12 to 24 days. We have also added in the knowledge gaps a note on the limited temporal and spatial scales of most experimental work.

r. 109 This could primarily be associated with weather changes given the periods investigated (i.e., mainly within a 30-year period).

Yes, that is a valid point, and the same time scale factor was pointed out by Reviewer #1. We have thoroughly rewritten this chapter, and also added more scrutiny to the attribution of the observed phenomena to global climate change, elsewhere in the ms. (see response to the General Comments, above).

The chapter "Phytoplankton and Cyanobacteria" are also divided in two separate chapters, "Phytoplankton" and "Cyanobacteria", for increased clarity.

New text: "Some studies have attributed these shifts to changes in environmental conditions associated with global change (Groetsch et al., 2016), while others have indicated a connection with the North Atlantic Oscillation (NAO); a decline in the intensity of NAO in the 1990s caused less cloudy conditions (more irradiance), and less windy conditions induced stronger stratification of surface water (Hjerne et al., 2019). Such shifts, if caused by variations in NAO, may be temporary and reversable, whereas shifts caused by changes in global climate may be more enduring. It has been suggested that, in the future climate, higher temperatures and less ice will cause an earlier bloom of both diatoms and dinoflagellates, with increased dinoflagellate dominance, but this development may be counteracted by increasing windiness and cloudiness, which have also been projected by certain modelling studies (Hjerne et al., 2019). However, more recent studies have suggested that, while the winter conditions will most probably become more cloudy and windy, the projections for spring and summer are more uncertain (Christensen et al., 2021). The exact nature of the changes in the structure of spring phytoplankton communities in the next 60 to 80 years cannot be projected with certainty."

r. 188-190 This firm conclusion would merit from a presentation of a strong relationship between eutrophication and "shallow coastal water areas". Please specify what quantities that is used to indicate both factors and the strength of the statistical relationship.

Yes, we have edited the text and added some relevant references. Olsson et al. (2015) did not explicitly report statistical significances between macroalgal communities and nutrients, or climate related factors (only relationships between food web structure and principal component axes were reported), so we do not cite any statistics here.

Edited text: "For many shallow coastal ecosystems of the Baltic Sea, it has been concluded that eutrophication, whether being caused solely by anthropogenic nutrient loads, or amplified by climate change, has been the most important pressure affecting the ecosystem components (Olsson et al., 2015). This is plausible, because of the strong influence of anthropogenic nutrient loading in coastal areas, especially those that are prone to hypoxia due to topography (Virtanen et al., 2018), and which often are affected by internal loading of phosphorus from the sediment (Puttonen et al., 2014; Puttonen et al., 2016)."

r. 195-196 The presented ranges of temperatures investigated does not appear to include the natural variation observed of the annual cycle. Please comment.

The temperatures used by Graiff et al. (2015) and Takolander et al. (2017), which were intended to simulate heat waves, 27 to 29 °C, do cover the possible natural variation observed during an annual cycle. We do not think it is necessary here, in the context of SST warming, to comment on the lower end of the temperature range (ca 0 °C). No change is made in the ms.

r.221 Good that also adaptation is discussed here. Please also include in the introduction.

We have now amended the text on the issue of adaptation in several places in the ms.

New text [Abstract]: "Experimental studies can indicate how species and populations respond to projected levels of abiotic variables, such as temperature, salinity, oxygen or pH, but they cannot show how much species can adapt to slow shifts in the environment, in the time scale of 50 to 100 years."

New text [Introduction]: "It is also challenging to assess the capacity of species to genetically evolve and adapt to the relatively small and very slow changes in abiotic parameters, and associated changes in species interactions."

New text [Conclusions]: "While it has been suggested that Baltic marine species may have, due to isolation and genetic endemism, diminished potential for adaptation, several recent studies have pointed out that, e.g., macroalgae have phenotypic plasticity and potential for adaptation against gradual changes in the abiotic environment."

r. 255-258 This is one of several examples where direct or indirect effect by climate change on biota is not part of the conclusion (cf. title of the MS). The statement is just a list of factors influencing the organisms today and is assumed to do so in the future, however, without proposing the net outcome of this (i.e. the effect).

Yes, we agree that, from correlative studies, little can be projected for the future. We have modified this paragraph accordingly, and we have also added analyses on attribution to climate change in several places in the ms. (see also response to General Comments, above).

Edited text: "While such correlative studies provide evidence on the factors that have driven past changes, they cannot be used to deduce how species, populations, and benthic biomass, would change in the future. Several studies using ecosystem modelling have however suggested that climate-induced changes in salinity, temperature and eutrophication (affecting both food availability and oxygen levels), will also be of importance for development of benthic communities and their biomass (Timmermann et al., 2012; Ehrnsten et al., 2019a; Ehrnsten et al., 2019b)."

r. 267-269 Do you mean that projected climate driven temperature (increase) may lead to a rapid increase in hypoxia? increase in temperature? Please rephrase accordingly if so. The conclusion that potential phosphorous release alone will cause eutrophication is premature. Projected enhancement of precipitation and river discharge of organic matter may counteract this by reducing light irradiance to the water column.

Yes, we refer to the strengthening temperature stratification to summer. We rephrase and add relevant references.

Thank You for the comment on eutrophication. We do think that it is worth suggesting that increased temperature stratification may increase the risk of eutrophication in shallow areas with poor water exchange. We however add a notion on the uncertainties concerning the speculated process.

New/edited text: "As increasing sea surface temperature will strengthen stratification, late summer hypoxia may increase in such coastal areas. This may increase the release of phosphorus from anoxic sediments (Puttonen et al., 2016) and lead to a "vicious circle of eutrophication" (Vahtera et al., 2007), that will bring generate more nutrients to the system and impede the success of nutrient reductions from land (Stigebrandt et al., 2014). However, several processes may counteract this process. Decreasing ice cover and changes in future wind conditions (of which no consensus exists) may affect both seasonal nutrient dynamics and stratification. Also, changes in benthic species composition may affect the oxygen dynamics of muddy sediments via species specific bioirrigation patterns (Norkko et al., 2012). Such processes that are dependent of traits of a few species may be of particular importance in low-diversity systems such as the northern Baltic Sea (Gladstone-Gallagher et al., 2021)."

r. 305 Please specify what analysis you refer to? The modelling?

Now clarified. Edited text: "Modelled scenarios of temperature and salinity have also been used to project how the change in the abiotic environment could affect NIS already present in the Baltic Sea (Holopainen et al., 2016). The modelling suggests an increase of Ponto-Caspian cladocerans in the pelagic community, and an increase in dreissenid bivalves, amphipods and mysids in the coastal benthic areas of the northern Baltic Sea until 2100 (Holopainen et al., 2016)."

r. 331-335 The main sentence and the subordinate cl(a)use appear contradictory. If the factors are difficult to disentangle, how can you then derive significant climate factors? Please clarify and rephrase.

Yes, we delete the subordinate clause for clarity.

Edited text: "A long-term study (over four decades) made at different coastal areas of the Baltic Sea illustrates that it is hard to disentangle the abiotic and biotic interactions, e.g. between fish and their food-sources (benthos) (Törnroos et al., 2019). The study also highlights possible decoupling of benthic-feeding fish from long-term changes of zoobenthos."

r. 501 Or could it be referred to as a shift in weather conditions?

The event was in several studies called "a regime shift", so we retain this wording. No change in text.

r. 571 As pointed out above few if any studies including biological variables cover at least 2 climate periods. Most also lack coverage of adaptive and evolutionary processes. This should be recognized and the statements rephrased accordingly.

We agree. Text on the uncertainties and on the difficulty of attribution is added here and elsewhere in the ms.
New text: "The purpose of most studies has been to analyse the observed changes in terms of observed or projected climate change in the Baltic Sea, or to simulate the projected changes (often until the end of the 21st century) in experiments. It is however very difficult to attribute the long-term responses observed in field populations, or responses of individual organisms in small- or medium scale experiments, to global climate change. Not much is known of the adaptation potential of species, and attribution to the anthropogenic climate change is difficult also because of overlapping climatic cycles, like the NAO and, e.g., stochastic inter-annual variations in temperature. Correlative studies using field data cannot be used for projecting future changes, especially since very few studies have considered more than one climate period caused by cyclic phenomena such as NAO."

r. 575 Please correct and shorten the sentence. Message is unclear.

Edited text: "Responses of individual species to single parameters may be relatively straightforward, but when effects of several parameters on multiple species or trophic levels are studied, the results are challenging to interpret and become increasingly difficult to attribute to the global climate change, or to climate variations in general."

r.583 "…will promote cyanobacterial blooms…". "Improve" is ambiguous.

Yes, we agree. Edited text: Several recent modelling studies project that the combined effects of increased nutrient loads, increased stratification and increased internal loading will increase the frequency and intensity of cyanobacterial blooms in the central basin of the Baltic Sea, as well as the Gulf of Finland – unless nutrient loading from land will be drastically reduced."

r. 585 The dominating effect of reducing photosynthesis is overlooked (reduced light irradiance and intensified competition for the limiting nutrient with bacterioplankton). Please include and rephrase. Again, the proposed food chain cannot counteract this. The sentences are also close to repetition of what is said in the abstract. Consider replacing by complementing text.

Yes, we agree that the text was not self-explaining. We have rewritten the paragraph, and also add a notion on the uncertainties involved. We however retain the hypothesis that top-down control may increase if warming reduces time lags between functional groups.
Edited text: "In the northernmost areas – the Quark and the Bothnian Bay – in turn, the increasing allochtonous DOM may complicate the picture by reducing light availability for photosynthesis and due to intensified competition for the limiting nutrients with bacterioplankton. Also here, many open questions remain. If the projected warming results in shortened time lags between bacteria, phytoplankton, microzooplankton, suspension feeding cladocerans and microzooplankton-eating copepods, the system may change from a bottom-up controlled one to top-down controlled one, with potential effects on food web dynamics."

r. 604. I am sceptic that cyanobacterial bloom would be markedly reduced as they are also found in sediment records representing pre-industrial conditions. Consider rephrasing sentence.

Yes, there are differing opinions whether cyanobacteria will increase or decrease in the Baltic Sea in the future. We here refer to a modelling study, Meier et al (2019), which uses the term "record-breaking blooms" and claims: "*Under the BSAP, record-breaking cyanobacteria blooms will no longer occur in the future*." We change our wording and, in several places in the ms., highlight the uncertainties

concerning projecting the cyanobacteria blooms. Here we highlight the difference between a situation where only climate change affects the system, and where both climate change and nutrient reductions would take place.

Edited text: "It has also been suggested that, with successful nutrient reductions, record-breaking cyanobacteria blooms seen in the past few decades will no longer occur, despite the proceeding climate change (Meier et al., 2019)."

Table 1. Effects by increased precipitation and discharge of organic matter is overlooked. This primarily influence phytoplankton carbon dioxide fixation but also bacterioplankton and other parts of the food web. This is demonstrated both in long-term field data and controlled mesocosm experiments.

Suggested papers and some others are included in text and added into the Table 1.

**AC2 references**

Christensen, O. B., Kjellström, E., Dieterich, C., Gröger, M., and Meier, H.: Atmospheric regional climate projections for the Baltic Sea Region until 2100, Earth System Dynamics Discussions, 1-53, 2021.

Ehrnsten, E., Norkko, A., Timmermann, K., and Gustafsson, B. G.: Benthic-pelagic coupling in coastal seas–Modelling macrofaunal biomass and carbon processing in response to organic matter supply, Journal of Marine Systems, 196, 36-47, 2019a.

Ehrnsten, E. S., Bauer, B., and Gustafsson, B. G.: Combined effects of environmental drivers on marine trophic groups-a systematic model comparison, Frontiers in Marine Science, 6, 492, 2019b.

Gladstone-Gallagher, R. V., Hewitt, J. E., Thrush, S. F., Brustolin, M. C., Villnas, A., Valanko, S., and Norkko, A.: Identifying "vital attributes" for assessing disturbance-recovery potential of seafloor communities, Ecology and Evolution, 11, 6091-6103, 10.1002/ece3.7420, 2021.

Groetsch, P. M. M., Simis, S. G. H., Eleveld, M. A., and Peters, S. W. M.: Spring blooms in the Baltic Sea have weakened but lengthened from 2000 to 2014, Biogeosciences, 13, 4959-4973, 10.5194/bg-13-4959-2016, 2016.

Hjerne, O., Hajdu, S., Larsson, U., Downing, A. S., and Winder, M.: Climate driven changes in timing, composition and magnitude of the Baltic Sea phytoplankton spring bloom, Frontiers in Marine Science, 6, 15, 10.3389/fmars.2019.00482, 2019.

Holopainen, R., Lehtiniemi, M., Meier, H. M., Albertsson, J., Gorokhova, E., Kotta, J., and Viitasalo, M.: Impacts of changing climate on the non-indigenous invertebrates in the northern Baltic Sea by end of the twenty-first century, Biological invasions, 18, 3015-3032, 2016.

Meier, H. E. M., Dieterich, C., Eilola, K., Gröger, M., Höglund, A., Radtke, H., Saraiva, S., and Wåhlström, I.: Future projections of record-breaking sea surface temperature and cyanobacteria bloom events in the Baltic Sea, Ambio, 48, 1362-1376, 10.1007/s13280-019-01235-5, 2019.

Norkko, J., Reed, D. C., Timmermann, K., Norkko, A., Gustafsson, B. G., Bonsdorff, E., Slomp, C. P., Carstensen, J., and Conley, D. J.: A welcome can of worms? Hypoxia mitigation by an invasive species, Global Change Biology, 18, 422-434, 10.1111/j.1365-2486.2011.02513.x, 2012.

Olsson, J., Tomczak, M. T., Ojaveer, H., Gårdmark, A., Põllumae, A., Müller-Karulis, B., Ustups, D., Dinesen, G. E., Peltonen, H., Putnis, I., Szymanek, L., Simm, M., Heikinheimo, O., Gasyukov, P., Axe, P., and Bergström, L.: Temporal development of coastal ecosystems in the Baltic Sea over the past two decades, Ices Journal of Marine Science, 72, 2539-2548, 10.1093/icesjms/fsv143, 2015.

Puttonen, I., Kohonen, T., and Mattila, J.: Factors controlling phosphorus release from sediments in coastal archipelago areas, Marine Pollution Bulletin, 108, 77-86, 10.1016/j.marpolbul.2016.04.059, 2016.

Puttonen, I., Mattila, J., Jonsson, P., Karlsson, O. M., Kohonen, T., Kotilainen, A., Lukkari, K., Malmaeus, J. M., and Rydin, E.: Distribution and estimated release of sediment phosphorus in the northern Baltic Sea archipelagos, Estuarine Coastal and Shelf Science, 145, 9-21, 10.1016/j.ecss.2014.04.010, 2014.

Stigebrandt, A., Rahm, L., Viktorsson, L., Odalen, M., Hall, P. O. J., and Liljebladh, B.: A new phosphorus paradigm for the Baltic proper, Ambio, 43, 634-643, 10.1007/s13280-013-0441-3, 2014.

Timmermann, K., Norkko, J., Janas, U., Norkko, A., Gustafsson, B. G., and Bonsdorff, E.: Modelling macrofaunal biomass in relation to hypoxia and nutrient loading, Journal of Marine Systems, 105, 60-69, 10.1016/j.jmarsys.2012.06.001, 2012.

Törnroos, A., Pecuchet, L., Olsson, J., Gårdmark, A., Blomqvist, M., Lindegren, M., and Bonsdorff, E.: Four decades of functional community change reveals gradual trends and low interlinkage across trophic groups in a large marine ecosystem, Global Change Biology, 25, 1235-1246, 10.1111/gcb.14552, 2019.

Vahtera, E., Conley, D. J., Gustafsson, B. G., Kuosa, H., Pitkänen, H., Savchuk, O. P., Tamminen, T., Viitasalo, M., Voss, M., Wasmund, N., and Wulff, F.: Internal ecosystem feedbacks enhance nitrogen-fixing cyanobacteria blooms and complicate management in the Baltic Sea, Ambio, 36, 186-194, 10.1579/0044-7447(2007)36[186:iefenc]2.0.co;2, 2007.

Virtanen, E. A., Norkko, A., Nyström Sandman, A., and Viitasalo, M.: Identifying areas prone to coastal hypoxia – the role of topography, BIogeosciences, 16, 3183-3195, https://doi.org/10.5194/bg-16-3183-2019, 2018.

---

## Referee Report (RR1)

**Review ESD-2021-73_V2_RX**

" Global climate change and the Baltic Sea ecosystem: direct and indirect effects on species, communities and ecosystem functioning

Viitasalo and Bonsdorff

**General comments**

In general, the compilation of results are more distinct and text easy to read.

All the "To sum up,.." paragraphs are much appreciated. Significant improvement.

The fact that many more mesocosm studies relevant for climate-change effect has been reported need to be clarified and preferably some references added. Also, clarify that just a selected number of studies using defined test variables are presented in table 1.

Some correction of terminology and language is required as suggested below.

I recommend publication after these minor revisions.

**Specific comments**

r. 150 Variation between year is weather variations as climate variations are defined as differences between 30-year periods.

r. 293 The reported period is wrong.

r. 467 I would avoid "loop" as having an unclear and even misleading meaning (instead "microbial (part of the) food web". How is "loop" justified? Microorganisms constitute the original food web in the biosphere and is an integral part of the modern food web. It both contribute to biomass flow and remove biomass by respiration. As most organisms.

r. 481 Correct to "betaproteobacteria". Indeed, all "proteo" phrases need correction.

r. 506 Please correct to "In the northern Quark, …"

r 907-909 This is likely relevant for the Baltic proper and Kattegat but less for the Gulf of Bothnia. The potential effects of simultaneously increasing DOC discharge is neglected in the scenario proposed.

r.970-973. If flagellate pressure is released bacterioplankton would increase. Please match with the scenario proposed in the last sentence. Also, "loop" preferably changed to "food web".

r. 979. More correctly expressed "… maintained bacterial biomass production despite reduced phytoplankton production…".

r. 1182-1185 There are more valuable mesocosm studies so a "(e,g, Lindh et al. 2015….)" would be appropriate or better adding some more examples covering larger parts of the food web (some suggested below).

r. 1216 Please add " , microbial food web….". Typically overlooked in current models.

r. 1308 I suggest to add: "Continuation and expansion of long term ecological studies in collaboration with environmental monitoring programs is also crucial for validating experimental results and advance our knowledge of environmental and meteorological drivers on large spatial and temporal scales."

r. 1340 Please change to "...the Gulf of Bothnia (..." as demonstrated also in the Bothnian Sea.

r.1388-1390 Should ecological safe fish catches be added here as a reliable and required measure?

r. 1427 I suggest to specify "...in the Baltic Sea for selected test variables". Several studies referred to in the text, and some covering other test variables, are omitted

End of comments.

**Suggested selected mesocosm references**

Amin, R. M., et al. (2012). "Partition of planktonic respiratory carbon requirements during a phytoplankton spring bloom." Marine Ecology-Progress Series **451**: 15-29.

> We studied the effect of variable phytoplankton biomass and dominance of the diatom Skeletonema marinoi on the planktonic community respiratory carbon requirement over a period of 14 d (14 to 28 April 2008) in 3 different mesocosms filled with natural water at Espegrend marine biological field station in Raunefjord, Norway. The carbon requirement was measured on mesozooplankton (the calanoid copepod Calanus finmarchicus) and 3 other size fractions of plankton-<200 mu m (dominated by microzooplankton), <15 mu m (dominated by nanoplankton including most of the phytoplankton) and particles passing GF/C filters (dominated by bacterioplankton)-by measuring oxygen consumption using an optode system with 2 Sensor Dish Readers. The respiratory carbon requirement showed no clear trend over time for any of the 4 groups. The mesozooplankton contributed the least to the total community carbon requirement, corresponding to <6% of primary production. In contrast, microzooplankton and nanoplankton consistently dominated the community carbon requirement, corresponding to >50% of the primary production, while bacterioplankton showed an intermediate and variable contribution (ca. <20% with a maximum of 50%). Feeding experiments on mesozooplankton (C. finmarchicus) 2 d before the peak in phytoplankton biomass showed that the copepods ingested from 2.4 to 4.3 times their respiratory carbon requirements, thus providing a high potential for growth. Respiratory carbon requirements of mesozooplankton were not significantly related to dominance or quantity of food available, whereas the respiratory carbon requirements of other groups were all related to the production of 22:6(n-3) fatty acid. The present study confirms the important role of microorganisms in the biological carbon transformation through the food web during a phytoplankton spring bloom.

Andersson, A., et al. (2013). "Can Humic Water Discharge Counteract Eutrophication in Coastal Waters?" Plos One **8**(4): 13.

> A common and established view is that increased inputs of nutrients to the sea, for example via river flooding, will cause eutrophication and phytoplankton blooms in coastal areas. We here show that this concept may be questioned in certain scenarios. Climate change has been predicted to cause increased inflow of freshwater to coastal areas in northern Europe. River waters in these areas are often brown from the presence of high concentrations of allochthonous dissolved organic carbon ( humic carbon), in addition to nitrogen and phosphorus. In this study we investigated whether increased inputs of humic carbon can change the structure and production of the pelagic food web in the recipient seawater. In a mesocosm experiment unfiltered seawater from the northern Baltic Sea was fertilized with inorganic nutrients and humic carbon (CNP), and only with inorganic nutrients (NP). The system responded differently to the humic carbon addition. In NP treatments bacterial, phytoplankton and zooplankton production increased and the systems turned net autotrophic, whereas the CNP-treatment only bacterial and zooplankton production increased driving the system to net heterotrophy. The size-structure of the food web showed large variations in the different treatments. In the enriched NP treatments the phytoplankton community was dominated by filamentous >20 mu m algae, while in the CNP treatments the phytoplankton was dominated by picocyanobacteria <5 mu m. Our results suggest that climate change scenarios, resulting in increased humic-rich river inflow, may counteract eutrophication in coastal waters, leading to a promotion of the microbial food web and other heterotrophic organisms, driving the recipient coastal waters to net-heterotrophy.

Bamstedt, U. and J. Wikner (2016). "Mixing depth and allochthonous dissolved organic carbon: controlling factors of coastal trophic balance." Marine Ecology Progress Series **561**: 17-29.

> The interacting effects of different mixing depths and increased allochthonous dissolved organic carbon (DOC) on the ratio of heterotrophic to autotrophic production (i.e. trophic balance) was evaluated in a mesocosm study with a stratified water column. An autumn plankton community from the northern

Bothnian Sea showed significantly decreased phytoplankton production and somewhat increased bacterial production with added DOC. In addition, increased mixing depth further reduced phytoplankton production. With a deep pycnocline and added DOC, the system became net-heterotrophic, with an average bacteria-to-phytoplankton production ratio of 1.24. With a deep pycnocline without added DOC, the trophic balance was changed to 0.44 (i.e. autotrophic). With a shallow pycnocline, the system remained net-autotrophic irrespective of DOC addition. We propose that increased precipitation in northern Europe due to climate change may result in changed density stratification and increased allochthonous DOC transport to the sea, leading to more heterotrophic coastal aquatic ecosystems. Such a scenario may entail reduced biological production at higher trophic levels and enhanced $CO_2$ emission to the atmosphere.

Dahlgren, K., et al. (2011). "The influence of autotrophy, heterotrophy and temperature on pelagic food web efficiency in a brackish water system." Aquatic Ecology **45**(3): 307-323.

Climate change has been suggested to lead to higher temperature and increased heterotrophy in aquatic systems. The aim of this study was to test how these two factors affect metazooplankton and food web efficiency (FWE was defined as metazooplankton production divided by basal production). We tested the following hypotheses: (1) that lower metazooplankton production and lower FWE would be found in a food web based on heterotrophic production (bacteria) relative to one based on autotrophic production (phytoplankton), since the former induces a larger number of trophic levels; (2) the metazooplankton in the heterotrophic food web would contain less essential fatty acids than those from the autotrophic food web; and (3) that higher temperature would lead to increased FWE. To test these hypotheses, a mesocosm experiment was established at two different temperatures (5 and 10A degrees C) with a dominance of either autotrophic (NP) or heterotrophic basal production (CNP). Metazooplankton production increased with temperature, but was not significantly affected by differences in basal production. However, increased heterotrophy did lead to decreased fatty acid content and lower individual weight in the zooplankton. FWE increased with autotrophy and temperature in the following order: 5CNP < 10CNP < 5NP < 10NP. Our results indicate that in the climate change scenario we considered, the temperature will have a positive effect on FWE, whereas the increase in heterotrophy will have a negative effect on FWE. Furthermore, the quality and individual weight of the metazooplankton will be reduced, with possible negative effects on higher trophic levels.

Lefebure, R., et al. (2013). "Impacts of elevated terrestrial nutrient loads and temperature on pelagic food-web efficiency and fish production." Global Change Biology **19**(5): 1358-1372.

Both temperature and terrestrial organic matter have strong impacts on aquatic food-web dynamics and production. Temperature affects vital rates of all organisms, and terrestrial organic matter can act both as an energy source for lower trophic levels, while simultaneously reducing light availability for autotrophic production. As climate change predictions for the Baltic Sea and elsewhere suggest increases in both terrestrial matter runoff and increases in temperature, we studied the effects on pelagic food-web dynamics and food-web efficiency in a plausible future scenario with respect to these abiotic variables in a large-scale mesocosm experiment. Total basal (phytoplankton plus bacterial) production was slightly reduced when only increasing temperatures, but was otherwise similar across all other treatments. Separate increases in nutrient loads and temperature decreased the ratio of autotrophic:heterotrophic production, but the combined treatment of elevated temperature and terrestrial nutrient loads increased both fish production and food-web efficiency. CDOM: Chl a ratios strongly indicated that terrestrial and not autotrophic carbon was the main energy source in these food webs and our results also showed that zooplankton biomass was positively correlated with increased bacterial production. Concomitantly, biomass of the dominant calanoid copepod Acartia sp. increased as an effect of increased temperature. As the combined effects of increased temperature and terrestrial organic nutrient loads were required to increase zooplankton abundance and fish production, conclusions about effects of climate change on food-web dynamics and fish production must be based on realistic combinations of several abiotic factors. Moreover, our results question established notions on the net inefficiency of heterotrophic carbon transfer to the top of the food web.

---

## Author Response (AR2)

2.3.2022

Manuscript "Global climate change and the Baltic Sea ecosystem: direct and indirect effects on species, communities and ecosystem functioning"
by Markku Viitasalo and Erik Bonsdorff, submitted to Earth System Dynamics.

Author comments for reviewer no. 2 (esd-2021-73-referee-report.pdf).

All author replies to reviewer comments in red font.

**Anonymous reviewer #2**

**General comments**

In general, the compilation of results are more distinct and text easy to read.

All the "To sum up,.." paragraphs are much appreciated. Significant improvement.

The fact that many more mesocosm studies relevant for climate-change effect has been reported need to be clarified and preferably some references added. Also, clarify that just a selected number of studies using defined test variables are presented in table 1.

Some correction of terminology and language is required as suggested below.

I recommend publication after these minor revisions.

We thank the reviewer for the positive comments. We have now taken improved our review on mesocosm studies and added the suggested references.

In the *Review methods* we mention that "...we have not reviewed all experimental studies that have dealt with environmental variables that may change with climate change." In Table 1 caption we also mention that the Table concerns "selected variables".

Terminology and language have been corrected as suggested. Some typos and linguistic corrections have been made.

We have also taken all specific comments into account (see below).

**Specific comments**

r. 150 Variation between year is weather variations as climate variations are defined as differences between 30-year periods.

Good point, and changed accordingly. We agree that "climate change" mainly refers to climatic variations in the scale of decades. We however retain the possibility to review studies considering slightly shorter-term variations (taking "several years"). Such changes, often associated with cyclic phenomena such as NAO, may give valuable information on responses of organisms to longer-term climatic variation as well.

Edited text: "We are not considering short-term (between-year or seasonal) weather variations, but mainly include studies that attempt to reveal organisms' responses to longer term (several years – decades) variability in climate."

r. 293 The reported period is wrong.

Typo on line 297 corrected.
Edited text: "Although data on biogeochemical parameters was not available for the period 1**9**03-1911..."
[Note: The period referred to on line 293 is correct. Hällfors et al. (2013) refers to periods 1903-1911 and 1993-2005.]

r. 467 I would avoid "loop" as having an unclear and even misleading meaning (instead "microbial (part of the) food web". How is "loop" justified? Microorganisms constitute the original food web in the biosphere and is an integral part of the modern food web. It both contribute to biomass flow and remove biomass by respiration. As most organisms.

We agree, and we have now replaced "microbial loop" to "microbial food web" throughout the ms.

r. 481 Correct to "betaproteobacteria". Indeed, all "proteo" phrases need correction.

"Prototerobacteria" changed to "proteobacteria".

r. 506 Please correct to "In the northern Quark, …"

Thank you for the comment. Following HELCOM vocabulary we now use "the Quark" to refer to "Kvarken" or "Norra Kvarken" (in Swedish). In this sentence we do not use the wording "in the northern Quark", because the study area of Nydahl et al. (2013) (and the Öre River, from where the freshwater of the study was taken) are located in the *southernmost* part of the Quark.

r 907-909 This is likely relevant for the Baltic proper and Kattegat but less for the Gulf of Bothnia. The potential effects of simultaneously increasing DOC discharge is neglected in the scenario proposed.

We agree, and define the basins we are speaking of. In Kattegatt, Cyanobacteria will probably not increase due to too high salinity.

New text: "This hypothesis concerns especially the Baltic Proper and the Gulf of Finland, perhaps also the southern Bothnian Sea."

r.970-973. If flagellate pressure is released bacterioplankton would increase. Please match with the scenario proposed in the last sentence. Also, "loop" preferably changed to "food web".

A valid comment. New text: "On the other hand, decreasing of bacteria would also decrease competition for nutrients between bacteria and phytoplankton, which could counteract the negative effects of diminishing remineralization on phytoplankton."

r. 979. More correctly expressed "… maintained bacterial biomass production despite reduced phytoplankton production…".

We agree. Edited text: "This provided additional substrate for bacteria, which maintained bacterial biomass production despite reduced phytoplankton production (Wikner and Andersson,

2012). This suggests that increased humic-rich river inflow may counteract climate change induced eutrophication in the coastal waters (Andersson et al., 2013)."

r. 1182-1185 There are more valuable mesocosm studies so a "(e,g, Lindh et al. 2015….)" would be appropriate or better adding some more examples covering larger parts of the food web (some suggested below).

A valid comment. We have changed the text accordingly.

New/edited text: "A few mesocosm studies have exposed the communities to near-natural environmental conditions and have been able to shed light on the complex dynamics of the Baltic Sea ecosystem, e.g., the responses of the microbial food web to changes of environmental variables affected by the climate change. In studies made in the Gulf of Bothnia, bacterial, phytoplankton and zooplankton production increased with additions of inorganic carbon, and the systems remained net autotrophic. In contrast, when both nutrients and DOC was increased, only bacterial and zooplankton production increased, driving the system to net heterotrophy (Andresson et al., 2013; Båmstedt and Wikner, 2016). Increased heterotrophy led to a decreased fatty acid content and lower individual weight in the zooplankton (Dahlgren et al., 2011). With the combined treatment of elevated temperature and terrestrial nutrient loads, also fish production (of three-spined sticklebacks) increased, with terrestrial and not autotrophic carbon being the main energy source (Lefébure et al., 2013). The complex responses indicate that, to provide useful inferences about physiological and population-level responses of organisms to climate change, experimental work should use full communities, apply naturalistic exposure regimes, and investigate effects of stress at spatial and temporal scales appropriate to the species studied (Gunderson et al., 2016)."

r. 1216 Please add " , microbial food web….". Typically overlooked in current models.

We agree. Edited text: "...there are major gaps for key trophic groups, such as macrophytes and macrozoobenthos (Korpinen et al., 2022) as well as the microbial food web."

r. 1308 I suggest to add: "Continuation and expansion of long term ecological studies in collaboration with environmental monitoring programs is also crucial for validating experimental results and advance our knowledge of environmental and meteorological drivers on large spatial and temporal scales."

A very good suggestion, thank you. We also note the importance of spatial mapping programs, since they can also be used for validating both experimental results and models.

New text: "Also, continuation of both spatial mapping programs and long term ecological studies will be crucial for validating experimental results and for developing ecosystem models, advancing our understanding of environmental and meteorological drivers of the Baltic Sea ecosystem on large spatial and temporal scales."

r. 1340 Please change to "…the Gulf of Bothnia (…" as demonstrated also in the Bothnian Sea.

We agree that the studies investigating the role of DOC discharge refer to an area covering the Bothnian Bay, the Quark and the northern Bothnian Sea (at least River Öre discharge area). Changed as suggested.

Edited text: "Increase of DOM flowing via the rivers may decrease both primary and secondary production, at least in the Gulf of Bothnia (Wikner and Andersson, 2012; Andersson et al., 2013)."

r.1388-1390 Should ecological safe fish catches be added here as a reliable and required measure?

A good point. Edited text: "Despite the many uncertainties concerning the effects of climate and eutrophication on the state of the Baltic Sea (Munkes et al., 2021), it can be stated that continued abatement of anthropogenic nutrient loading, combined with sustainable fisheries, seem to be the most reliable, albeit slow, measures to solve the grand challenges of the Baltic Sea (Meier et al., 2018; Murray et l., 2019).

r. 1427 I suggest to specify "…in the Baltic Sea for selected test variables". Several studies referred to in the text, and some covering other test variables, are omitted

Yes we agree. We however do not speak of "selected *test* variables" but "selected variables", as we also summarize long term studies.

Edited text: "Summary of research findings and conclusions on the anticipated effects of climate change (CC) effects in the Baltic Sea for selected variables. The table only shows studies published in 2011-2021 and a part of studies referred to in the text are not included."

**Suggested selected mesocosm references**

Amin, R. M., et al. (2012). "Partition of planktonic respiratory carbon requirements during a phytoplankton spring bloom." Marine Ecology-Progress Series 451: 15-29.

Andersson, A., et al. (2013). "Can Humic Water Discharge Counteract Eutrophication in Coastal Waters?" Plos One 8(4): 13.

Bamstedt, U. and J. Wikner (2016). "Mixing depth and allochthonous dissolved organic carbon: controlling factors of coastal trophic balance." Marine Ecology Progress Series 561: 17-29.

Dahlgren, K., et al. (2011). "The influence of autotrophy, heterotrophy and temperature on pelagic food web efficiency in a brackish water system." Aquatic Ecology 45(3): 307-323.

Lefebure, R., et al. (2013). "Impacts of elevated terrestrial nutrient loads and temperature on pelagic food-web efficiency and fish production." Global Change Biology 19(5): 1358-1372.

We have now included these references in appropriate places, also in Table 1. We however did not include Amin et al., 2012, since this paper does not specifically concern effects of climate change.